# Ligand and G-protein selectivity in the κ-opioid receptor

Jianming Han[1,2], Jingying Zhang[3,4,12,13], Antonina L. Nazarova[5,6,7], Sarah M. Bernhard[1,2], Brian E. Krumm[8], Lei Zhao[1], Jordy Homing Lam[5,6,7], Vipin A. Rangari[2], Susruta Majumdar[1,2,9], David E. Nichols[10], Vsevolod Katritch[5,6,7], Peng Yuan[3,4,12,13], Jonathan F. Fay[11 ✉] & Tao Che[1,2,9 ✉]

The κ-opioid receptor (KOR) represents a highly desirable therapeutic target for treating not only pain but also addiction and affective disorders[1]. However, the development of KOR analgesics has been hindered by the associated hallucinogenic side effects[2]. The initiation of KOR signalling requires the $G_{i/o}$-family proteins including the conventional ($G_{i1}$, $G_{i2}$, $G_{i3}$, $G_{oA}$ and $G_{oB}$) and nonconventional ($G_z$ and $G_g$) subtypes. How hallucinogens exert their actions through KOR and how KOR determines G-protein subtype selectivity are not well understood. Here we determined the active-state structures of KOR in a complex with multiple G-protein heterotrimers—$G_{i1}$, $G_{oA}$, $G_z$ and $G_g$—using cryo-electron microscopy. The KOR–G-protein complexes are bound to hallucinogenic salvinorins or highly selective KOR agonists. Comparisons of these structures reveal molecular determinants critical for KOR–G-protein interactions as well as key elements governing $G_{i/o}$-family subtype selectivity and KOR ligand selectivity. Furthermore, the four G-protein subtypes display an intrinsically different binding affinity and allosteric activity on agonist binding at KOR. These results provide insights into the actions of opioids and G-protein-coupling specificity at KOR and establish a foundation to examine the therapeutic potential of pathway-selective agonists of KOR.

Opioid receptors are G-protein-coupled receptors (GPCRs) that have important roles in pain sensation. Almost all clinically used opioids act through the μ-opioid receptor (MOR). However, their use is associated with severe side effects, including a high potential for abuse, addiction and death due to respiratory depression in overdose[3]. The magnitude of these problems has led to a search for opioid alternatives for the treatment of pain and related conditions[4]. The activation of opioid receptors recruits downstream effectors, including heterotrimeric G proteins (including Gα, Gβ and Gγ subunits) and β-arrestins. Specifically, opioid receptors primarily couple to the $Gα_{i/o}$ family ($G_{i1}$, $G_{i2}$, $G_{i3}$, $G_{oA}$, $G_{oB}$, $G_z$ and gustducin ($G_g$)) (Extended Data Fig. 1a). Some of these subtypes can mediate non-overlapping signalling pathways depending on the GPCR involved[5–8]. Whether signalling through individual pathways has redundant roles or separately drives the therapeutic efficacy and side effects of opioids remains mostly unclear.

KOR is a highly desirable therapeutic target for treating not only pain but also addiction and affective disorders. KORs have gained increasing attention owing to their unique analgesic activity—they are predominantly expressed in pain-related neurons, and drugs that target KOR

do not lead to addiction or cause death due to overdose as observed for MOR agonists[1]. The lack of rewarding/euphorigenic effects initially encouraged the development of KOR-agonist drugs as non-addictive analgesics[9]. Potent and selective KOR agonists have been developed, and these agonists produce effective peripheral and central analgesia. However, mood disorders such as dysphoria and psychotomimesis have been frequently observed as side effects of KOR agonists, which has limited their therapeutic application[2]. Here we determined the atomic structures of KOR in complex with different G-protein transducers and hallucinogenic ligands to help to elucidate the actions of opioids and the molecular basis for $Gα_{i/o}$ subtype selectivity.

## Overall structures of KOR–G-protein complexes

Although many efforts have been dedicated to the structural and molecular basis underlying the differences between G-protein and arrestin signalling, the roles of individual G-protein subtypes and the molecular determinants of subtype selectivity remain largely unclear. Sequence alignment of the seven $G_{i/o}$ subtypes suggests that they could

[1]Department of Anesthesiology, Washington University in St Louis, St Louis, MO, USA. [2]Center for Clinical Pharmacology, University of Health Sciences and Pharmacy in St Louis and Washington University School of Medicine, St Louis, MO, USA. [3]Department of Cell Biology and Physiology, Washington University School of Medicine, St Louis, MO, USA. [4]Center for the Investigation of Membrane Excitability Diseases, Washington University School of Medicine, St Louis, MO, USA. [5]Department of Quantitative and Computational Biology, University of Southern California, Los Angeles, CA, USA. [6]Department of Chemistry, University of Southern California, Los Angeles, CA, USA. [7]Center for New Technologies in Drug Discovery and Development, Bridge Institute, Michelson Center for Convergent Biosciences, University of Southern California, Los Angeles, CA, USA. [8]Department of Pharmacology, University of North Carolina School of Medicine, Chapel Hill, NC, USA. [9]Washington University Pain Center, Washington University in St Louis, St Louis, MO, USA. [10]Division of Chemical Biology and Medicinal Chemistry, Eshelman School of Pharmacy, University of North Carolina, Chapel Hill, NC, USA. [11]Department of Biochemistry and Molecular Biology, University of Maryland Baltimore, Baltimore, MD, USA. [12]Present address: Department of Pharmacological Sciences, Icahn School of Medicine at Mount Sinai, New York, NY, USA. [13]Present address: Department of Neuroscience, Icahn School of Medicine at Mount Sinai, New York, NY, USA. ✉e-mail: jfay@som.umaryland.edu; taoche@wustl.edu

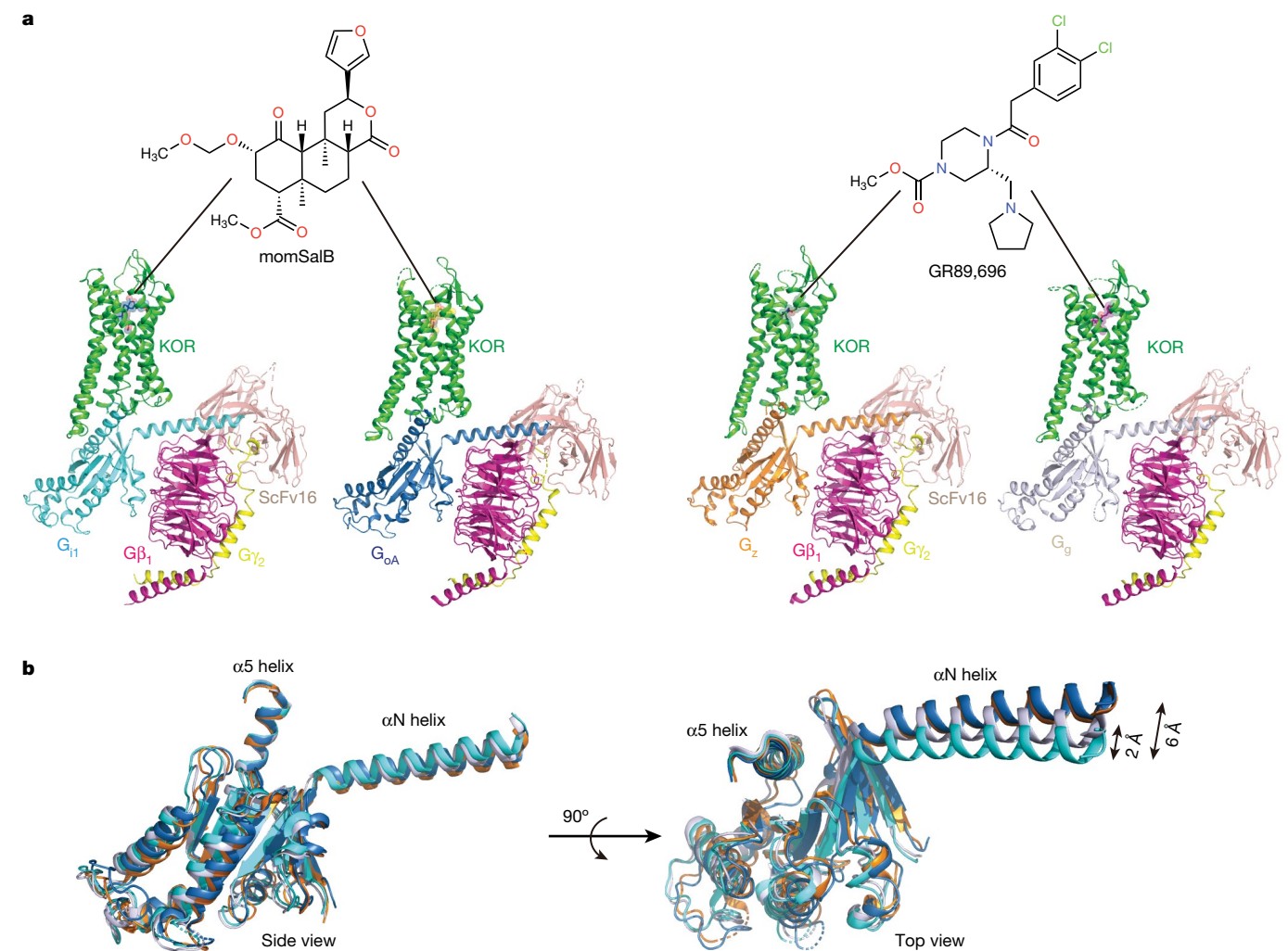

**Fig. 1 | Cryo-EM structures of KOR in complex with $G_{i/o}$ family subtypes.** **a**, Cartoon representations of KOR–G-protein complexes. Structures of KOR–$G_{i1}$ and KOR–$G_{oA}$ are bound to momSalB. Structures of KOR–$G_z$ and $G_g$ are bound to GR89,696. **b**, Structural alignment of the four Gα subunits. Distances of movement from the N terminus are labelled.

be further grouped into four subclasses on the basis of sequence identity ($G_{i1}$, $G_{i2}$ and $G_{i3}$; $G_{oA}$ and $G_{oB}$; and $G_z$ and $G_g$) (Extended Data Fig. 1b). To further understand the role of KOR–G-protein coupling and signalling, we determined the structures of KOR in complexes with four representative $G_{i/o}$ subtypes ($G_{i1}$, $G_{oA}$, $G_z$ and $G_g$) at nominal resolutions of 2.71 Å, 2.82 Å, 2.65 Å, and 2.61 Å, respectively, using single-particle cryo-electron microscopy (cryo-EM; Fig. 1a, Supplementary Fig. 1 and Extended Data Table 1). In particular, KOR–$G_{i1}$ and KOR–$G_{oA}$ are bound to a psychotropic salvinorin analogue, methoxymethyl-salvinorin B (momSalB)[10]. However, cryo-EM experiments of KOR–$G_z$ or KOR–$G_g$ bound to momSalB yielded only low-resolution reconstructions (resolution of around 4.5–5 Å) that prevented the delineation of detailed molecular interactions. Thus, we leveraged another highly potent KOR agonist, GR89,696 (ref. 11) to obtain high-resolution structures of KOR–$G_z$ and KOR–$G_g$.

The high-resolution maps of the four structures enabled unambiguous modelling of the agonist-bound heterotrimeric complexes (Supplementary Fig. 2). The overall differences between the four structures are subtle (root mean square deviations (r.m.s.d.) of 0.5 Å), with the exception of the Gα subunit in each complex (Fig. 1b). G-protein interactions with the receptor are canonically driven by the α5 and the N-terminal (αN) helices of the Gα subunit. The overlay of the four different G-protein subtypes showed that they adopt similar conformations

in the α5 helix but differ in the extent of movement in the αN helix (Fig. 1b). In particular, relative to $G_{i1}$, both $G_{oA}$ and $G_z$ exhibit a 6 Å displacement in the αN helix, whereas $G_g$ has a smaller 2 Å displacement. Notably, alignments of the MOR–$G_{i1}$ structure[12] with KOR–$G_{i1}$ indicate that the αN helix of $G_{i1}$ in MOR displays a position that is distinct from that of KOR–$G_{i1}$, whereas the α5 helix shows an orientation and interaction pattern similar to those in KOR (Extended Data Fig. 1c).

The overall structures of KOR in the $G_{i1/oA/z/g}$-bound states are similar to the previously reported nanobody-stabilized active conformation (KOR–Nb39)[13] (r.m.s.d., 0.8 Å) (Extended Data Fig. 1d). Notably, a comparison of these two structures reveals that the intracellular end of transmembrane helix 6 (TM6) in the KOR–$G_{i1}$ protein complex moves 2 Å closer to TM7. Nb39 stabilizing a different receptor conformation is further supported by its positive allosteric ability to enhance agonist binding affinity (Extended Data Fig. 1e). Another feature unique to G-protein-bound KOR is the presence of a well-defined intracellular loop 3 (ICL3) conformation that is absent in the Nb39-stabilized KOR, presumably due to its inherent flexibility (Extended Data Fig. 1d). Similar differences have also been captured between MOR–$G_{i1}$[12] or $β_2$AR–$G_s$[14] and their corresponding nanobody-stabilized active states[15,16], which further corroborate that a nanobody can stabilize a conformational state that mimics but is not identical to the G-protein-coupled state.

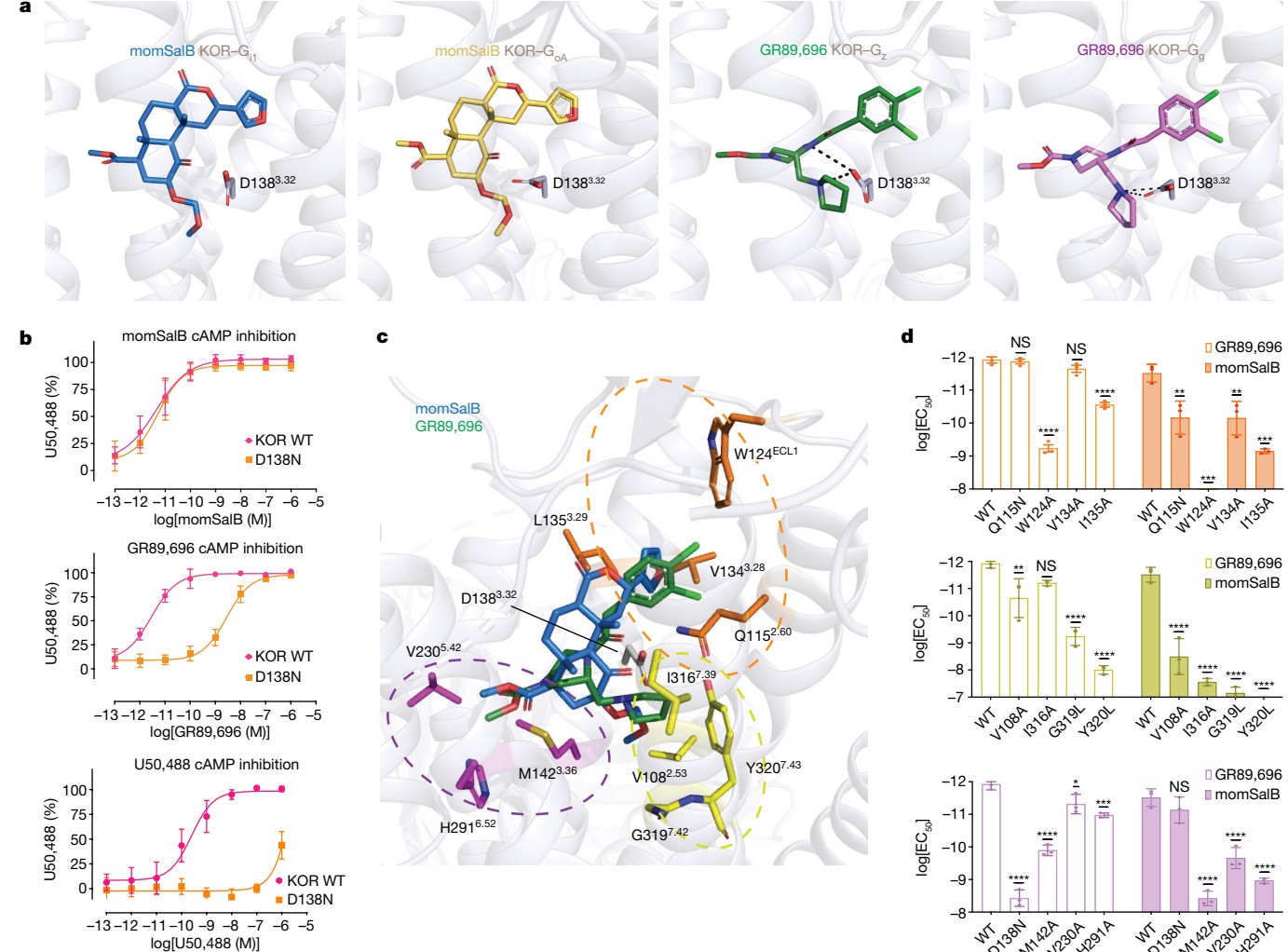

**Fig. 2 | Ligand-specific interactions with KOR. a**, The binding poses of momSalB and GR89,696 in their respective complex structures. The salt bridge or H-bond interactions in G$_z$- and G$_q$-coupled structures are shown as black dashed lines. This salt bridge or H-bond interaction is absent in momSalB-bound KOR. **b**, The highly conserved anchoring residue Asp138[3.32] has a different role in momSalB, GR89,696 or U50,488-mediated KOR activation. Data are normalized to the percentage of the reference agonist U50,488. Data are grouped data ± s.e.m. of $n = 3$ biological replicates. The full quantification parameters for this experiment are provided in Supplementary Table 1. **c**, Specific residues in the orthosteric pockets that interact with momSalB or GR89,696. Note that the I135[3.29]L mutation was included in KOR structure constructs to increase the expression level. **d**, Mutagenesis screening of

binding-pocket residues using G-protein-mediated cAMP inhibition assays. The effect on the potency of momSalB or GR89,696 was quantified on the basis of the log[EC$_{50}$] values. Data are log[EC$_{50}$] ± s.e.m. of $n = 3$ biological replicates. Statistical significance for each mutant was calculated using one-way analysis of variance (ANOVA) with Dunnett's multiple-comparison test compared with the wild type (WT); *$P < 0.05$, **$P < 0.01$, ***$P < 0.001$, ****$P < 0.0001$; NS, not significant. GR89,696: $P = 0.016$ (V230A), $P = 0.0008$ (H291A), $P = 0.1758$ (V134A), $P = 0.9814$ (Q115N), $P = 0.006$ (V108A), $P = 0.1165$ (I316A); momSalB: $P = 0.344$ (D138N), $P = 0.0009$ (W124A), $P = 0.0064$ (V134A), $P = 0.0068$ (Q115N), $P = 0.0002$ (I135A). The full quantification parameters for this experiment are provided in Supplementary Table 2.

## Interactions of KOR with hallucinogenic salvinorins

KORs have a prominent role in the modulation of human perception. Salvinorins, such as salvinorin A (SalA)[17,18], are a group of naturally occurring hallucinogens with dissociative effects elicited by activating the central KORs. momSalB is a semi-synthetic analogue of SalA and displays similar in vivo pharmacology compared to SalA[19,20]. GR89,696 is a potent and long-lasting KOR agonist that produces antinociception and dysphoria but with unknown hallucinogenic properties[21]. Different binding poses of momSalB and GR89,696 were observed in the KOR orthosteric pocket. This is consistent with their divergent chemical structures—GR89,696 is an alkaloid (containing basic nitrogen atoms) and momSalB is a terpenoid (lacking basic nitrogen atoms) (Fig. 1a). The pyrrolidine nitrogen atom in GR89,696, as well as many other ligands including KOR's endogenous dynorphin

ligands[22], is essential for the binding to KOR and enables the ligand to act as a hydrogen-bond (H-bond) donor and forms a salt bridge with the carboxylate side chain of Asp138[3.32] in the binding pocket (where the superscript values indicate Ballesteros–Weinstein numbering for GPCRs[23]) (Fig. 2a). As salvinorin ligands (such as momSalB) lack the basic nitrogen atom, there are no attractive electrostatic interactions observed between the salvinorins and Asp138[3.32]. Indeed, neither D138[3.32]A nor D138[3.32]N (the mutation in KOR DREADD[24]) showed detrimental effects in the binding affinity or agonistic potency of SalA, whereas both mutants abolished the interaction with endogenous dynorphin ligands[24–26]. The mutation D138[3.32]N resulted in a significant loss of potency in U50,488 and GR89,696, but had minimal effects on momSalB (Fig. 2b). The side chain of Asp138[3.32] pointing to the methoxymethyl group of momSalB also explains an interesting observation that D138[3.32]N could further enhance the binding affinity and potency

of SalA and salvinorin B (SalB)[24], probably due to the switch from the unfavourable acceptor–acceptor interaction to attraction resulting from the new H-bond interactions between the side chain of mutated asparagine and methoxy oxygen of the ligand.

Both momSalB and GR89,696 are highly selective and potent agonists at KOR (Fig. 2b and Extended Data Fig. 2a), making them ideal templates to investigate the molecular determinants for ligand selectivity and efficacy. Although the two agonists overlap in the orthosteric binding pocket of KOR, the core rings occupy different planes that are perpendicular to each other (Extended Data Fig. 2b). As a result, the subgroups of the two ligands form different interactions with residues in the corresponding subpockets (Fig. 2c). Mutations of the majority of residues in these subpockets reduced the agonist activity of momSalB or GR89,696, but with different amplitudes (for example, Val108$^{2.53}$, Gln115$^{2.60}$, Met142$^{3.36}$, Val230$^{5.42}$ or His291$^{6.52}$) (Fig. 2d and Extended Data Fig. 2c–e). The observation that the binding-pocket mutations have greater effects on momSalB-mediated cAMP inhibition than GR89,696 (for example, for H291$^{6.52}$A, Δlog[median effective concentration (EC50)$_{mutant-WT}$] = 2.23 ± 0.25 (momSalB) and 0.78 ± 0.27 (GR89,696)) is probably due to the lack of the anchoring interactions with Asp138$^{3.32}$, which makes salvinorins more sensitive to other residue contacts. The double mutation in KOR (for example, D138$^{3.32}$N and H291$^{6.52}$A, pEC$_{50}$ = 9.95 ± 0.06) displays a less deleterious effect on the potency of momSalB than H291$^{6.52}$A does (pEC$_{50}$ = 9.08 ± 0.06) alone (Extended Data Fig. 3a). A 2-fold to 2.5-fold improvement in potency was also observed from other mutations (Q115$^{2.60}$N or V230$^{5.42}$A) in combination with D138$^{3.32}$N when compared with the single mutation without D138$^{3.32}$N (Extended Data Fig. 3a). This effect might be specific to momSalB or salvinorin ligands as the double mutations (V230$^{5.42}$A/D138$^{3.32}$N or H291$^{6.52}$A/D138$^{3.32}$N) led to an inactive U50,488 or a further loss of potency for GR89,696-mediated cAMP inhibition in V230$^{5.42}$A/D138$^{3.32}$N (9,120-fold) or H291$^{6.52}$A/D138$^{3.32}$N (1,288-fold) compared with the respective single mutation (Extended Data Fig. 3a). Another major difference between momSalB and GR89,696 is that momSalB mainly forms hydrophobic interactions with residues that specifically contribute to the high potency of momSalB, such as Val108$^{2.53}$, Val134$^{3.28}$, Val230$^{5.42}$ and Ile316$^{7.39}$. In particular, Val108$^{2.53}$ has also been indicated as a determinant of ligand selectivity between KOR and MOR or DOR, as the latter two opioid receptors have an Ala$^{2.53}$ at the corresponding position[27]. Another hydrophobic pocket formed by the side chains of Val108$^{2.53}$ and Tyr320$^{7.43}$ and the backbone of Gly319$^{7.42}$ appears to be a key determinant for agonist activity and receptor activation, as mutations of these residues significantly decreased or eliminated signal transduction of momSalB with a threefold reduction in its ligand-binding affinity (Fig. 2d and Extended Data Fig. 3b). Notably, the amplitude of interactions with residues in this hydrophobic pocket positively correlates with agonist potency because ligands with more extended interacting groups—such as SalB (-O-H), momSalB (-O-CH$_2$-O-CH$_3$) and ethoxy-methyl SalB (-O-CH$_2$-O-CH$_2$-CH$_3$) (Extended Data Fig. 3c,d)—have displayed increased potency in activating KOR[19]. This subpocket at the bottom of the ligand-binding pocket acts as a potential allosteric connector to initiate the conformational changes of other microswitch motifs, including the sodium site, CW$^{6.48}$xxP and Pro$^{5.50}$-Ile$^{3.40}$-Phe$^{6.44}$ motifs[28,29].

The overall binding pose of GR89,696 in KOR–G$_z$ is similar to that in KOR–G$_g$. One notable difference is that GR89,696 forms stronger salt-bridge interactions with Asp138$^{3.32}$ (2.9 and 3.4 Å) in KOR–G$_z$ than those in KOR–G$_g$ (3.5 and 3.9 Å) (Fig. 2a and Supplementary Fig. 3), which probably contributes to the higher potency in activating G$_z$ compared with G$_g$ (ref. 30). Mapping the atomic distances between the ligand and receptor showed that GR89,696 makes closer contact with residues in the KOR–G$_z$ structure than in the KOR–G$_g$ structure (in terms of distance), whereas momSalB in KOR–G$_{i1}$ and KOR–G$_{oA}$ largely overlaps and displays similar strength (Supplementary Fig. 3).

For example, GR89,696 in KOR–G$_z$ also forms H-bond interactions with Gln115$^{2.60}$ (2.8 Å) and His291$^{6.52}$ (3.3 Å), and, in KOR–G$_g$, Gln115$^{2.60}$ (3.9 Å) and His291$^{6.52}$ (4.0 Å). This suggests that GR89,696 leads to more contractions of the ligand-binding pocket in the presence of G$_z$ compared with G$_g$.

## Structural basis of G-protein subtype selectivity

Similar to other opioid receptors, KOR exclusively couples to the G$_{i/o}$ family[30], including the canonical G$_{i/o}$ subtypes (G$_{i1}$, G$_{i2}$, G$_{i3}$, G$_{oA}$ and G$_{oB}$) and the noncanonical G$_z$ and G$_g$. Whereas G$_z$ is predominantly expressed in the central nervous system, G$_g$ is the endogenous transducer of taste receptors, such as the bitter taste receptor 2 (TAS2R). Mice expressing engineered KORs in bitter-receptor cells show a strong aversion to a designed KOR agonist (inert to endogenous wild-type KOR but active in engineered KOR)[31], suggesting that the KOR–G$_g$ interaction and signalling may also occur in vivo. Using bioluminescence resonance energy transfer (BRET)-based transducerome profiling (Fig. 3a), we confirmed that both momSalB and GR89,696 could activate all four G-protein subtypes, although with different potencies (Fig. 3b). The primary interaction sites in KOR bound to different G$_{i/o}$ subtypes involve nearly the entire intracellular regions of the receptor (ICL2, ICL3, TM3, TM5, TM6, TM7 and helix 8) and the αN and α5 helices of the Gα subunits (Extended Data Fig. 4). The key residues involved in KOR–G-protein interactions were mapped (Extended Data Fig. 4) and screened by alanine substitutions. In this section, we first report the effects of interface residues from the KOR side and then the residues from the Gα protein side.

Although the KOR conformations in each G-protein-bound structure are similar to each other, notable differences were observed for the KOR residues involved in receptor–G-protein interactions. Mutagenesis screening using G-protein-mediated cAMP inhibition assays suggested that almost all of the residues on the KOR side contribute to KOR–G-protein signalling (Extended Data Fig. 5a,b). That was further confirmed by the BRET-based transducer profiling, which showed that mutations of these residues on the intracellular KOR side decreased agonist-mediated G$_{i1}$, G$_{oA}$, G$_z$ and G$_g$ coupling (Extended Data Fig. 5c and Supplementary Fig. 4). Although most of the residues in the KOR interface affect the four G-protein couplings in a similar manner (Supplementary Fig. 4), some display subtype selectivity. Arg156$^{3.50}$ is a highly conserved residue in the classic Asp$^{3.49}$-Arg$^{3.50}$-Tyr$^{3.51}$ motif that has been implicated in having an important role in receptor activation and signal transduction (Fig. 3c). An 'ionic lock' has been frequently observed between Arg$^{3.50}$ and Glu$^{6.30}$ in class A GPCRs, keeping the receptor in an inactive state with TM3 and TM6 in close proximity. Thus, the breaking of this ionic lock is an important step towards the coupling of G proteins, as the TM6 movement away from TM3 is critical for penetration of the G-protein α5 helix into the cytoplasmic pocket[12]. The R156$^{3.50}$A mutation significantly reduced the potency of agonist-mediated activation by momSalB or GR89,696 (Fig. 3d). Furthermore, R156$^{3.50}$A specifically reduced the efficacy of agonist-mediated G$_g$ activation, momSalB (WT (106 ± 3%) versus R156$^{3.50}$A (38 ± 5%)) or GR89,696 (WT (103 ± 3%) versus R156$^{3.50}$A (46 ± 7%)) (Fig. 3d and Extended Data Fig. 5d). In the inactive-state KOR[32], the partially formed ionic lock is between Arg156$^{3.50}$ and Thr273$^{6.34}$; in the fully active KOR–agonist–G-protein states, this interaction is broken due to the insertion of the α5 helix of Gα protein, leading to the release of the side chain of Arg156$^{3.50}$ to extend towards TM7 and form hydrophobic interactions with the second-to-last leucine (Leu353$^{H5.25}$ in Gα$_{i1}$, Gα$_{oA}$ or Gα$_g$; Leu354$^{H5.25}$ in Gα$_z$) of the Gα subunits (superscript notes for G proteins represent the CGN numbering system[33]) (Fig. 3c). This is further supported by the molecular dynamics simulations showing that the KOR-Arg156$^{3.50}$ can form hydrophobic interactions with Gα-L353$^{H5.25}$ or Leu354$^{H5.25}$ maintaining <4 Å distances with these side chains (Extended Data Fig. 6). In this extended conformation, the KOR-Arg156$^{3.50}$ guanidine

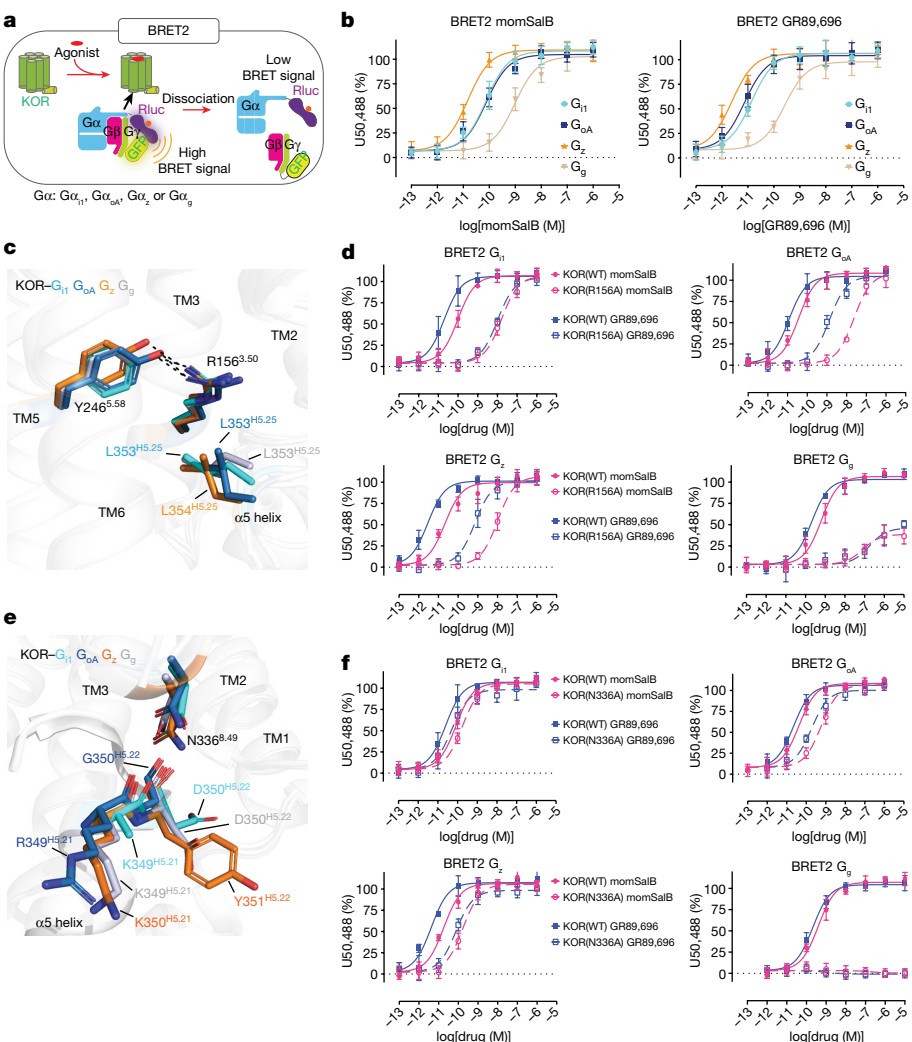

**Fig. 3 | Comparison of the receptor–G-protein-binding interface of the KOR–G$_{i1}$, KOR–G$_{oA}$, KOR–G$_z$ and KOR–G$_g$ complexes. a**, Schematic of the BRET2 assay. **b**, momSalB- or GR89,696-mediated G-protein-subtype activation measured by BRET2. Data are grouped data ± s.e.m. of $n$ = 4 biological replicates. The full quantification parameters for this experiment are provided in Supplementary Table 3. **c**, Interactions of Arg156$^{3.50}$ in the Asp (D)-Arg (R)-Tyr (Y) motif with KOR and Gα. **d**, Mutagenesis analysis of Arg156$^{3.50}$ by BRET2. Data

are grouped data ± s.e.m. of $n$ = 3 biological replicates. The full quantification parameters for this experiment are provided in Supplementary Table 4. **e**, Interactions of Asn336$^{8.49}$ in different KOR–g-protein complexes. **f**, The N336$^{8.49}$A mutation differentially affects KOR-mediated G-protein subtype activation. Data are global fit of grouped data ± s.e.m. of $n$ = 3 independent biological replicates. The full quantification parameters for this experiment are provided in Supplementary Table 4.

group also forms a persistent H bond with Tyr246$^{5.58}$ observed in all four complexes (Fig. 3c and Extended Data Fig. 6). These data suggest that the KOR-Arg156$^{3.50}$ has an important role in KOR activation by directly interacting with the G proteins. A recent study also suggested that G proteins might need to precouple to the receptor and break the Arg156$^{3.50}$-mediated ionic interaction before agonist binding and signalling[34]. Other important interactions are formed by residue Asn336$^{8.49}$ in helix 8 of KOR, which engages different H-bond interactions with the backbone of the α5 helix in each Gα protein, such as Lys/Arg$^{H5.21}$ and Gly/Asp/Tyr$^{H5.22}$ (Fig. 3e and Extended Data Fig. 4). The molecular dynamics simulations also provide support for these interactions, suggesting dynamic patterns of switches between specific interaction pairs (Extended Data Fig. 7). The mutation N336$^{8.49}$A completely abolished KOR–G$_g$ coupling (for example, momSalB), and a 2-fold loss in potency in G$_{i1}$, 14-fold in G$_{oA}$ or 9-fold in G$_z$ coupling (Fig. 3f). Together with the effects observed from Arg156$^{3.50}$ and Asn336$^{8.49}$, these data indicate that these residues have differential roles in G-protein association, probably by engaging at different intermediate stages. The observation that several mutations have the largest effect on G$_g$

compared with the other G$_{i/o}$ subtypes suggests a non-canonical role of G$_g$ in KOR-mediated signalling.

Next, we examined the Gα subunit by mutating the non-conserved residues in the αN or α5 helix to alanine (Extended Data Fig. 8a). However, we did not observe significant changes in the potency of agonist-mediated G-protein activation in BRET2 assays (Extended Data Fig. 8b–f and Supplementary Fig. 5). One exception is that C351$^{H5.23}$A in Gα$_{i1/oA/g}$ or I352$^{H5.23}$A in Gα$_z$ led to a significant decrease in potency for momSalB or GR89,696-mediated G-protein activation. This Ile352$^{H5.23}$ in Gα$_z$, compared with the corresponding Cys351$^{H5.23}$ in other G$_{i/o}$ subtypes, is known as the site that makes Gα$_z$ insensitive to pertussis toxin. The relative conformation of the α5 helix has been implicated as a key determinant between G$_s$ and G$_i$ specificity, in which the α5 helix adopts distinct positions and results in a larger outward movement of TM6 (13 Å in β$_2$AR–G$_s$ versus 9 Å in MOR–G$_{i1}$)[12,35,36]. The subtle differences in the α5 helix conformation of Gα$_{i1/oA/z/g}$ and the mutational evidence suggest that the G-protein-coupling specificity in the G$_{i/o}$ family is probably determined by a more complex and/or dynamic three-dimensional network interaction[37,38].

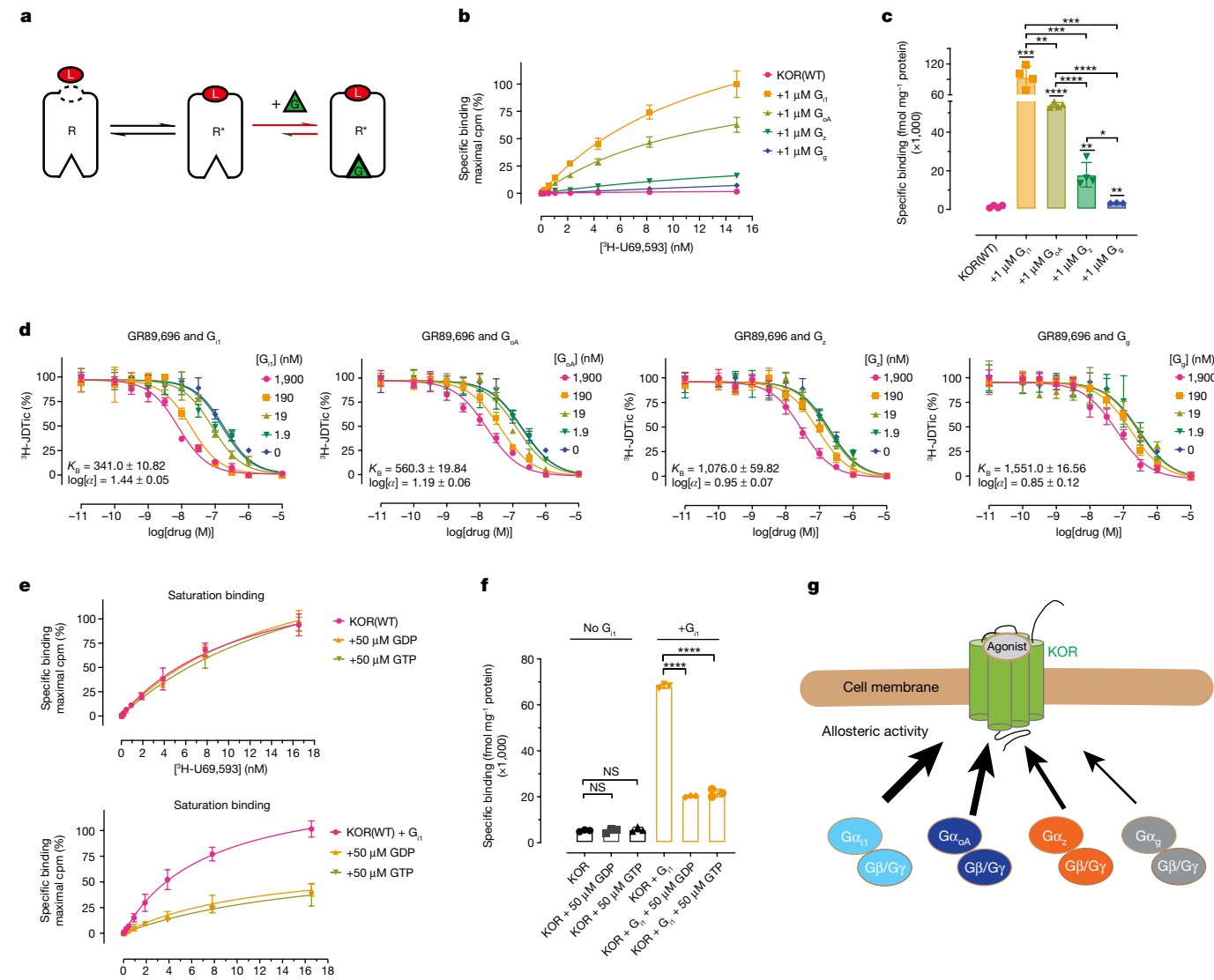

**Fig. 4 | The intrinsic differences of individual G-protein subtypes.**
**a**, Schematic of the GPCR–G-protein–ligand ternary model. G, G protein; L, agonist; R, receptor. **b**, KOR saturation binding reveals that G proteins potentiate agonist binding with different amplitudes. Data are global fit of grouped data ± s.e.m. from $n = 4$ independent biological replicates. **c**, Summary of $B_{max}$ values in the presence of G proteins. Statistical analysis between groups was performed using the unpaired two-tailed Student's $t$-tests; $P = 0.0001$ (KOR + $G_{i1}$ versus KOR), $P = 0.0021$ (KOR + $G_z$ versus KOR), $P = 0.0075$ (KOR + $G_g$ versus KOR), $P = 0.0081$ (KOR + $G_{i1}$ versus KOR + $G_{oA}$), $P = 0.0004$ (KOR + $G_{i1}$ versus KOR + $Gz$), $P = 0.0007$ (KOR + $G_{i1}$ versus KOR + $G_g$) and $P = 0.0116$ (KOR + $G_z$ versus KOR + $G_g$). **d**, Competition binding reveals that G proteins have

a different binding affinity and allosteric activity on KOR. Data are global fit of grouped data ± s.e.m. of $n = 3$ independent biological replicates. **e**, The effect of GDP or GTP on the allosteric activity of G proteins. Data are global fit of grouped data ± s.e.m. of $n = 3$ independent biological replicates. **f**, Summary of $B_{max}$ values in the presence or absence of GDP/GTP. Statistical analysis was performed using unpaired two-tailed Student's $t$-tests compared with the KOR or KOR + $G_{i1}$ group; $P = 0.9874$ (KOR + 50 μM GDP versus KOR), $P = 0.4147$ (KOR + 50 μM GTP versus KOR). **g**, A representative model of different allosteric activity of $G_{i/o}$ family subtypes ($G_{i1} > G_{oA} > G_z > G_g$). The full quantification parameters for the experiments in **b**, **d** and **e** are provided in Supplementary Tables 5–7, respectively.

The overall interfaces of $Gα_{i1}$, $Gα_{oA}$, $Gα_z$ and $Gα_g$ with KOR are highly conserved (Extended Data Fig. 4), but there are critical differences in the α5 and αN helices. The major contacts made by $Gα_{oA}$ with KOR are through residues in the α5 helix (Extended Data Fig. 4b), whereas contacts made by $Gα_{i1}$, $Gα_z$ and $Gα_g$ involve regions in both the α5 and αN helices (Extended Data Fig. 4a,c,d). Similarly, the 5HT$_{1B}$R–G$_o$[39] interaction is also mediated solely by the α5 helix, but a structural comparison between KOR–$Gα_{oA}$ and 5HT$_{1B}$R–G$_o$ shows that the α5 helix in 5HT$_{1B}$R–G$_o$ tilts an additional 9°, leading to a larger 3 Å outward movement of TM6 (Extended Data Fig. 9a). Alignment of the cytoplasmic regions of KOR–$Gα_{i1}$, β$_2$AR–G$α_s$ and 5HT$_{2A}$R–G$α_q$ shows that the α5 helices are positioned differently. There are 6° and 12° tilts of the C-terminal end away from the

plane of the membrane compared with $Gα_q$ and $Gα_s$, respectively, leading to different magnitudes of outward movement of TM6 (Extended Data Fig. 9b). As a result of intracellular conformational differences, the KOR–$Gα_{i1}$ forms an interface area of 1,219 Å$^2$ (Extended Data Fig. 4a), compared with a slightly larger area of β$_2$AR–G$α_s$ (1,260 Å$^2$) and a much smaller area of 5HT$_{2A}$R–G$α_q$ (1,077 Å$^2$) (Extended Data Fig. 9c). Whereas $Gα_{i1}$, $Gα_z$ and $Gα_g$ have similar interface areas (1,219, 1,262 and 1,221 Å$^2$, respectively) (Extended Data Fig. 4a,c,d), $Gα_{oA}$ has a much smaller area (1,096 Å$^2$) (Extended Data Fig. 4b). Notably, the 822 Å$^2$ surface area of $G_o$ in contact with 5HT$_{1B}$R[39] is closer to that of KOR and $Gα_{oA}$ compared with other G-protein subtypes, suggesting a shared mechanism between different GPCRs and the same G protein.

## Intrinsic differences in G-protein subtypes

GPCR signalling is transduced through the allosteric changes between the extracellular ligand pocket and the intracellular G-protein-binding pocket. Conformational changes induced by agonist binding can enhance the binding affinity of G-protein heterotrimers. Conversely, G protein acts as a positive allosteric modulator and further enhances agonist-binding affinity by stabilizing the ternary complex[40] formed by the receptor, ligand and G proteins (Fig. 4a). We next sought pharmacological evidence to test whether G-protein subtypes have intrinsic differences, including binding affinity and allosteric activity at KOR in the presence of agonists. On the basis of the ternary complex model[41], the high-affinity agonist-binding states should increase in the presence of G-protein heterotrimers, as the latter can stabilize the active-state receptor favouring agonist binding. We performed saturation binding assays to test the binding of agonist radioligand $^3$H-U69,593 to KOR in the presence of $G_{i1}$, $G_{oA}$, $G_z$ and $G_g$. Notably, the four G proteins display substantial differences in the allosteric enhancement of agonist binding (Fig. 4b,c). Compared with the wild type alone ($B_{max} = 1,350 \pm 116$), the high-affinity binding sites for $^3$H-U69,593 were increased 62-, 38-, 14- and 7-fold in the presence of $G_{i1}$ ($B_{max} = 84,324 \pm 4,214$), $G_{oA}$ ($B_{max} = 52,086 \pm 2,465$), $G_z$ ($B_{max} = 18,623 \pm 1,468$) and $G_g$ ($B_{max} = 9,866 \pm 3,493$), respectively. These data are consistent with the ternary model that at least two binding states predominate in the unliganded receptor[42]—a high-affinity (G-protein-coupled) and a low-affinity (G-protein-uncoupled) binding state. We also compared the wild-type G proteins with the engineered G proteins used in our structural determination and observed similar patterns of $B_{max}$ increases (Extended Data Fig. 10b).

The different magnitudes of $B_{max}$ increases among G-protein subtypes suggest that individual G proteins have different allosteric effects on ligand binding. To test this hypothesis, we next compared the cooperativity of the four G-protein subtypes by radioactive competition binding assays designed to quantify the G-protein binding affinity ($K_B$) and cooperativity ($\alpha$, G-protein cooperativity value, $\alpha > 1$ indicates positive effects in increasing agonist affinity)[43]. The G-protein-insensitive antagonist radioligand $^3$H-JDTic was used for the following series of experiments. The inhibition of $^3$H-JDTic binding at KOR by GR89,696 progressively improved as the concentration of G protein increased, indicating positive cooperativity between G proteins and agonist binding (Fig. 4d). The calculated $K_B$ and $\alpha$-cooperativity displayed a pattern similar to that observed in the saturation binding, in which $G_{i1}$ has the highest binding affinity and allosteric effects at KOR in the presence of agonists, and $G_g$ has the least (Fig. 4d). The different binding affinities could have a role in G-protein-subtype selectivity, as G-protein subtypes with higher affinity can outcompete other G-protein subtypes, depending on subtype abundance, especially in cells expressing several or all $G_{i/o}$ family subtypes.

Guanosine diphosphates (GDPs) or guanosine triphosphates (GTPs) are important regulators of GPCR–G-protein assembly and signalling[44]. We therefore examined whether the presence of GDP or GTP can affect the allosteric activity of G-protein subtypes. The specific binding of $^3$H-U69,593 was significantly reduced in the presence of GDP or GTP compared with the nucleotide-free state (Fig. 4e,f). The four G proteins exhibited a uniform pattern, showing similar responses to GDP or GTP (Extended Data Fig. 10c). These nucleotide-specific effects are consistent with the results of single-molecule studies of the $\beta_2$AR–$G_s$ complex showing that the presence of GDP or GTP accelerates the dissociation of $\beta_2$AR and the $G_s$ heterotrimer[45], which is achieved through a sequential conformational change in the G$\alpha$ subunit after the binding of GDP or GTP[44,46]. Note that the dominant negative $G_{i2}$ has been reported to abolish GTP binding and GTPase activity[47]; however, the engineered G proteins in this study appear to maintain the GTP binding affinity and GTP turnover activity, although at weaker levels compared with the wild type (Extended Data Fig. 10a).

## Discussion

After activation, KOR can interact with up to seven G proteins, the coupling of which determines the direction of ligand-induced signalling. The seven G-protein subtypes are highly homologous but not structurally or functionally identical. The binding of signal transducers is coupled with specific receptor conformational changes, such as the magnitude of TM6 displacement. Such conformational differences have been observed in GPCRs bound to the $G_s$, $G_i$ and $G_q$ families. However, analysis of the four structures of KOR in complex with different $G_{i/o}$ subtypes shows that the receptor adopts a similar conformation. In particular, the receptor conformations in KOR–$G_z$ and KOR–$G_g$ are nearly identical, although the bound agonist GR89,696 activates the two G proteins with a 100-fold difference in potency. This structural observation agrees with an original postulation that the cross-reactivity between receptors and G proteins speaks to the conservation of structure among the receptor-binding domains of the G proteins and the G-protein-binding domains of the receptors[48]. This conformational similarity, irrespective of transducer subtypes, has also been observed in other GPCRs engaging G proteins versus arrestins[49]. These subtle differences could be due to the limitation of the structures that reveal a well-resolved population of the nucleotide-free G-protein-bound conformational state of the receptor. These nucleotide-free states of G$\alpha$ subunits (G$\alpha_{i1}$, G$\alpha_{oA}$, G$\alpha_z$ and G$\alpha_g$) tend to stabilize a specific conformational state of KOR. In the absence of G proteins or in the presence of nucleotide-bound G proteins, the receptor can adopt dynamic conformations that are different from that captured by nucleotide-free G$\alpha$[45]. Other approaches, including nuclear magnetic resonance (NMR)[50] and molecular dynamics simulations[51], have identified dynamic conformational states in the intracellular regions of the receptor related to transducer couplings in the presence of GDP or GTP.

Different GPCR–G-protein interfaces have been proposed to contribute to the differential kinetics of G proteins during association and dissociation with the receptor[52,53]. Our pharmacological characterization of the KOR–G-protein interface identified key residues that have different roles in G-protein coupling. The complexes that we targeted in this study displayed varying interface areas dependent on the receptors and G proteins. However, time-resolved studies are needed in the future for the direct measurement of G-protein association and dissociation rates, especially for different $G_{i/o}$ subtypes, as the strength of the receptor–G-protein interface may be another factor that affects the coupling efficiency.

In the structures of KOR in complex with different G-protein subtypes, we also revealed the binding poses of two selective KOR agonists—momSalB and GR89,696. Although they occupied the same binding pocket, they adopted different conformations and interacting patterns, probably due to their unique chemical structures. GR89,696 displays stronger interactions in KOR–$G_z$ than in KOR–$G_g$, which may contribute to its higher potency observed in the BRET2 assay. Owing to the unique scaffold and pharmacology of salvinorin ligands, extensive studies have been conducted to elucidate their binding and function[13,25,26]. Several residues or motifs in KOR (for example, Val108[2.53], His291[6.52], Ile316[7.39]) have been identified that are important for salvinorin's agonism, which can now be explained by their direct interactions with momSalB. Notably, using multiple structural templates, previous salvinorin docking suggested different binding poses[13,25,26], and our structures now provide direct evidence of how momSalB sits in the binding pocket of KOR.

We also observed allosteric differences among these highly conserved G-protein subtypes. As positive allosteric modulators, the four representative G proteins display distinct allosteric activity in potentiating agonist binding ($G_{i1} > G_{oA} > G_z > G_g$) (Fig. 4g). This is consistent with measurements of the binding affinity of different G proteins ($G_{i1} > G_{oA} > G_z > G_g$) at KOR. These intrinsic differences in G proteins, including the binding affinity and coupling efficiency, add

pharmacological evidence to determinants for G-protein-subtype selectivity. Our structural observations from different G proteins in complex with KOR show that the $G_{i/o}$ family subtypes share a highly conserved mechanism in interacting with KOR, but that each maintains pharmacological differences. Considering that many GPCRs can couple to different G-protein families, such as the $\beta_2AR$ coupling with both $G_s$ and $G_{i/o}$ (refs. 38,54), whether $\beta_2AR$ displays differential binding affinities with $G_s$ and $G_{i/o}$ may help to explain its G-protein coupling specificity.

Furthermore, the allosteric activity of G proteins can be regulated by GDPs or GTPs that decouple G proteins from the receptor. It is known that nucleotide-specific conformations exist between nucleotide-free G proteins and GDP- or GTP-bound G proteins. When comparing the crystal structure of the uncoupled GDP-bound $G_{i1}$ heterotrimer[55] and nucleotide-free (KOR–$G_{i1}$, $G_{oA}$, $G_z$ and $G_g$) heterotrimers (Extended Data Fig. 11a), several conformational displacements are noted (Extended Data Fig. 11b). The activated receptor engages the C terminus of the $\alpha5$ helix of $G\alpha_{i1}$, which undergoes an upward helical extension (8.6 Å) into the receptor core (Extended Data Fig. 11c) compared with the uncoupled G protein structure. The insertion of the $\alpha5$ helix into the transmembrane helical bundle of the receptor has the following two consequences. First, the loop connecting the $\alpha5$ helix and $\beta6$ strand moves outward 5 Å. Second, the movement of the $\alpha5$ helix disrupts the original hydrophobic interactions between the $\alpha5$ and $\alpha1$ helices, leading to a displacement of the P loop. Both the P loop translocation and loss of coordination with GDPs are necessary for GDP release[44,45]. In agreement with the ternary complex model, our saturation binding data show that GDPs or GTPs act as negative regulators of agonist binding kinetics in the presence of G proteins. One limitation of this study is that we used an in vitro overexpressed system with engineered receptors and G proteins to measure ligand activity, which cannot be extended to in vivo without further experiments.

In summary, we have elucidated the molecular interaction details between highly conserved $G_{i/o}$ subtypes and KOR using cryo-EM-derived atomic models. We have also examined the structural determinants of ligand selectivity and efficacy at KOR. Using structural pharmacology analysis, we revealed the intrinsic differences between these previously under-represented $G_{i/o}$ subtypes and demonstrated that subtype selectivity is probably a combinational result of receptor conformational dynamics, the binding affinity of G proteins and cooperativity between agonist binding and G-protein coupling. Such findings are important both in understanding GPCR-mediated signalling and in the generation of new research tools and therapeutics based on the potential of G-protein-selective agonists.

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

## Methods

### Generation of constructs for cryo-EM

For the human KOR, we used a construct the same as the previously determined active-state KOR[13]. In brief, the construct (1) lacks N-terminal residues 1–53; (2) lacks C-terminal residues 359–380; (3) contains Met1–Leu106 of the thermostabilized apocytochrome b562 RIL (BRIL) from *E. coli* (M7W, H102I, R106L) in place of receptor N terminus residues Met1–His53. This N-terminal Bril will be removed using a PreScission cleavage site in the end. The single chain Fab scFv16 has the same sequence as previously reported[56]. A 6×His tag was added to the C-terminal scFv16 sequence with a PreScission cleavage site inserted between. For the G-protein heterotrimers, individual G-protein constructs ($G_{i1}$, $G_{oA}$, $G_z$ and $G_g$) were engineered (labelled as dominant negative, DN)[47] for the binding of scFv16, and then subcloned into a designed vector that co-expresses the $G\beta_1$ and $G\gamma_2$. Further modifications were made to enable a stable complex between KOR, G-protein heterotrimer and scFv16. Specifically, $G_{i1}$(DN) includes S47N, E245A, G203A and A326S. $G_{oA}$(DN) includes C3S, S47N, G204A, E246A, A326S and M249K. For $G_z$(DN), the N-terminal sequence was replaced with the $G_{i2}$ sequence to allow for better interaction with scFv16; other mutations include S47N, G204A, E246A, R249K, N262D and A327S. For $G_g$(DN), the N-terminal sequence was replaced with the $G_{i2}$ sequence; other mutations include S47A, G203A, E245A, H248K, T261D, A326S and N251D.

### Expression of KOR–G-protein–scFv16 complex

The Bac-to-Bac Baculovirus Expression System was applied to generate high-quality recombinant baculovirus (>10$^{-9}$ viral particles per ml) for protein expression (KOR, G-protein heterotrimers and scFv16). For the expression of KOR–G–scFv16 protein complex, each heterotrimeric G protein, including Gα ($G_{i1}$, $G_{oA}$, $G_z$ or $G_g$), $G\beta_1$ and $G\gamma_2$ was coexpressed with KOR and scFv16, respectively, by infection of *Spodoptera frugiperda* Sf9 cells at a cell density of $2.5 \times 10^6$ cells per ml in ESF921 medium (Expression System) with the P1 baculovirus at a multiplicity of infection (MOI) ratio of 2:2:0.5. Cells were collected by centrifugation (125 rpm at 27 °C) for 48 h after infection, washed with HN buffer (25 mM HEPES pH 7.4, 100 mM NaCl), and stored at −80 °C for future purification.

### Purification of the KOR–G-protein–scFv16 complex

The compounds used in this study—(−)-U50,488 (0496) and GR89,696 (1483)—were purchased from Tocris. momSalB was synthesized by a method described previously[57]. After purification by silica gel column chromatography, momSalB was a single spot on TLC (silica, 20% ethyl acetate, dichloromethane) with an $R_f$ of 0.49. An NMR spectrum of momSalB was collected to confirm the chemical identity (Supplementary Fig. 6), which is consistent with the expected spectrum reported previously[58].

We thawed the cell pellet and incubated it in buffer containing 20 mM HEPES pH 7.5, 50 mM NaCl, 1 mM MgCl$_2$, 2.5 units Apyrase (NEB), 10 μM agonist (final concentration) and protease inhibitors (500 mM AEBSF, 1 mM E-64, 1 mM leupeptin, 150 nM aprotinin) for 1.5 h at room temperature. We then collected the membrane by centrifugation at 25,000 rpm for 30 min at 4 °C. The membrane was solubilized in buffer (40 mM HEPES pH 7.5, 100 mM NaCl, 5% (w/v) glycerol, 0.6% (w/v) lauryl maltose neopentyl glycol (LMNG), 0.06% (w/v) cholesteryl hemisuccinate (CHS), 10 μM agonist and protease inhibitors) with 200 μg scFv16 in the cold room. After 5 h, the supernatant was collected by centrifugation at 30,000 rpm for 30 min at 4 °C and incubated with 1 ml TALON IMAC resin (Clontech) and 20 mM imidazole overnight in the cold room. The next day, the resin was collected and washed with 10 ml buffer containing 20 mM HEPES pH 7.5, 100 mM NaCl, 30 mM imidazole, 0.01% (w/v) LMNG, 0.001% (w/v) CHS, 5% glycerol and 5 μM agonist. The protein was then eluted with the same buffer supplemented with 300 mM imidazole, concentrated and further purified by size-exclusion chromatography on the Superdex 200 increase 10/300 column (GE healthcare), which was pre-equilibrated with 20 mM HEPES pH 7.5, 100 mM NaCl, 100 μM TCEP, 0.00075% (w/v) LMNG, 0.00025% (w/v) glyco-diosgenin (GDN) and 0.00075% (w/v) CHS, 1 μM agonist. Peak fractions were collected, concentrated and incubated with PNGase F (NEB), PreScission protease (GenScript) to remove the potential glycosylation and N-terminal His–BRIL, respectively, and 100 μg scFv16 at 4 °C overnight. The next day, cleaved His–BRIL and protein, uncleaved protein and proteases were separated by the same procedure as described above. Peak fractions were concentrated to 3–5 mg ml$^{-1}$ for electron microscopy analysis. Four KOR–G-protein–scFv16 complexes were purified according to the same procedure except that different agonists were used.

### Expression and purification of scFv16 protein

The scFv16 protein was expressed by infection of Sf9 cells at a cell density of $2.5 \times 10^6$ cells per ml in ESF921 medium (Expression System) with the P1 baculovirus at an MOI of 2. After 96 h, the cell culture medium containing secreted scFv16 protein was collected by centrifugation at 4,000 rpm for 15 min. The pH of the supernatant was adjusted to 7.5 by addition of Tris-base power. Chelating agents were quenched by the addition of 1 mM nickel chloride and 5 mM calcium chloride and incubation with stirring for 1 h at room temperature and 5 h in the cold room. We removed the precipitates by centrifugation and the resultant supernatant was further cleaned with 0.45 μm filter paper, and incubated with 2 ml Ni-NTA resin and 10 mM imidazole overnight in the cold room. The Ni-NTA resin was washed the next day with 20 ml buffer (20 mM HEPES pH 7.5, 100 mM NaCl, 0.00075% (w/v) LMNG, 0.000075% (w/v) CHS, 0.00025% (w/v) GDN, 20 mM imidazole). The protein was eluted with the same buffer supplemented with 300 mM imidazole, concentrated and further purified on the Superdex 200 increase 10/300 column. Monomeric fractions were pooled, concentrated, flash-frozen in liquid nitrogen and stored at −80 °C until future use.

### Expression and purification of heterotrimeric G proteins

The expression of heterotrimeric G protein was achieved by infection of Sf9 cells at a cell density of $2.5 \times 10^6$ cells per ml in ESF921 medium (Expression System) with the P1 baculovirus at an MOI of 2. After 48 h, cells were collected and lysed in buffer containing 200 mM NaCl, 40 mM HEPES pH 7.5, 0.2% Triton X-100, 5% glycerol, 3 mM β-me and protease inhibitors. The supernatant was isolated by centrifugation at 40,000 rpm for 50 min and incubated with 1 ml Ni-NTA resin and 20 mM imidazole overnight at 4 °C. The resin was collected the next day and washed with 20 ml buffer containing 100 mM NaCl, 20 mM HEPES pH 7.5, 5% glycerol, 20 mM imidazole and 3 mM β-me. The protein was then eluted with elution buffer (300 mM NaCl, 20 mM HEPES pH 7.5, 5% glycerol, 3 mM β-me and 300 mM imidazole), concentrated and further purified on the Superdex 200 increase 10/300 column, which was pre-equilibrated with buffer the same as the elution buffer except without the imidazole. The peak fractions were concentrated, flash-frozen in liquid nitrogen and stored at −80 °C for future binding assays.

### Cryo-EM data collection and 3D reconstruction

The purified samples (3–4 μl) were applied to glow-discharged 300-mesh Au grids (Quantifoil R1.2/1.3) individually and vitrified using a Vitrobot mark IV (Thermo Fisher Scientific). Cryo-EM imaging was performed on the Talos Artica system operated at 200 kV at a nominal magnification of ×45,000 using a Gatan K3 direct electron detector at a physical pixel size of 0.88 Å. Each stack video was recorded for 2 to 2.7 s in 60 frames at a dose rate of about 15 e$^-$ px$^{-1}$ s$^{-1}$, leading to a total exposure dose indicated in Extended Data Table 1. Videos were collected automatically with SerialEM[59] using an optimized multishot array procedure[60].

Dose-fractioned image stacks were processed for beam-induced motion correction followed by contrast transfer function estimation.

Particles were selected using Blob particle picker, extracted from the micrograph and then used for 2D classification and 3D classification followed by non-uniform refinement. All of these steps were performed in cryoSPARC[61,62].

### Model building and refinement

Maps from cryoSPARC were used for map building, refinement and subsequent structural interpretation. The dominant-negative $G_{i1}$ trimer model and scFv16 model were adapted from the cryo-EM structure of the MRGPRX2–$G_{i1}$ complex (Protein Data Bank (PDB): 7S8M)[63]. $G_{oA}$, $G_z$ and $G_g$ trimer models were built from the $G_{i1}$ trimer model, followed by mutating the non-conserved residues back to the wild-type $G_{oA}$, $G_z$ and $G_g$. The receptor KOR model was taken from the active-state KOR–Nb39 structure (PDB: 6B73)[13]. The receptor, G proteins and scFv16 were docked into the cryo-EM map using Chimera[64]. The complex models (KOR–G-protein–scFv16) were manually built in Coot[65], followed by several rounds of real-space refinement using Phenix[66]. The model statistics were validated using Molprobity[67]. Structural figures were prepared using Chimera or PyMol (https://pymol.org/2/).

### cAMP inhibition assay

For the KOR–$G_{\alpha i}$-mediated cAMP inhibition assay, HEK293T (ATCC CRL-11268) cells were co-transfected with human KOR or various mutants along with a split-luciferase-based cAMP biosensor (GloSensor, Promega) at a 1:1 ratio (KOR:GloSensor). After 16 h, the transfected cells were plated into poly-L-lysine-coated 96-well white clear-bottom cell culture plates with DMEM + 1% dialysed FBS at a density of 40,000–50,000 cells per 200 µl per well and incubated at 37 °C with 5% $CO_2$ overnight. The next day, 3× drug solutions were prepared in fresh drug buffer (20 mM HEPES, 1× HBSS, 0.3% bovine serum albumin (BSA), pH 7.4). The plates were decanted the next day and received 40 µl per well of drug buffer (20 mM HEPES, 1× HBSS, pH 7.4) followed by addition of 20 µl of 3× drug solutions for 15 min in the dark at room temperature. Cells then received 20 µl luciferin (4 mM final concentration) supplemented with isoproterenol (300 nM final concentration), stimulating the production of endogenous cAMP through $\beta_2$ adrenergic $G_s$ activation, and incubated in the dark at room temperature. After 15 min, luminescence intensity was quantified using the Mithras LB 940 multimode microplate reader (Berthold Technologies). Data were plotted as a function of drug concentration, normalized to percentage U50,488 stimulation, and analysed using log (agonist) versus response in GraphPad Prism (v.9.3.1).

### BRET2 assay

To measure the agonist-stimulated G-protein (wild type and mutants) activation by KOR and various mutants, a BRET2-based cell assay was used. Specifically, four plasmids (KOR, $G\alpha$, $G\beta$, $G\gamma$) were used, in which each $G\alpha$ is tagged with a luciferase (Rluc8) and $G\gamma$ is tagged with an N-terminal GFP. Specifically, the $G\alpha_{i1}/G\beta_3/G\gamma_9$, $G\alpha_{oA}/G\beta_3/G\gamma_8$, $G\alpha_z/G\beta_3/G\gamma_1$ and $G\alpha_g/G\beta_3/G\gamma_1$ combinations were used for BRET2 $G_{i1}$, $G_{oA}$, $G_z$ and $G_g$ experiments, respectively. Detailed information of the GFP-$G\gamma$ and $G\alpha$-Rluc8 constructs was described previously[30]. HEK293T cells were then transfected with the four plasmids (KOR, $G\alpha$-Rluc8, $G\beta$, $G\gamma$–GFP) using a 1:5:5:5 DNA ratio of receptor:$G\alpha$-RLuc8:$G\beta$:$G\gamma$-GFP2 (100 ng receptor, 500 ng $G\alpha$–RLuc8, $G\beta$ and $G\gamma$–GFP2 for 10 cm dishes). Transit 2020 (Mirus Biosciences) was used to complex the DNA at a ratio of 2 µl Transit per µg DNA in Opti-MEM (Gibco-Thermo Fisher Scientific). Then, 16 h after transfection, cells were plated in poly-L-lysine-coated 96-well white clear-bottom plates in plating medium (DMEM + 1% dialysed FBS) at a density of 40,000–50,000 cells in 200 µl per well and incubated overnight. The next day, the plates were decanted and washed once with 60 µl drug buffer (20 mM HEPES, 1× HBSS, pH 7.4) and then 60 µl drug buffer containing coelenterazine 400a (Nanolight Technology) at a final concentration of 5 µM was added to each well. After 5 min for substrate diffusion, 30 µl 3× drug solutions in fresh drug buffer

(20 mM HEPES, 1× HBSS, 0.3% BSA, pH 7.4) was added to each well and incubated for an additional 5 min. Finally, the plates were read on the Mithras LB 940 multimode microplate reader (Berthold Technologies) with 400 nm (RLuc8-coelenterazine 400a) and 510 nm (GFP2) emission filters for 1 s per well. The GFP to Rluc8 ratio was calculated, plotted as a function of drug concentration, normalized to percentage U50,488 stimulation and analysed using log (agonist) vs response in GraphPad Prism (v.9.3.1).

### Radioligand-binding assay

Saturation binding assays were performed using the construct BRIL-wt-KOR$_{(54–368)}$ reconstituted into nanodiscs comprised of KOR, spMSP1D1 and lipid mixture (POPC:POPE:POPG = 3:1:1) at a molar ratio of 1:3:100. Binding assays were set-up in 96-well plates in standard binding buffer (50 mM Tris-HCl, 10 mM MgCl$_2$, 0.1 mM EDTA, 0.1% BSA, pH 7.4) at room temperature. Saturation binding assays with 0.1–20 nM $^3$H-U69,593 in the standard binding buffer were performed to determine the equilibrium dissociation constant ($K_d$) and $B_{max}$. To determine the effects of G proteins on $^3$H-U69,593 binding, each G protein (final concentration 1 µM) was incubated with $^3$H-U69,593 and homogenous membrane fractions for 3.5 h at room temperature. Data were analysed using GraphPad Prism (v.9.3.1) using a one-site model.

For the competitive binding assay, $^3$H-JDTic (0.68 nM), homogenous membrane fractions expressing KOR and 3× GR89,696 solutions were incubated in 96-well plates in standard binding buffer in the absence or presence of four G proteins in various concentrations (final concentration: 1,900 nM, 190 nM, 19 nM, 1.9 nM, 0 nM) for 3.5 h at room temperature in the dark, and then terminated by rapid vacuum filtration onto chilled 0.3% PEI-soaked GF/A filters followed by three quick washes with cold wash buffer (50 mM Tris-HCl, pH 7.4) and read. Results (with or without normalization) were analysed using GraphPad Prism (v.9.3.1) using one-site or allosteric IC$_{50}$ shift models.

### Cell-surface expression studies

The cell-surface expression levels of wild-type KOR and its mutants were measured using an enzyme-linked immunosorbent assay (ELISA). In brief, HEK293T (ATCC CRL-11268) cells were transiently transfected with wild-type KOR and KOR mutant DNA at the same quantity. After 24 h, cells were plated in poly-L-lysine-coated 96-well white clear-bottom plates in plating medium (DMEM + 1% dialysed FBS) at a density of 40,000–50,000 cells in 200 µl per well and incubated overnight. The next day, plates were decanted and fixed with 4% (w/v) paraformaldehyde for 10 min at room temperature. Cells were then washed twice with 1× phosphate-buffered saline (PBS) (pH 7.4) and blocked by 1× PBS containing 0.5% (w/v) non-fat milk for at least 30 min at room temperature followed by incubation with anti-Flag (M2)–horseradish peroxidase-conjugated antibodies (Sigma-Aldrich, A8592) diluted 1:20,000 in the same buffer for 1 h at room temperature. After washing three times with 1× PBS, 1-Step Ultra-TMB ELISA substrate (Thermo Fisher Scientific, 34028) was added to the plates and the plates were incubated at 37 °C for 15–30 min and terminated by addition of 1 M sulfuric acid ($H_2SO_4$) stop solution. Finally, the plates were read at a wavelength of 450 nm using the BioTek Luminescence reader. The data were analysed using GraphPad Prism (v.9.3.1).

### G-protein expression studies

To measure the expression levels of four wild-type G proteins and their mutants, HEK293T (ATCC CRL-11268) cells were transiently transfected with the same quantity of wild-type and mutant G proteins DNA. After 16 h, cells were plated in poly-L-lysine-coated 96-well white clear-bottom plates in plating medium (DMEM + 1% dialysed FBS) at a density of 40,000–50,000 cells in 200 µl per well and incubated overnight. The next day, the plates were decanted and washed once with 60 µl drug buffer (20 mM HEPES, 1× HBSS, pH 7.4), then 60 µl drug buffer containing coelenterazine 400a (Nanolight Technology) at a final

concentration of 5 μM was added to each well. After 5 min for substrate diffusion, plates were read in a Mithras LB 940 multimode microplate reader (Berthold Technologies) with 400 nm (RLuc8-coelenterazine 400a) and 510 nm (GFP2) emission filters for 1 s per well. The Rluc8 values represented the G-protein expression levels and were plotted in the GraphPad Prism (v.9.3.1).

## GTP turnover assay
Analysis of GTPase activity of G proteins ($G_{i1}$, $G_{oA}$, $G_z$, $G_g$) was performed by using a modified protocol of the GTPase-Glo assay (Promega). G proteins were serially (1:1) diluted into various concentrations with a buffer of 300 mM NaCl, 20 mM HEPES pH 7.5 and 1 mM DTT, and 5 μl was dispensed into each well of a 384-well plate. The reaction was initiated by adding 5 μl 1 μM GTP solution to 5 μl G proteins. After incubation for 90 min at room temperature, 10 μl reconstituted GTPase-Glo reagent was added to the sample and incubated for 30 min at room temperature. Luminescence was measured after addition of 20 μl detection reagent and incubation for 10 min at room temperature using the Mithras LB 940 multimode microplate reader (Berthold Technologies). The data were analysed using GraphPad Prism (v.9.3.1).

## Molecular dynamics simulations
The Gromacs simulation engine (v.2020.3)[68] was used to run all molecular dynamics simulations under the Charmm36 force-field topologies and parameters[69,70]. Charmm force-field parameters and topologies for the ligands momSalB and GR89,696 were generated using Charmm-GUI's Ligand Reader & Modeller tool[70]. The loop grafting and optimization for modelling missing side chains and loops was performed in the ICM-Pro (v.3.9-2b) molecular modelling and drug discovery suite (Molsoft)[71]. The structurally conserved helix-8 (Hx8) amphipathic helical motifs in KOR were modelled using human antagonist-bound KOR (PDB: 4DJH)[26] as the template structure. The lobe in $G_{i1}$, $G_{oA}$, $G_z$ and $G_g$ proteins was modelled using a human agonist-bound CB2–$G_i$ structure (PDB: 6PT0)[72]. Structure regularization and torsion profile scanning were performed using ICMFF force field[73]. The GR89,696-bound structures of KOR complexes with $G_z$ and $G_g$ proteins as well as momSalB-bound KOR with $G_{i1}$ and $G_{oA}$ proteins were then uploaded to the Charmm-GUI webserver[69], where the starting membrane coordinates were determined by the PPM[74] server using the Charmm-GUI interface. The complexes were then embedded in a lipid bilayer composed of 1,2-dipalmitoyl-sn-glycero-3-phosphatidylcholine (DPPC), 1,2-dioleoyl-sn-glycero-3-phosphatidylcholine (DOPC) and cholesterol (CHL1) following the recommended ratio of 0.55:0.15:0.30, respectively[75]. The GR89,696-bound KOR complex with $G_z$ contained 330 DPPC, 90 DOPC and 180 CHL1 lipids, 64,400 water molecules, and 178 sodium and 176 chloride ions. The GR89,696-bound KOR complex with $G_g$ contained 330 DPPC, 90 DOPC and 180 CHL1 lipids, 64,227 water molecules, and 184 sodium and 175 chloride ions. The momSalB-bound KOR complex with $G_{i1}$ contained 220 DPPC, 60 DOPC and 120 CHL1 lipids, 43,172 water molecules, 124 sodium and 116 chloride ions. The momSalB-bound KOR complex with $G_{oA}$ contained 220 DPPC, 60 DOPC and 120 CHL1 lipids, 41,663 water molecules, and 126 sodium and 113 chloride ions. All of the systems were first processed for 50,000 steps of initial energy minimizations, then 60 ns of equilibration, followed by production runs of up to 750 ns for the KOR–$G_g$ based system and 550 ns for the rest ($G_{i1}$, $G_{oA}$ and $G_z$-bound KOR systems). The simulations were carried out on GPU clusters at the University of Southern California's High-Performance Computing Center. The temperature of 310 K and *v*-rescale thermostat algorithm were used during the production run[76]. The analyses of molecular dynamics trajectories were performed using the GROMACS software package[68].

## Data statistical analysis
For BRET2 and cAMP-inhibition assays, in the case of more than two groups, log-transformed $EC_{50}$ values were first analysed using one-way ANOVA. If significant, the Dunnett's multiple-comparison test was used to compare each mutant with the wild-type one, and the Tukey's multiple-comparison test was used to compare log-transformed $EC_{50}$ values between each group. In the case of two groups, log-transformed $EC_{50}$ values were analysed using unpaired two-tailed Student's *t*-tests to compare each mutant with a wild-type receptor. For the cell-surface expression studies, the optical density at 450 nm values of each mutant were normalized to the wild-type KOR receptor (normalized as 100%), and the resultant values were then first analysed using one-way ANOVA. If significant, a Dunnett's multiple-comparison test was used to compare each mutant with the wild-type receptor. For G-protein expression studies, the Rluc values of each mutant were normalized to the wild-type G protein (normalized as 100%), and the resultant values were then first analysed using one-way ANOVA. If significant, a Dunnett's multiple-comparison test was used to compare each mutant with the wild-type G protein. For radioligand binding and GTP turnover assays, data were analysed using unpaired two-tailed Student's *t*-tests. In one-way ANOVA and unpaired two-tailed Student's *t*-test analysis, the significance threshold was set at $\alpha = 0.05$. Asterisks denote statistical significance; *$P < 0.05$, **$P < 0.01$, ***$P < 0.001$; ****$P < 0.0001$; NS represents not significant.

## Reporting summary
Further information on research design is available in the Nature Portfolio Reporting Summary linked to this article.

## Data availability
The coordinate and cryo-EM map of KOR–$G_{i1}$–momSalB, KOR–$G_{oA}$–momSalB, KOR–$G_z$–GR89,696 and KOR–$G_g$–GR89,696 have been deposited at the PDB and Electron Microscopy Data Bank under accession codes 8DZP (EMD-27804), 8DZQ (EMD-27805), 8DZS (EMD-27807) and 8DZR (EMD-27806), respectively. All data supporting the findings of this study are available within the Article and its Supplementary Information.

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

**Acknowledgements** We thank J. Peck and J. Strauss for technical assistance in this project; the staff at the Washington University Center for Cellular Imaging for sample screening; the staff of the NIDA Drug Supply Program for providing the $^3$H-JDTic; X.-P. Huang and the members of the Psychoactive Drug Screening Program (PDSP) at UNC-Chapel Hill for the GPCRome screening analysis; and staff at the Center for Advanced Research Computing (CARC) at the University of Southern California for providing computing resources that have contributed to the research results reported within this publication. The work is supported by NIH grants R35GM143061 (to T.C.) and R01NS099341 (to P.Y.). The Titan X Pascal used for this research was donated to J.F.F. by NVIDIA.

**Author contributions** J.H. prepared the four KOR–G protein complexes, performed the functional validation assays and prepared the manuscript. J.Z. helped with the cryo-EM data processing. A.L.N. and J.H.L. performed the molecular dynamics simulations. S.M.B. helped with construct optimization, sample preparation and saturation binding experiments. B.E.K. provided the expression plasmid for $G_g$. L.Z. helped with protein expression. V.A.R. and S.M. helped to collect the NMR spectrum of momSalB. D.E.N. synthesized the compound momSalB. V.K. supervised the molecular dynamics simulations and manuscript discussion. P.Y. helped with the map discussion. J.F.F. prepared grids, collected the cryo-EM data and resolved the cryo-EM density map. P.Y., J.F.F. and T.C. prepared the manuscript and supervised the project.

**Competing interests** The authors declare no competing interests.

**Additional information**
**Correspondence and requests for materials** should be addressed to Jonathan F. Fay or Tao Che.

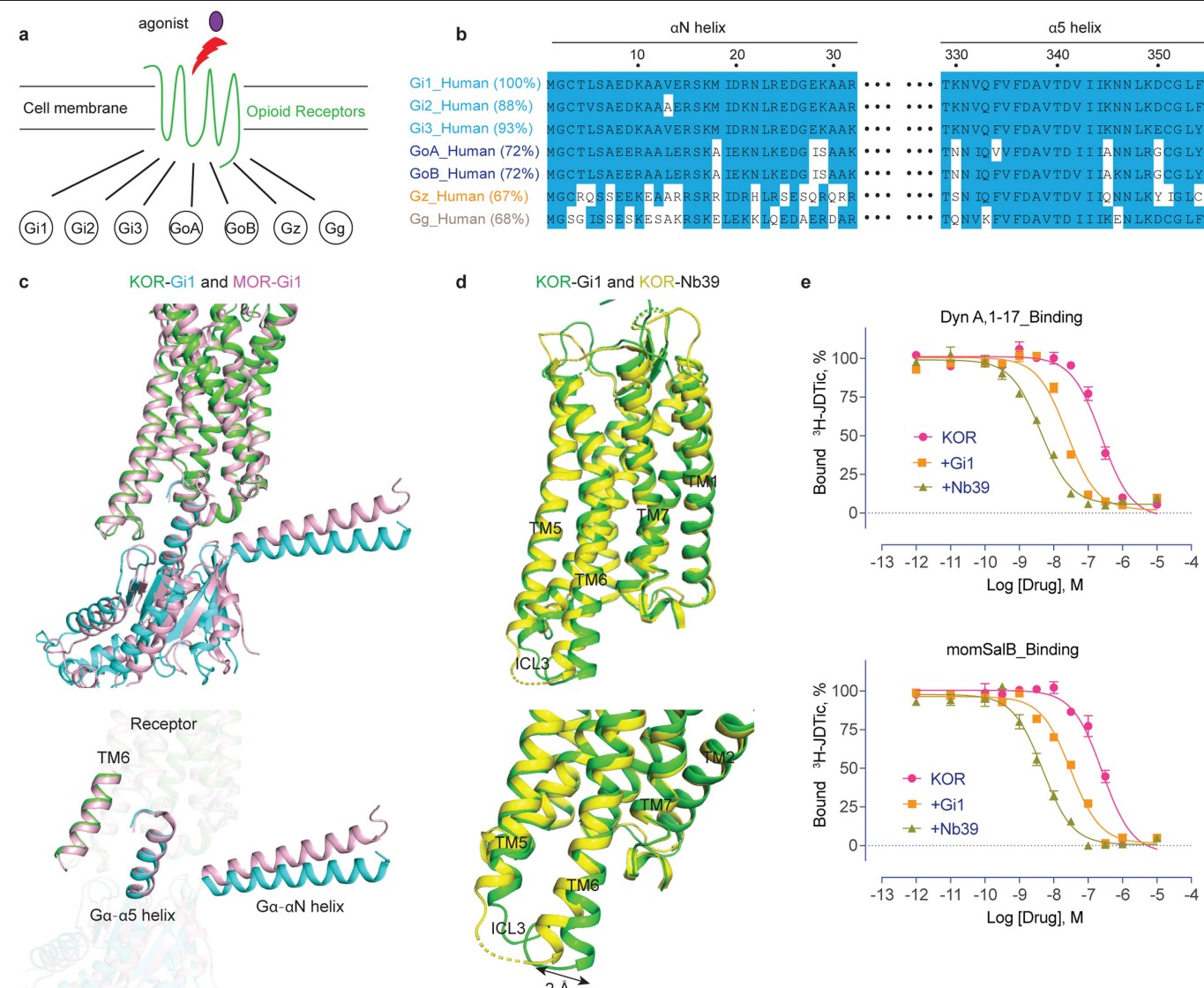

**Extended Data Fig. 1 | Comparison of KOR-Gi1 complex structure with MOR-Gi1 and active-state KOR-Nb39 structures. a**. Opioid receptors primarily couple to Gi/o family subtypes upon activation. **b**. Sequence alignment (αN and α5 helices) of Gαi/o family subtypes. The percentage represents the sequence similarity related to Gi1 (set as 100%). **c**. Overall alignment of KOR-Gi1 and MOR-Gi1 structures. The two structures are globally similar to each other, including TM6 of the receptor and α5 helix of the Gα subunit, but different in

the αN helix of Gα. **d**. The Gi1-bound KOR differs from the Nb39-bound KOR in the degrees of TM6 outward movement (by 2 Å). **e**. Both Gi1 and Nb39 act as positive allosteric modulators of KOR, but differentially increase the binding affinity of KOR agonists. Data are grouped data ± s.e.m. from n = 3 biological replicates. Full quantitative parameters from this experiment are listed in Supplementary Table 8.

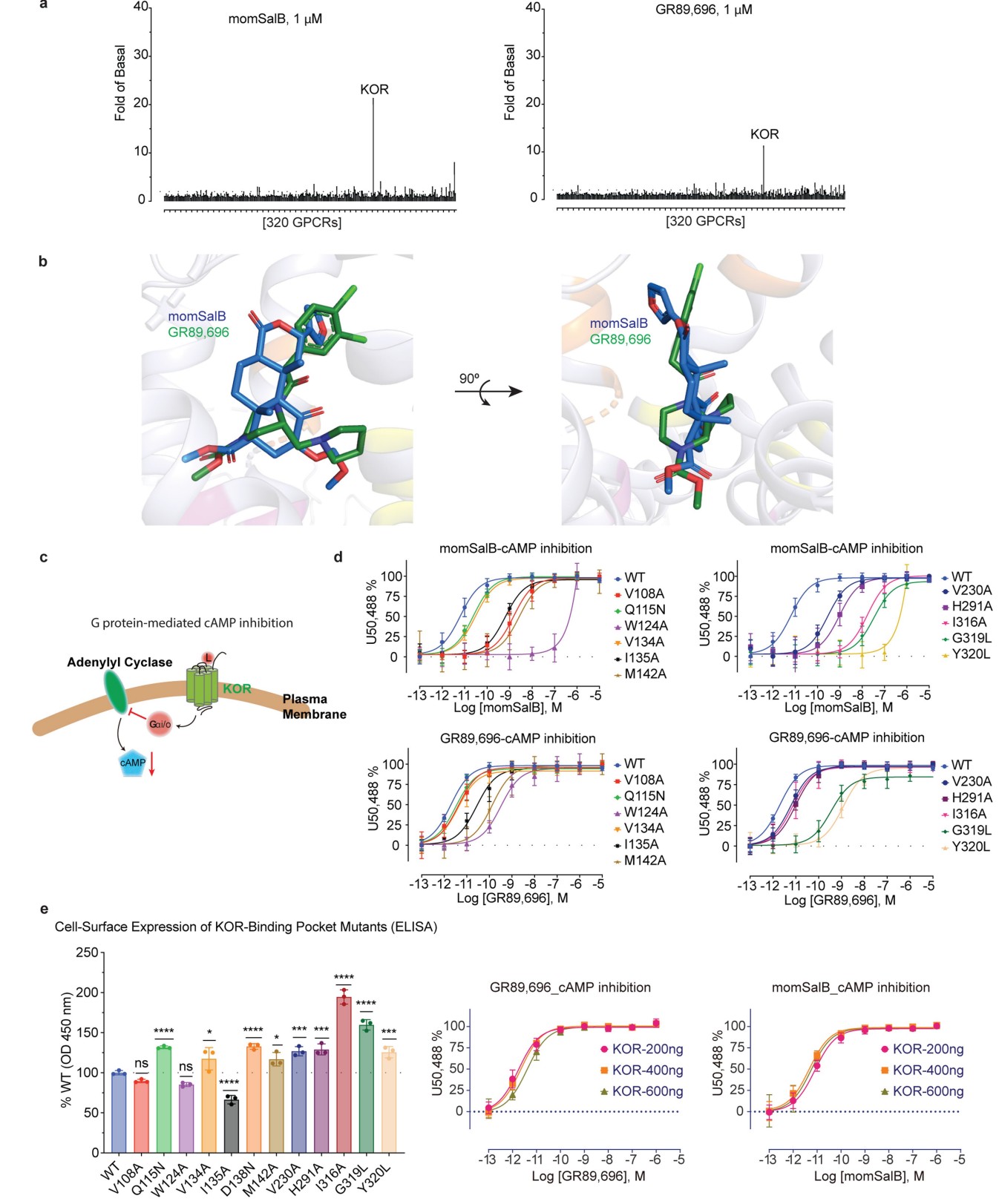

**Extended Data Fig. 2** | See next page for caption.

**Extended Data Fig. 2 | The binding pharmacology of momSalB and GR89,696. a**. GPCRome screening at 320 GPCRs that measures agonist activity of tested ligands shows that momSalB and GR89,696 are selective at KOR. **b**. The binding poses of momSalB and GR89,696 show that they adopt different planes in the orthosteric pocket. **c**. A cartoon model of G protein-mediated cAMP reporter assays. The Gαi/o here represents all subtypes expressed in the cells. **d**. Mutagenesis screening of key binding-pocket residues using G protein-mediated cAMP reporter assay. Data are grouped data ± s.e.m. of n = 3 biological replicates. Full quantitative parameters from this experiment are listed in Supplementary Table 2. **e**. Measurement of cell surface expression of KOR binding pocket mutants by ELISA. In general, these mutants maintained robust cell surface expression. Although some mutations significantly altered surface expression compared to the wild type, the increased or decreased expression appeared to minimally affect the potency in agonist-mediated cAMP inhibition. Bar-graphs are $OD_{450}$ ± s.e.m. from n = 3 biological replicates. Statistical significance for each mutant is compared in a one-way analysis of variance (ANOVA) with Dunnett's multiple comparisons test to the wild type (* = $p < 0.05$, ** = $p < 0.01$, *** = $p < 0.001$, **** = $p < 0.0001$, "ns" represents no significance; V108A: $p = 0.3919$, W124A: $p = 0.0893$, V134A: $p = 0.0308$, M142A: $p = 0.0398$, V230A: $p = 0.0004$, H291A: $p = 0.0002$, Y320L: $p = 0.0008$). Signalling curves are grouped data ± s.e.m. of n = 3 biological replicates. Full quantitative parameters from this experiment are listed in Supplementary Table 9.

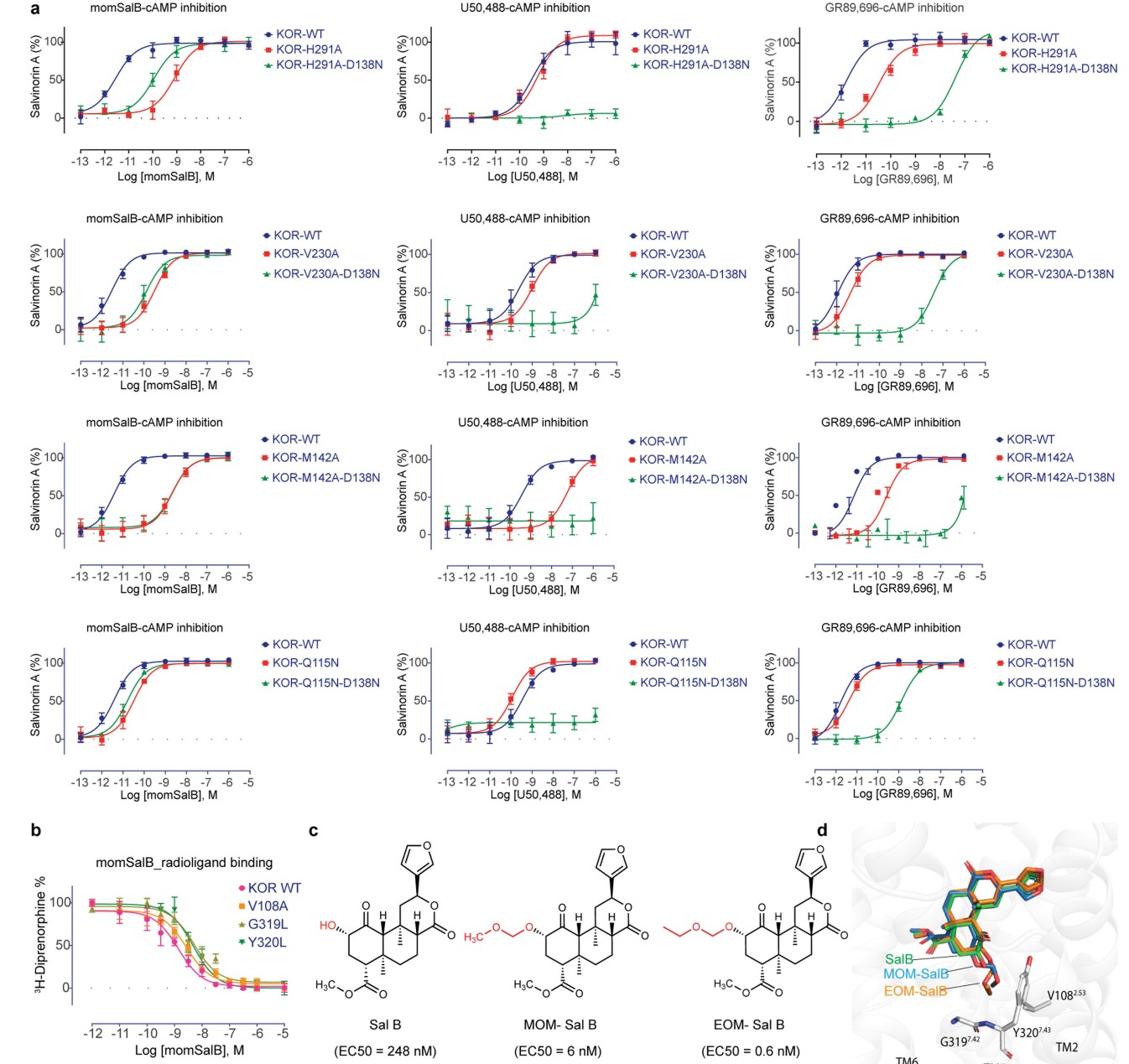

**Extended Data Fig. 3 | Molecular determinants of momSalB agonism. a**. The positive effect of additional D138[3.32]N mutation on the momSalB-mediated cAMP inhibition through KOR. The additional D138[3.32]N mutation does not rescue U50,488 or GR89,696-mediated cAMP inhibition. Data are grouped data ± s.e.m. from n = 3 biological replicates. Full quantitative parameters from this experiment are listed in Supplementary Table 10. **b**. Effects of mutations in the hydrophobic pocket on the binding affinity of momSalB. Data are grouped data ± s.e.m. from n = 3 independent biological replicates. Full quantitative

parameters from this experiment are listed in Supplementary Table 11. **c** and **d**. Chemical structures of SalB, momSalB, and EOM-SalB. Differences are highlighted by red colour. The agonist activity of each analogue is shown in the parentheses. Data for EOM-SalB was taken from ref. 19. Binding poses of SalB and EOM-SalB at KOR were revealed by molecular docking performed in the Schrodinger Maestro v12.9. The three ligands occupy a similar binding pocket with different extents toward the hydrophobic pocket.

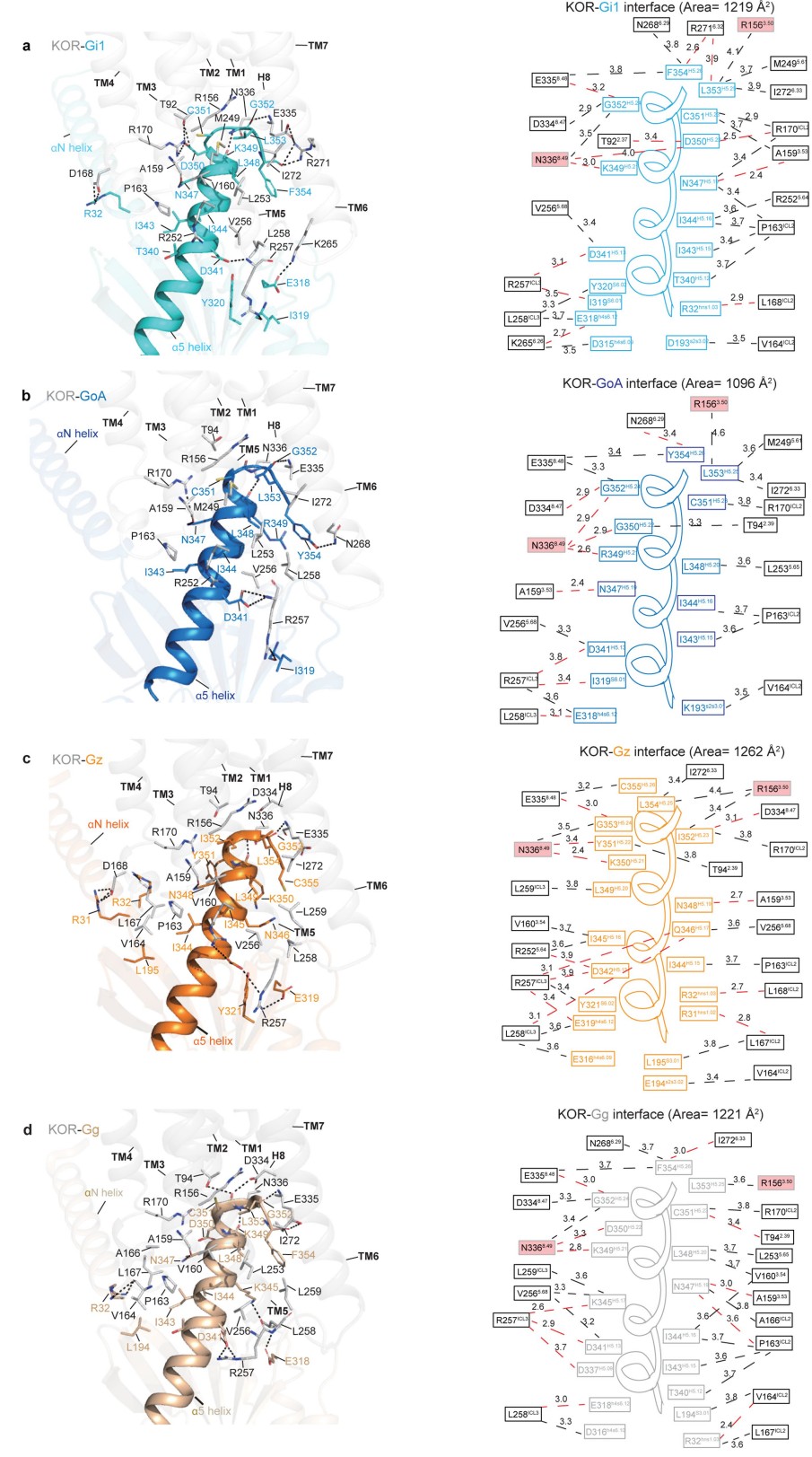

**Extended Data Fig. 4 | Comparison of interface details and areas between KOR and each G protein subtypes.** Specific interactions between the receptor and individual Gα subunits, Gi1 (**a**), GoA (**b**), Gz (**c**) and Gg (**d**). (Cartoon) Key residues from the intracellular side of KOR, and residues in the αN and α5 helix were mapped out. The H-bond or salt-bridge interactions are shown as red dashed lines. The closest distances between the intracellular KOR residues and the Gα residues were labelled. The distance cutoff is 4 Å. The interface area was calculated by the online server PDBePISA[77].

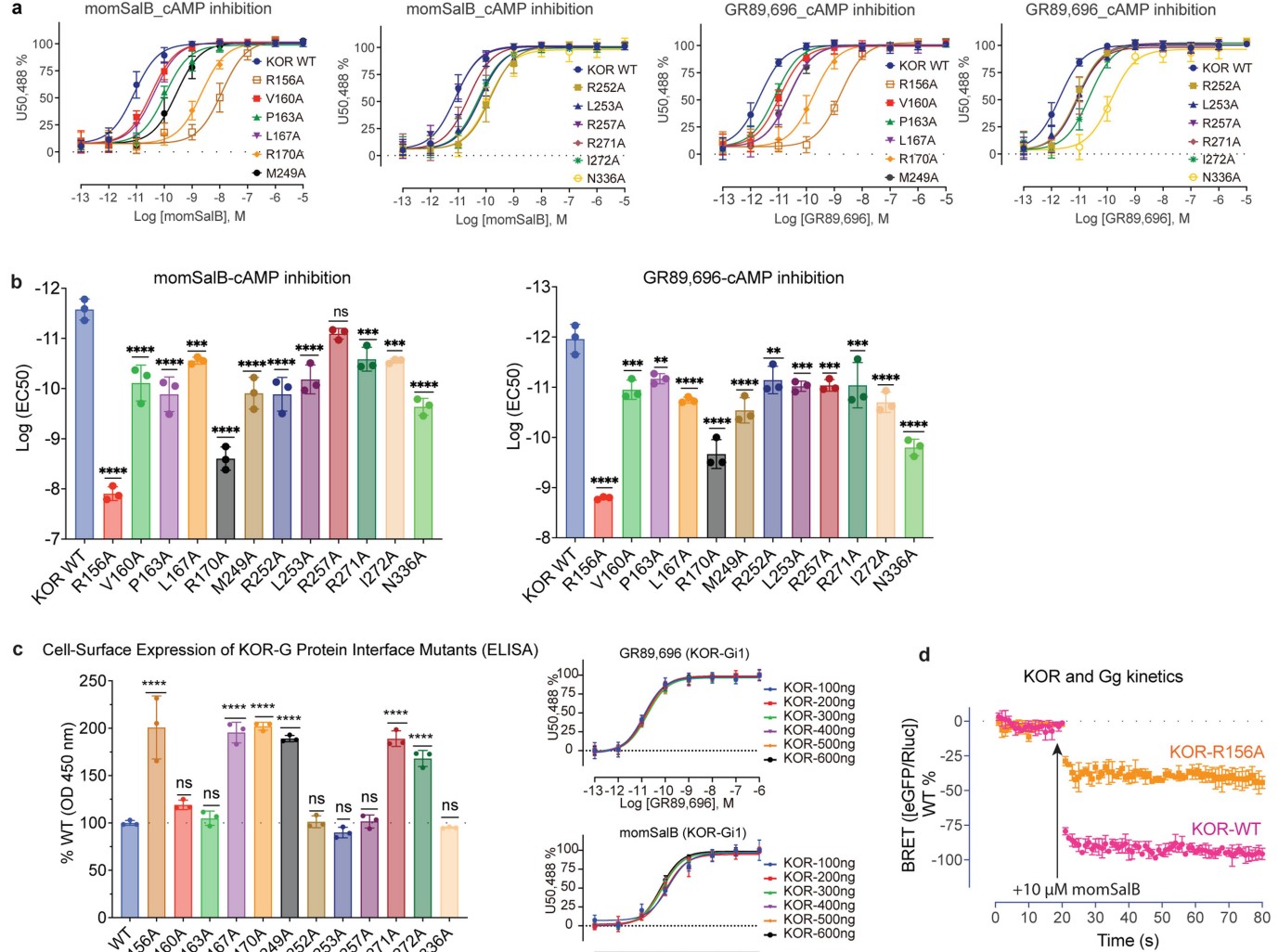

**Extended data Fig. 5 | The effect of KOR-G-protein interface residues on the G protein coupling. a**. Mutagenesis screening of KOR-G protein interface (KOR side) residues by cAMP inhibition assays. Data are grouped data ± s.e.m. from n = 3 biological replicates. **b**. Mutagenesis analysis of intracellular KOR residues by cAMP inhibition assays. Data are mean LogEC50 ± s.e.m. from n = 3 biological replicates. Statistical significance for each mutant is compared in a one-way analysis of variance (ANOVA) with Dunnett's multiple comparisons test to the wild type (* = $p < 0.05$, ** = $p < 0.01$, *** = $p < 0.001$, **** = $p < 0.0001$, "ns" represents no significance, GR89,696: V160A: $p = 0.0001$, P163A: $p = 0.0020$, R252A: $p = 0.0014$, L253A: $p = 0.0003$, R257A: $p = 0.0003$, R271A: $p = 0.0004$; momSalB: L167A: $p = 0.0003$, R257A: $p = 0.1514$, R271A: $p = 0.0004$, I272A: $p = 0.0002$). Full quantitative parameters from this experiment are listed in Supplementary Table 12. **c**. Measurement of cell surface expression of

KOR-G protein interface mutants by ELISA. The BRET results suggest a minor effect with different concentrations of KOR plasmids. Bar-graphs are OD$_{450}$ ± s.e.m. from n = 3 independent biological replicates. Statistical significance for each mutant is compared in a one-way analysis of variance (ANOVA) with the Dunnett's multiple comparisons test to the wild type (* = $p < 0.05$, ** = $p<0.01$, *** = $p < 0.001$, **** = $p < 0.0001$, "ns" represents no significance; V160A: $p = 0.2732$, P163A: $p = 0.9992$, R252A: $p = 0.9998$, L253A: $p = 0.8943$, R257A: $p = 0.9997$, N336A: $p = 0.9992$). Signalling curves are grouped data ± s.e.m. of n = 3 biological replicates. Full quantitative parameters from this experiment are listed in Supplementary Table 13. **d**. The KOR-R156A mediated decrease of efficacy was confirmed by the BRET2-based kinetic measurement. Data are net BRET ± s.e.m. from n = 3 independent biological replicates.

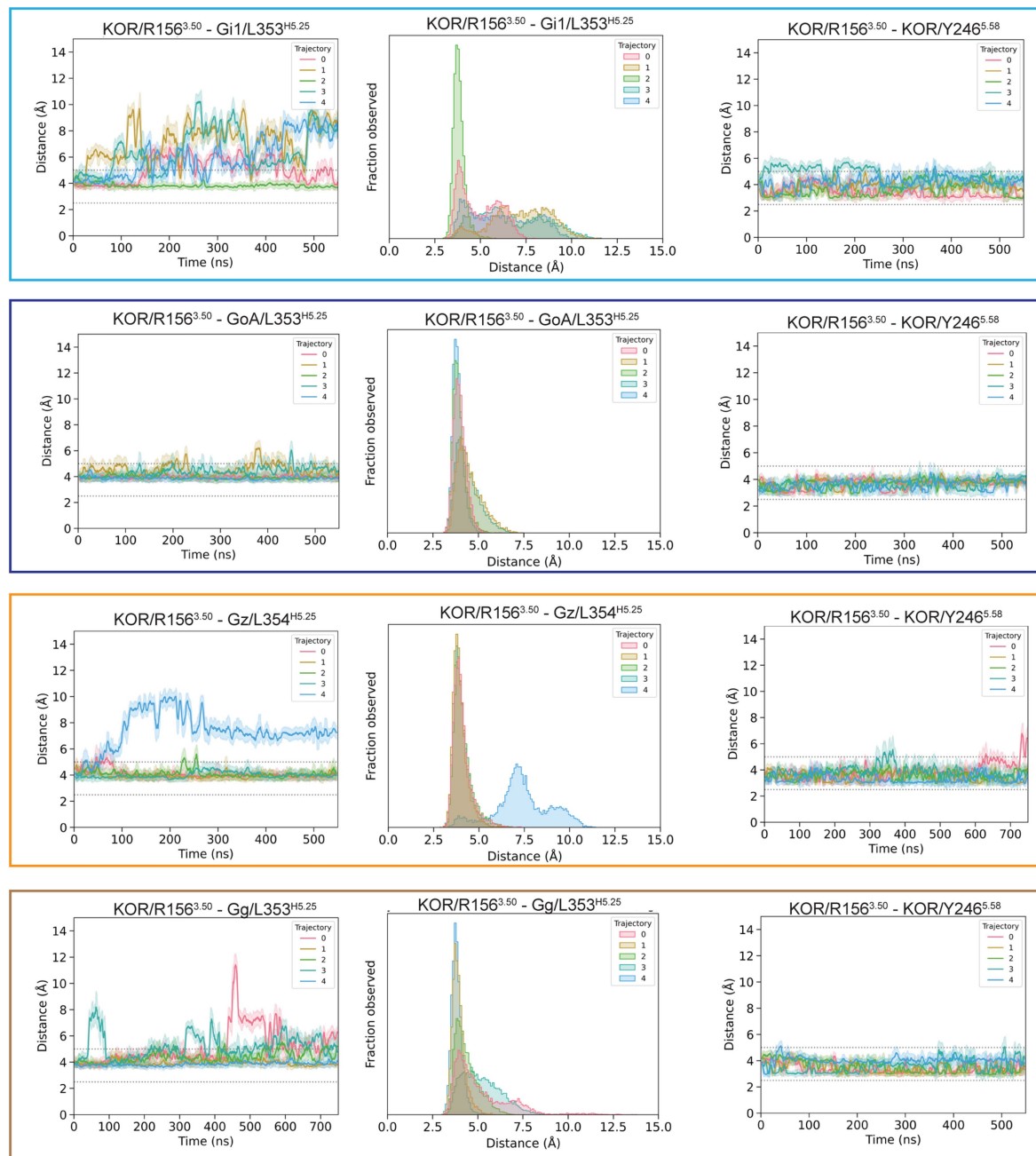

**Extended Data Fig. 6 | Molecular dynamics (MD) simulations reveal distance traces of interactions between the KOR residues (R156[3.50]) with Gα protein residues.** MD simulations revealed the distance traces and frequency between KOR-R156[3.50] and L353/354[H5.25] in the α5 of each Gα subunit. The position of KOR-R156[3.50] is also supported by the stable interaction with KOR-Y246[5.58].

Five independent simulations of KOR-G complex (coral, orange, green, cyan, blue) are shown, spanning 0.55-0.75 μs of cumulative time per system, with the sampling rate of 10 frames per ns, solid lines and same-colour shadows representing moving average values and one standard deviation respectively from 50 frames in all cases.

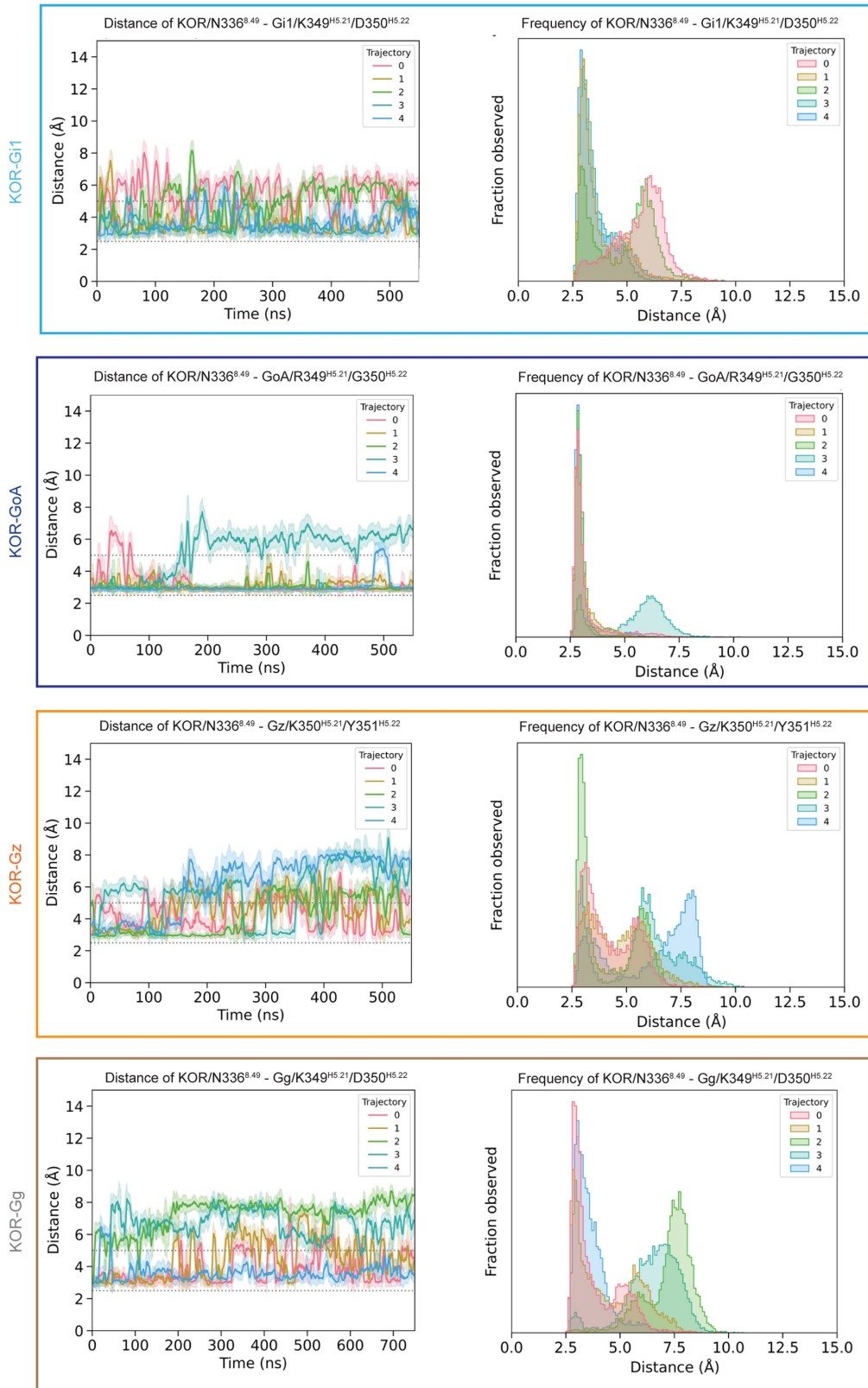

**Extended Data Fig. 7 | Molecular dynamics (MD) simulations reveal distance traces of interactions between the KOR residues (N336[8.49]) with Gα protein residues.** The closest distances of polar residues N336[8.49] in KOR with residues in Gi1, GoA, Gz, or Gg are shown. Distance histograms for each plot are also shown in parallel. Five independent simulations of KOR-G complex (coral, orange, green, cyan, blue) are shown, spanning 0.55 μs (KOR-Gi1, GoA, Gz) and 0.75 μs (KOR-Gg) of cumulative time per system, with the sampling rate of 10 frames per ns, solid lines and same-colour shadows representing moving average values and one standard deviation respectively from 50 frames in all cases.

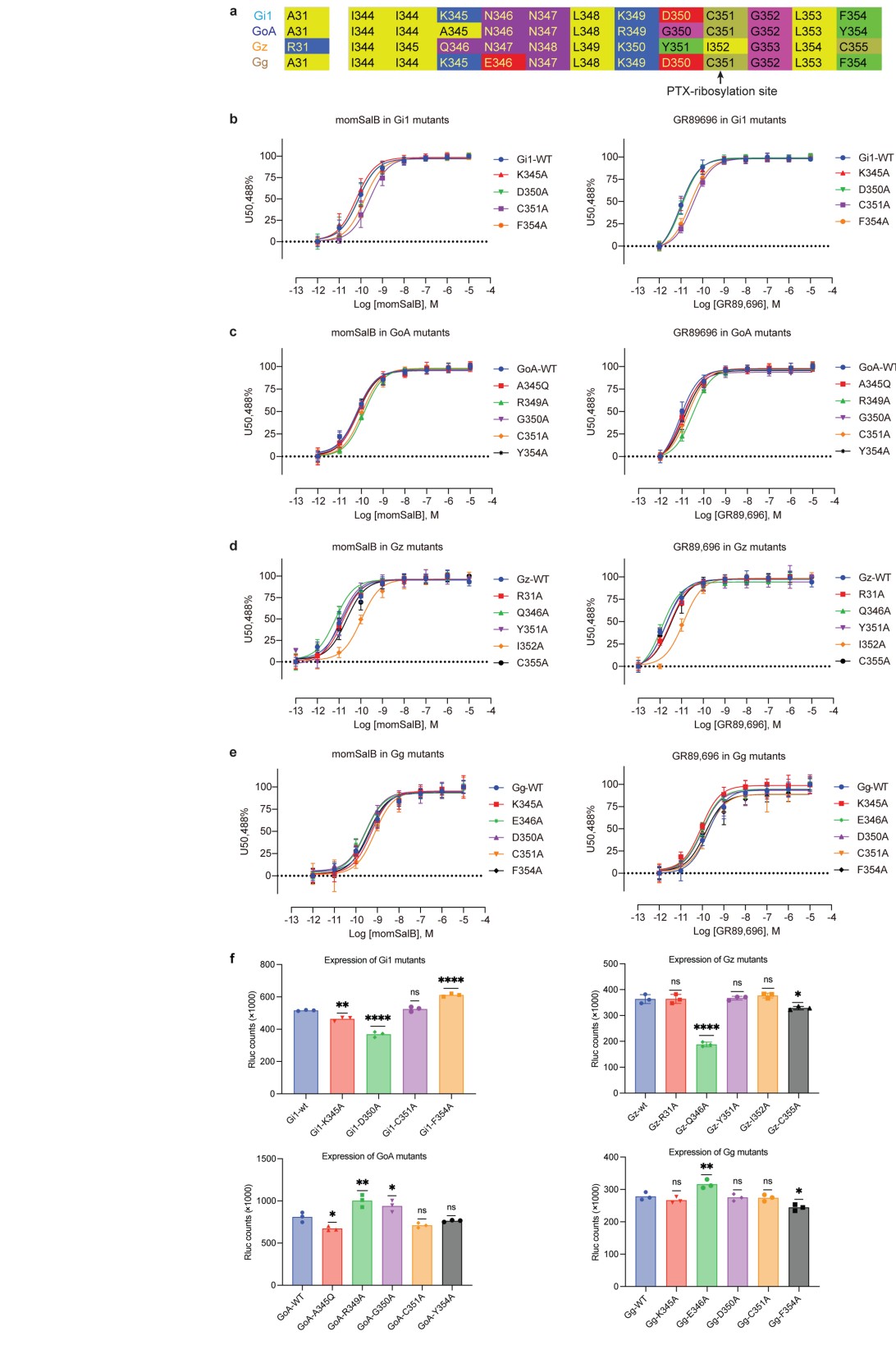

**Extended Data Fig. 8** | See next page for caption.

**Extended Data Fig. 8 | The role of G protein interface residues on KOR-G protein coupling. a**. The nonconserved residues in the G protein α5 and αN helices. **b-e**. The effects of individual mutations in each G protein subtype were screened using the BRET2 KOR-G protein assays. Data are grouped data ± s.e.m. from n = 3 biological replicates. Full quantitative parameters from this experiment are listed in Supplementary Table 14. **f**. The expression levels of G protein mutants were quantified based on the luminescence counts from the Gα-Rluc excitation. Data are BRET ratio ± s.e.m. from n = 3 biological replicates. Statistical significance for each mutant is compared in a one-way analysis of variance (ANOVA) with Dunnett's multiple comparisons test to the wild type ($* = p < 0.05$, $** = p < 0.01$, $*** = p < 0.001$, $**** = p < 0.0001$, "ns" represents no significance; Gi1-K345A: $p = 0.0015$, Gi1-C351A: $p = 0.7984$; GoA-A345Q: $p = 0.0210$, GoA-R349A: $p = 0.0019$, GoA-G350A: $p = 0.0284$, GoA-C351A: $p = 0.1102$, GoA-Y354A: $p = 0.6955$; Gz-R31A: $p > 0.9999$, Gz-Y351A: $p = 0.9947$, Gz-I352A: $p = 0.5286$, Gz-C355A: $p = 0.0161$; Gg-K345A: $p = 0.5849$, Gg-E346A: $p = 0.0050$, Gg-D350A: $p = 0.9936$, Gg-C351A: $p = 0.9808$, Gg-F354A: $p = 0.0091$).

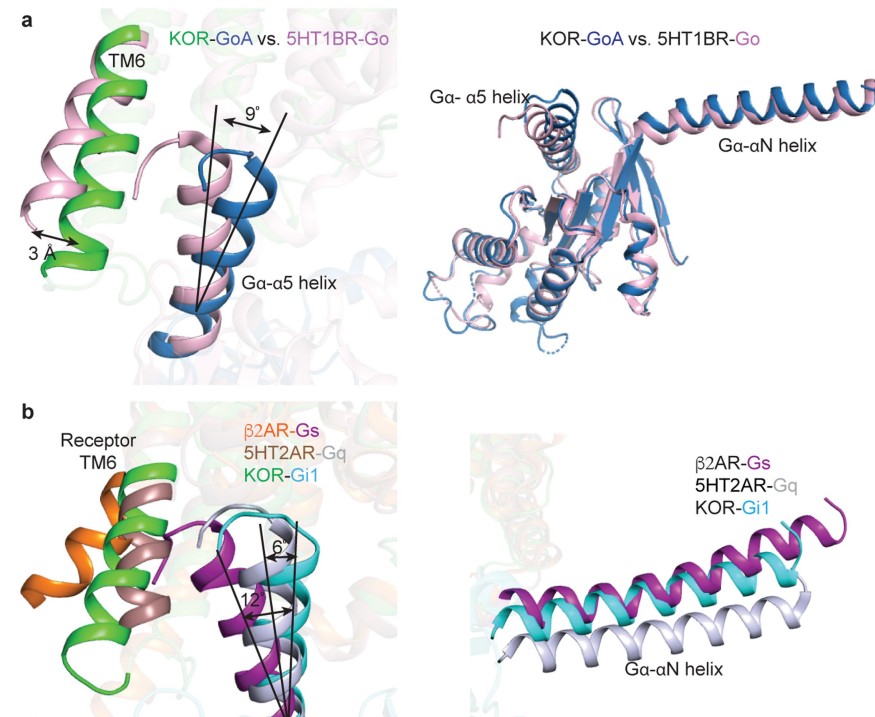

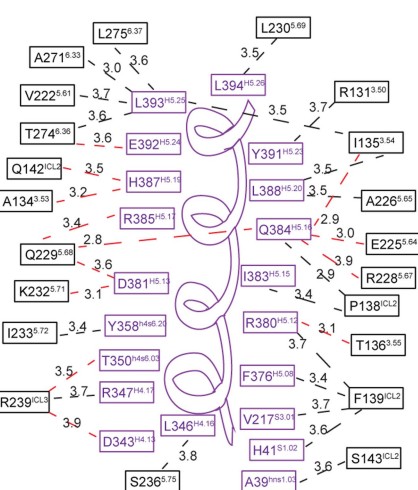
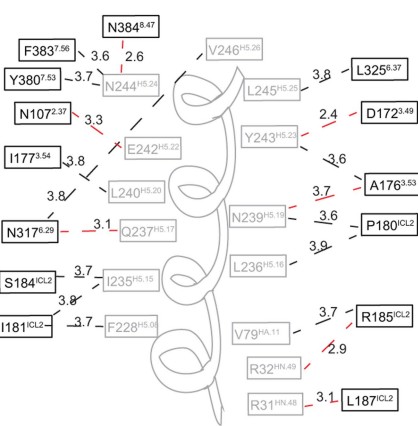

**Extended Data Fig. 9 | Comparison of KOR-coupled Gα subunit with other G protein families. a**. The KOR-GoA displays a different conformation from the 5-HT1BR-miniGo structure, including both TM6 of the receptor and α5 helix of the Gα subunit. **b**. Comparison of receptor-G protein interface among Gs, Gi1, Gq bound complexes. The TM6 of the receptor, αN and α5 helices of the Gα subunits adopt different conformations. **c**. The receptor-Gα interface of β2AR-Gαs and 5-HT2AR-Gαq. The dashed lines represent the closest distance between the intracellular receptor residues and the Gα residues. The distance cutoff is 4 Å. The interface area shown in the brackets was calculated by the online server PDBePISA.

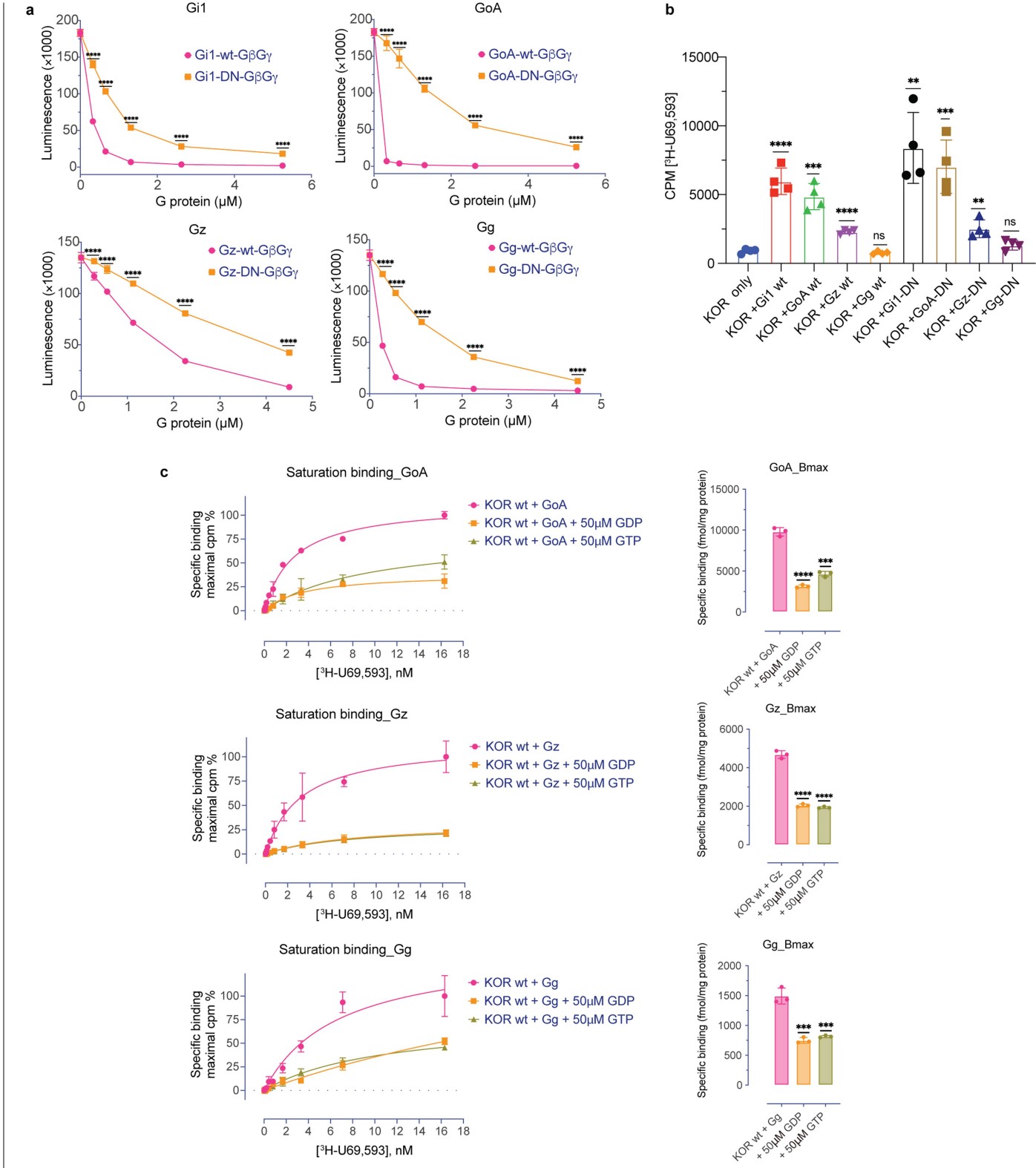

**Extended Data Fig. 10** | See next page for caption.

**Extended Data Fig. 10 | The presence of GDP or GTP reduces the allosteric potentiation of GoA, Gz, or Gg, respectively. a.** Comparison of functional activity between wild-type G proteins and engineered G proteins by GTP turnover assay. Data are luminescence ± s.e.m. of n = 3 biological replicates with each performed in triplicate. Significance analyses were performed using unpaired two-tailed student's t-test to compare each point in engineered G protein (G protein-DN) to the corresponding point in wild-type G proteins. (* = $p < 0.05$, ** = $p < 0.01$, *** = $p < 0.001$, **** = $p < 0.0001$). **b.** Measurement of allosteric potentiation of engineered G proteins on KOR agonist binding. The concentration of $^3$H-U69,593 was 8 nM, and the G protein was 250 nM. DN, dominant negative. Data are cpm ± s.e.m. from n = 4 biological replicates. Significance analyses were performed using the unpaired two-tailed student's t-test to compare each group to the KOR-only group (* = $p < 0.05$, ** = $p < 0.01$, *** = $p < 0.001$, **** = $p < 0.0001$, "ns" represents no significance; (KOR+GoAwt) vs KOR: $p = 0.0002$, (KOR+Ggwt) vs KOR: $p = 0.2469$, (KOR+Gi1DN) vs KOR: $p = 0.0012$, (KOR+GoADN) vs KOR: $p = 0.0007$, (KOR+GzDN) vs KOR: $p = 0.0029$, (KOR+GgDN) vs KOR: $p = 0.0719$). **c.** The saturation binding assays were conducted with or without 50 µM GDP/GTP. Data are Bmax ± s.e.m. from n = 3 biological replicates with each performed in duplicate. Statistical significance analyses between groups (with GTP/GDP and without GTP/GDP) are compared in the unpaired two-tailed student's t-test (* = $p < 0.05$, ** = $p < 0.01$, *** = $p < 0.001$, **** = $p < 0.0001$; (KOR+GoA) vs (KOR+GoA+GTP): $p = 0.0001$, (KOR+Gg) vs (KOR+Gg+GDP): $p = 0.0008$, (KOR+Gg) vs (KOR+Gg+GTP): $p = 0.0010$). Full quantitative parameters from this experiment are listed in Supplementary Table 15.

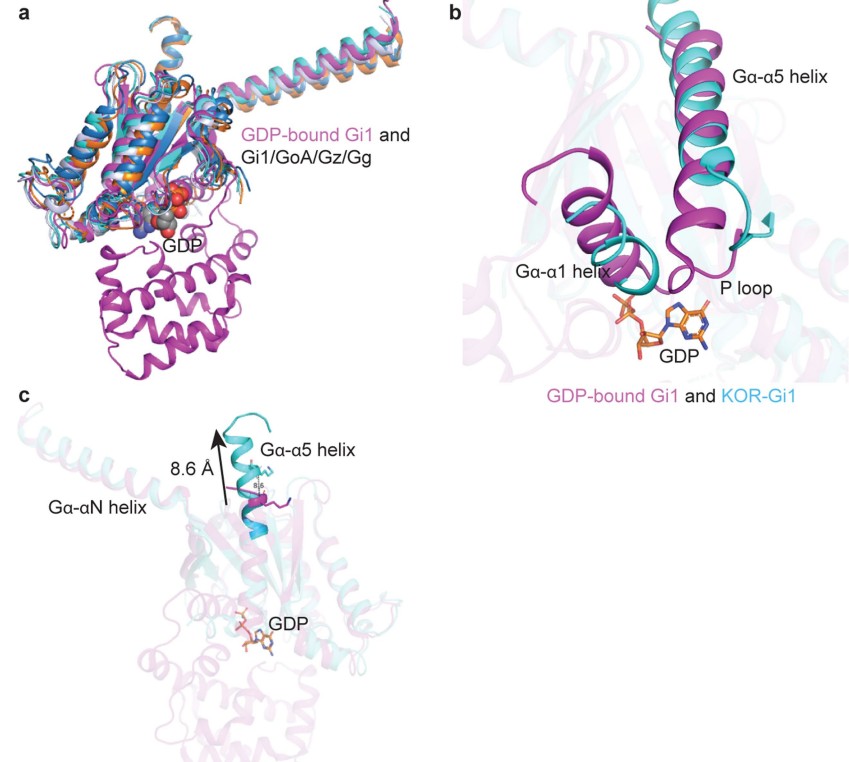

**a**

GDP-bound Gi1 and
Gi1/GoA/Gz/Gg

GDP

**b**

Gα-α5 helix

Gα-α1 helix

P loop

GDP

GDP-bound Gi1 and KOR-Gi1

**c**

8.6 Å

Gα-α5 helix

Gα-αN helix

GDP

GDP-bound Gi1 and KOR-Gi1

**Extended Data Fig. 11 | Comparison of uncoupled GDP-bound Gi1 and KOR-coupled nucleotide-free Gi1. a**. Comparison of GDP-bound Gi1 with KOR bound Gi1, GoA, Gz, and Gg. **b**. Conformational translocation of α1 and α5 helices upon GPCR engagement. **c**. α5 helix shows an 8.6 Å upward movement into the KOR intracellular core.

**Extended Data Table 1 | Cryo-EM data collection, refinement, and validation statistics**

| | KOR-Gi1-momSalB | KOR-GoA-momSalB | KOR-Gz-GR89,696 | KOR-Gg-GR89,696 |
|---|---|---|---|---|
| | EMD-27804 PDB: 8DZP | EMD-27805 PDB: 8DZQ | EMD-27807 PDB: 8DZS | EMD-27806 PDB: 8DZR |
| **Data collection and processing** | | | | |
| Magnification | 45,000 | 45,000 | 45,000 | 45,000 |
| Voltage (kV) | 200 | 200 | 200 | 200 |
| Electron exposure (e-/$\text{Å}^2$) | 47.4 | 42.7 | 43.8 | 29.07 |
| Number of movies used | 4175 | 3853 | 3529 | 5752 |
| Defocus mean (SD) μm | 1.2 (0.4) | 1.4 (0.4) | 1.1 (0.3) | 1.4 (0.2) |
| Pixel size (Å) | 0.88 | 0.88 | 0.88 | 0.88 |
| Symmetry imposed | C1 | C1 | C1 | C1 |
| Initial particle images (No.) | 969,195 | 1,431,336 | 973,407 | 1,618,725 |
| Final particle images (No.) | 370,551 | 991,076 | 643,193 | 725,271 |
| Map resolution (Å) | 2.71 | 2.82 | 2.65 | 2.61 |
| FSC threshold 0.143 | 0.143 | 0.143 | 0.143 | 0.143 |
| Map resolution range (Å) | 2.3-6.5 | 2.5-6.6 | 2.4-6.9 | 2.3-6.3 |
| **Refinement** | | | | |
| Initial model used (PDB code) | 6DDE | KOR-Gi1 in this study | | |
| Model resolution (Å) | 2.87 | 3.09 | 2.96 | 3.06 |
| FSC threshold | 0.5 | 0.5 | 0.5 | 0.5 |
| Map sharpening B factor ($\text{Å}^2$) | -69.58 | -48.19 | -76.93 | -94.28 |
| **Model composition** | | | | |
| Non-hydrogen atoms | 8864 | 8741 | 8529 | 8846 |
| Protein residues | 1130 | 1118 | 1115 | 1129 |
| Ligands | 1 | 1 | 1 | 1 |
| Protein - B factor ($\text{Å}^2$) | 52.78 | 47.41 | 55.08 | 54.67 |
| Ligands - B factor ($\text{Å}^2$) | 68.88 | 67.1 | 75.95 | 61.31 |
| **r.m.s. deviations** | | | | |
| Bond lengths (Å) | 0.004 | 0.003 | 0.003 | 0.003 |
| Bond angles (°) | 0.526 | 0.55 | 0.499 | 0.547 |
| **Validation** | | | | |
| MolProbity score | 1.47 | 1.46 | 1.59 | 1.48 |
| Clashscore | 5.43 | 4.54 | 5.45 | 4.91 |
| Poor rotamers (%) | 0 | 0 | 0 | 0 |
| **Ramachandran plot** | | | | |
| Favored (%) | 97.0 | 96.5 | 95.7 | 96.6 |
| Allowed (%) | 3.0 | 3.5 | 4.3 | 3.4 |
| Disallowed (%) | 0 | 0 | 0 | 0 |

# nature research

# Reporting Summary

Nature Research wishes to improve the reproducibility of the work that we publish. This form provides structure for consistency and transparency in reporting. For further information on Nature Research policies, see Authors & Referees and the Editorial Policy Checklist.

## Statistics

For all statistical analyses, confirm that the following items are present in the figure legend, table legend, main text, or Methods section.

| n/a | Confirmed | |
|---|---|---|
| ☐ | ☒ | The exact sample size (*n*) for each experimental group/condition, given as a discrete number and unit of measurement |
| ☐ | ☒ | A statement on whether measurements were taken from distinct samples or whether the same sample was measured repeatedly |
| ☐ | ☒ | The statistical test(s) used AND whether they are one- or two-sided *Only common tests should be described solely by name; describe more complex techniques in the Methods section.* |
| ☒ | ☐ | A description of all covariates tested |
| ☒ | ☐ | A description of any assumptions or corrections, such as tests of normality and adjustment for multiple comparisons |
| ☐ | ☒ | A full description of the statistical parameters including central tendency (e.g. means) or other basic estimates (e.g. regression coefficient) AND variation (e.g. standard deviation) or associated estimates of uncertainty (e.g. confidence intervals) |
| ☐ | ☒ | For null hypothesis testing, the test statistic (e.g. *F*, *t*, *r*) with confidence intervals, effect sizes, degrees of freedom and *P* value noted *Give P values as exact values whenever suitable.* |
| ☒ | ☐ | For Bayesian analysis, information on the choice of priors and Markov chain Monte Carlo settings |
| ☒ | ☐ | For hierarchical and complex designs, identification of the appropriate level for tests and full reporting of outcomes |
| ☒ | ☐ | Estimates of effect sizes (e.g. Cohen's *d*, Pearson's *r*), indicating how they were calculated |

*Our web collection on statistics for biologists contains articles on many of the points above.*

## Software and code

Policy information about availability of computer code

| | |
|---|---|
| Data collection | SerialEM V3.8.2 |
| Data analysis | Structure determination and refinement: cryoSPARC V3.1.0, Topaz 0.2.2, DeepEMhancer, UCSF Chimera 1.15, UCSF ChimeraX 1.3, PyMol 2.4.1, Coot 0.9.5, MolProbity (built in phenix 1.18.2), Phenix 1.18.2 Functional data analysis: GraphPad Prism 9 Ligand illustrator: ChemDraw 20.0. Docking and molecular simulation: CHARMM-GUI (https://www.charmm-gui.org/), AMBER v20.12 (including CUD and PMEMD), Gromacs simulation (version 2020.3), matplotlib v3.4.3, seaborn v0.11.2, pytraj v2.0.6 |

For manuscripts utilizing custom algorithms or software that are central to the research but not yet described in published literature, software must be made available to editors/reviewers. We strongly encourage code deposition in a community repository (e.g. GitHub). See the Nature Research guidelines for submitting code & software for further information.

## Data

Policy information about availability of data

All manuscripts must include a data availability statement. This statement should provide the following information, where applicable:

- Accession codes, unique identifiers, or web links for publicly available datasets
- A list of figures that have associated raw data
- A description of any restrictions on data availability

The coordinate and cryoEM map of KOR-Gi1-momSalB, KOR-GoA-momSalB, KOR-Gz-GR89,696, and KOR-Gq-GR89,696 have been deposited to PDB and EMDB databases with accession code 8DZP (EMD-27804), 8DZQ (EMD-27805), 8DZS (EMD-27807) and 8DZR (EMD-27806), respectively. PDB database (https://www.rcsb.org/) was used in this study to download the structures of 5-HT2AR-Gq (PDB ID 6WHA), MOR-Gi (PDB ID 6DDE), β2AR-Gs (PDB ID 3SN6) and 5HT1B-miniGo complex (6G79) for structural analysis.

# Field-specific reporting

Please select the one below that is the best fit for your research. If you are not sure, read the appropriate sections before making your selection.

☒ Life sciences  ☐ Behavioural & social sciences  ☐ Ecological, evolutionary & environmental sciences

For a reference copy of the document with all sections, see nature.com/documents/nr-reporting-summary-flat.pdf

# Life sciences study design

All studies must disclose on these points even when the disclosure is negative.

| Sample size | Sample size was not predetermined by statistical methods. For functional assays ( such as cAMP inhibition, BRET2, Radio ligand binding assay, ELISA), there are at least three technical replicates and biological replicates that are reported in the figure legends. For the cryo-EM studies, the number of images is determined by the available microscope time and the requirement of the resolution and 3D reconstruction of EM map. The number of images used for structural determination is sufficient to gain high-resolution maps and build accurate atomic models. |
|---|---|
| Data exclusions | No data were excluded for this study. |
| Replication | For functional assay, data were replicated using technical and independent biological replicates. See figure legends for specific details. For the GPCRome assay, one biologically independent experiment was performed with n=4 technical replicates. The four replicates were reliably reproduced. |
| Randomization | No data is required randomization, because this study did not allocate experimental groups. For cryoEM study, meshes on the grids with good ice thickness were randomly selected for data collection. |
| Blinding | No blinding was performed in this study. For both cryoEM structure determination and functional studies, blinding is not necessary due to the nature of these experiments do not requires subject assessment of the data that may influence the validity of the results |

# Reporting for specific materials, systems and methods

We require information from authors about some types of materials, experimental systems and methods used in many studies. Here, indicate whether each material, system or method listed is relevant to your study. If you are not sure if a list item applies to your research, read the appropriate section before selecting a response.

### Materials & experimental systems

| n/a | Involved in the study |
|---|---|
| ☐ | ☒ Antibodies |
| ☐ | ☒ Eukaryotic cell lines |
| ☒ | ☐ Palaeontology |
| ☒ | ☐ Animals and other organisms |
| ☒ | ☐ Human research participants |
| ☒ | ☐ Clinical data |

### Methods

| n/a | Involved in the study |
|---|---|
| ☒ | ☐ ChIP-seq |
| ☒ | ☐ Flow cytometry |
| ☒ | ☐ MRI-based neuroimaging |

## Antibodies

| Antibodies used | gp64-PE antibody (expression system, #97-201), anti-FLAG-horseradish peroxidase-conjugated antibody (Sigma, A8592) |
|---|---|
| Validation | gp64-PE antibody was purchased from Expression Systems and was used for baculovirus titration. The detailed information can be found at https://expressionsystems.com/product/gp64-pe-antibody/ anti-FLAG-horseradish peroxidase-conjugated antibody was ordered from Sigma-Aldrich and was used in ELISA experiment for the detection of KOR receptor (wt and mutants) expression level. The detailed information can be found at https://www.sigmaaldrich.com/US/en/product/sigma/a8592 |

## Eukaryotic cell lines

Policy information about cell lines

| Cell line source(s) | HEK293T cells were purchased from the American Type Culture Collection (ATCC, ATCC CRL-11268). Spodoptera frugiperda (Sf9) cells are from Expression Systems (#94-001S). |
|---|---|
| Authentication | All cells used in this study are commercial as indicated in the manuscript. HEK293T cells were authenticated by the supplier |

Authentication

(ATCC) using morphology and growth characteristics, and STR profiling. Sf9 cells are commercial and obtained from vendors as indicated in the manuscript. No additional authentication was performed by the authors of this study.

Mycoplasma contamination

HEK293T cells have been tested and shown to be free from mycoplasma (Hoechst DNA strain and Direct Culture methods employed). Sf9 cell line was certified as mycoplasma-free by the source company.

Commonly misidentified lines
(See ICLAC register)

No commonly misidentified cell lines were used.

