## [Peer Review File · Nature]

Manuscript Title: Ligand and G Protein Selectivity in Kappa Opioid Receptor

Reviewer Comments & Author Rebuttals

Reviewer Reports on the Initial Version:

Referees' comments:

Referee #1 (Remarks to the Author):

This study combines the use of cryo-EM and molecular pharmacology to study the structural determinates for Ligand and G protein selectivity in Kappa Opioid receptors. The authors report 4 individual cryo-EM structures of the Kappa Opioid receptor docked to different ligands (methoxymethyl-Salvinorin B/ GR89,696) and different G proteins (Gi1, GoA, Gz, Gg). The authors then use the structures to aid interpretation of data related to signalling outcomes (cAMP inhibition) and G protein activation and explain how these features map to the different structures.

I am not able to review the cryo-EM work as I am not a structural biologist so I will restrict my comments to the molecular pharmacology and G protein work. Overall, I found this aspect of the study extremely disappointing.

Major criticism of the data:

ALL the data lacks any statistical analysis. The authors make continuous comments such as “significantly reduced the potency” or “not observe significant changes” or “expression levels of G α mutants were found to be comparable” etc. All these comments must be disregarded since there are no statistical analysis to back them up. Indeed, the methods section does not even contain a section related to data analysis and statistical analysis.

A further concern I have is that all the experimental data is reported as S.E.M of n=3 “biological repeats”. For pharmacological data I would expect this to be S.E.M. of n=3 “technical repeats” e.g., where the experiment is repeated in duplicate or triplicate on the day as an individual repeat. Based upon these 2 facts, I cannot agree with any of the authors conclusions.

Further examples where this is most prevalent relate to expression data for the considerable mutant receptors and G proteins. We are told that “expression levels of G α mutants were found to be comparable to those of the wild-type G α , as quantified by the luminescence they produced in the BRET assays” however when you consider at the data provided this is impossible to conclude e.g. “for K314A the Rluc counts are ~90,000 compared to WT of 42,000. This is at least 2-fold.” This is but only one example. Many more are prevalent in the receptor expression data and all the other G proteins. Without some attempt to correct for the expression I cannot see how the authors can conclude anything about the activities etc.

This lack of statistical rigour really concerns me and therefore I am also unsure as to the validity of the models and cryo-em work based upon the flippant comments related to the expression and pharmacology work.

Specific questions comments etc:

For the cryo-EM structures of the complexes:

1. In the section titled: "Overall structures of KOR-G protein complexes" – authors mention "GoA and Gz are among the most abundant G proteins expressed in central and peripheral neuronal", what about Gi2 which has highest expression level in nearly all tissues? Indeed, why use Gi1 not Gi2 for a structure? What is the rationale for the 4 G proteins used?
2. Why do the authors use different ligands for different G proteins?
 - i. momSalB failed to stabilise KOR-Gz and KOR-Gg but what about GR89696 with KOR-Gi1 and KOR-Goa
 - ii. If the reference compound U50,488 can activate all the Gi/o proteins, why not use this if they want to study the critical residues involved in differential G proteins coupling with KOR
3. The sentence "It is worth pointing out that momSalB failed to generate a stable complex of KOR-Gz or KOR-Gg compared to GR89,696, suggesting a ligand-specific transducer coupling." Was this tested using the BRET G protein assay?
4. The comment "The overall structures of KOR in Gi1/oA/z/g bound states are similar to the previously nanobody-stabilized active conformation (KOR-Nb39)²¹" – does this limit novelty here in the study?

Interactions of KOR with hallucinogen salvinorins.

1. A continuous use of the phrases "Significant effect" but no stats.
2. Expression data –what does this mean for mutants with higher levels of expression? How is this factored into the data?
3. Figure 2C – why is D138 not shown on the figure?
4. Extended Figure 4 - No consistency for the color scheme of the mutants
5. Extended Figure 5 - How did the authors select the mutants
Why D138N but not mutate to A like the other positions
Why do they only select D138 and H291 for double mutation?
What is meant by "high vulnerability of momSalB"?
6. How do the authors conclude that "This ligand-binding similarity, irrespective of G protein subtypes, is partially due to the nucleotide-free state of Gasubunits"

Structural basis of G protein subtype selectivity

1. The authors use "BRET-based transducerome profiling" what are these reporters, the methods makes no mention of the Rluc8 in the constructs, the different G β and G γ used. What are the reporters?
2. There is inconsistent use of reference compounds, colours in figures, symbols etc. Why do they use U50,488 for Gi1-3, GoA, GoB and Gz but use GR89,696 for Gg? Why was the same compound not used to test all G protein complexes first?
3. Extended Figure 6, reference is U50,588 instead of U50,488
4. Extended Figure 7, use U50,488 as reference for Gg
5. Why were not all the mutants shown in cAMP assay/cell surface expression tested in the G protein assay
 - i. Where is R170A?

b. Representation of the G protein assay data

i. Extended Figure 6 and extended Figure 7b are not informative at all

c. Consistency of the color

i. Figure 4

1. For b and c, colors are correlated with G protein

2. For d and e, not correlated here

d. What is the rationale of the choosing mutants"

i. For the KOR

1. What was the rational for showing the data in Figure 3c-f?

a. Only select the "working" ones?

ii. For G proteins,

1. GoA, why do they use G350Y but not G350A?

a. Also, in extended Figure 7 c, only GoA has a different color scheme for the mutants from other G proteins

6. For most of the graphs it is hard to see any differences with the symbols used. No more that 6 lines should be used on graphs for easy of viewing.

7. The sentence "Unexpectedly, R1563.50A conversely affects the BRET signal during KOR-Gg coupling, likely by slowing down Gg dissociation and thus leading to accumulation of stable KOR-Gg complexes." How is this concluded? Why not show the kinetic traces for the data?

8. The sentences: "However, we did not observe significant changes in potency in agonist-mediated G protein activation (Extended Data Fig. 7a and 7b). The expression levels of G α mutants were found to be comparable to those of the wild-type G α , as quantified by the luminescence they produced in the BRET assays" I do not believe this without statistical analysis as discussed above.

9. The sentence "The high-affinity binding sites for 3H-U69,593 were increased 38-, 20-, 9-, and 3-fold in the presence of Gi1, GoA, Gz, and Gg, respectively." How can this be concluded without statistical analysis?

10. The authors state "The calculated KB and α -cooperativity displayed a similar pattern to that observed in the saturation binding, in which Gi1 has the highest binding affinity and allosteric effects at KOR in the presence of agonists, and Gg has the least (KB, nM= 383(Gi1), 563(GoA), 1023(Gz), 1587(Gg); Log α = 1.41(Gi1), 1.21(GoA), 1.06(Gz), 0.89(Gg)) (Fig. 5d)." How were the Log α calculated? What model was used? Why is there no error? How can they conclude anything about how important these values are without stats?

Conclusion: This section cannot be reviewed objectively since the lack of statistical analysis means that any conclusions the authors make are subjective and do not stand up to rigorous questioning.

Methods: Lack of any data analysis, models used and of course no mention of statistical analysis. The methods lack any detail for how to reproduce the study for example, we are not told any information about the "BRET-based transducerome" except for a passing reference to the TRUPATH system. When one looks at the methods section the we are told that "HEK239T cell were transfected

with a 1:5 of KOR : (G α (Gi1, GoA, Gz)-G γ -G β DNA or a 6:5 ratio of KOR : Gg-G γ -G β DNA.” There is no mention of Rluc or YFP for the BRET expts and no explanation as to why different ratios were used. If this is TRUPATH, which G γ -G β subunits were used? If not, then more detail is needed on these “BRET-based transducerome”. This is very important because we are required to compare the reporter results, yet we have no information about the constructs.

Referee #2 (Remarks to the Author):

This is an interesting manuscript that has determined the active state structure of the kappa opioid receptor with multiple G protein heterotrimers. These structures show differences for different chemotypes and provide insight into the molecular requirements for G protein selectivity.

The work is highly original and this reviewer is unaware of a previous Cryo-EM structure of the kappa opioid receptor. The data and methodology are appropriate and valid. The use of statistics also appears appropriate. One issue that is concerning is there is no mention where the compounds were procured or their level of purity. At a minimum, this information should be added to the supplemental material.

The following points are raised to improve the work presented:

1. Page 3, Lines 62-65: A citation from 2011 (ref 13) does not qualify as "recently".
2. Page 5, Line 90: Reference 18 is not appropriate. It does not show that momSalB is psychotropic. A more appropriate reference would be Baker 2011. The ref 18 reports the synthesis of MOMSalB.
3. Page 6, Line 122: Reference 25 is not appropriate. This reference reports biological activity. A more appropriate citation would be to the first isolation which is Ortega et al 1982. Suggest including both.
4. Page 8, Line 168: The structure of salvinorin B is incorrect in the sentence and in extended data fig 5c and 5d. Salvinorin B contains an OH group, not OCH₃. Similarly, the groups for momSalB and (OCH₂OCH₃) and ethoxySalB (OCH₂OCH₂CH₃) need to be corrected in the text.
5. Pages 15-18, Lines 331 - 405: The manuscript contains little discussion of the present work is similar or different than Vardy et al 2013 which described the chemotype-selective modes of action of kappa agonists.

Referee #3 (Remarks to the Author):

In their manuscript “Ligand and G Protein Selectivity in Kappa Opioid Receptor Revealed by Structural Pharmacology”, Han and coauthors provide novel insights into the selective G protein coupling for the kappa opioid receptor (KOR). The recent advances in single particle cryo-EM have allowed structure determination of a number of individual GPCRs in complex with distinct classes of heterotrimeric G proteins (such as US28-Gi1 and -G11, GP139-Gs and -Gi1, NK1R-Gq and -Gs, MRGX2-Gi1 and -Gq, CCK2-Gi2 and -Gq, CCK1-Gs, -Gi1 and -Gq, GCGR-Gs and -Gi). While these structures provided important insights into the molecular determinants of G protein coupling of different G protein families to GPCRs, there is no clear consensus sequence or structural feature known that clearly determines G protein coupling specificity. Even less is known for the selective coupling of a single GPCR to different members of the same G protein family. To my knowledge only one receptor ($\alpha 2B$) has been structurally described in complex with two distinct G protein subtypes (Gi1 and GoA) from the Gi/o G protein family. On this background, the authors determined four cryo-electron microscopy structures of the fully active state of KOR in complex with four different G protein subtypes (Gi1, GoA, Gz and Gg) and bound to the psychotropic salvinorin analog momSalB or the highly potent KOR agonist GR89,696. Based on the obtained structural insights, the authors propose molecular determinants critical for subtype-selective ligand binding and G protein subtype specificity. Furthermore, the authors provide comprehensive mutagenesis, cell signaling and ligand binding studies to support their structural findings. The major success of this study is that it provides four high resolution structures of the KOR in complex with four different G protein subtypes. Two of these complexes provide a first structural view on Gz and Gg subtype coupling to a GPCR. Another important finding is the structural determination of the binding pose of a salvinorin A analog (momSalB) to the orthosteric ligand binding site of the KOR. Salvinorin A is a unique naturally occurring hallucinogenic ligand that binds very selective to the KOR without containing a single nitrogen atom. The KOR structures bound to momSalB and the selective agonists GR89,696 thus provide important insights into ligand binding specificity at the KOR, a promising target for the treatment of pain and addiction. Overall, this is an interesting study that will be of broad interest in the GPCR field.

While the conclusions of the study are sound, some caveats need to be addressed:

Specific comments:

- 1) While previous studies have shown that KOR can couple to Gg in transfected HEK293 cells (Olsen et al., 2020) and when expressed in bitter taste cells of genetically engineered mice (Mueller et al., 2005), it is not really known, if this interaction is of physiological relevance. As stated by the authors, the KOR is mainly expressed in the brain in pain-related neurons, whereas Gg is mostly found in taste receptor cells and in the gut. This should be discussed in the manuscript, especially with regard to the mutagenesis analyses that show the biggest effects for Gg in comparison to the other Gi/o family subtypes.
- 2) The authors discuss differences in the GR89,696 and momSalB-dependent generation of biochemically stable complexes, especially for Gz and Gg complexes, as possible effects of ligand-specific transducer coupling. Do the authors have cell signaling data that would support this? The data presented in Fig. 3a and 3b shows significant differences in the potency of GR89,696 for activation of Gz and Gg, but what about Gi1 and GoA? It would be helpful to test both ligands for all

four G protein subtypes to support the ligand-specific transducer coupling mentioned above.

3) I am not sure what the authors want to say in the last paragraph on page 8. I totally agree that structures of nucleotide-free GPCR-G protein complexes represent a very transient state in which the conformation of the receptor on the intracellular side is stabilized by the G protein. However, I am not sure why the ligand binding pose should necessarily be different in the presence of GDP. I would rather assume that in the presence of GDP, the complex becomes more dynamic, thus allowing TM6 and other TMs to sample alternative conformational states. The authors need to clarify this part.

4) The authors discuss the role of N336 in G protein coupling specificity. Based on the structure, they report that N336 does not form a H-bond interaction with the C-terminal alpha5 helix of the G protein. However, the density in this region is not great and the refined map suggest a different side chain position that would allow formation of a polar interaction with K349. Also, for the Gg complex the density does not show enough features to model the side chain orientation of N336. The authors should try to improve the map quality/model or perform MD simulations to analyze intermolecular interactions in this area .

5) The key point of the discussion of the interfaces between KOR and the different G protein subtypes and the comparison with the β 2AR-Gs and 5HT2A-Gq complex is not clear. Do the authors want to say that the overall size of the interacting surface area of the individual G protein subtypes is relatively conserved across different ClassA GPCRs complexes? Furthermore, is there a real correlation between the interaction area and the kinetics of G protein association and dissociation or G protein activation in general? It would be helpful to list or include a schematic figure with the interacting residues and the size of the interacting area. Since no experimental kinetic data is presented, time-resolved BRET studies could be performed to support the model of the authors that the receptor-G protein interface impact the kinetics of G protein activation.

6) Please, provide the error for the K_B and α values in Fig. 5d

7) Inspection of the maps and models provided to the reviewers shows that several side chains were incorrectly modeled. The entire models should be carefully checked and remodeled, but especially the extracellular and intracellular loops of the receptor and regions around the P-loop and TCAT motif of the G proteins that show rather fragmented density. Entire residues or side chains that do not show any density should be deleted, e.g. in all maps, no or very fragmented density is observed for the N-terminal residues 51-61, indicating that this region is very flexible. I would suggest to remove this part completely.

Minor points:

Line 110: Differences between the Nb and G protein-bound KOR structures should be described for the G protein complexes, meaning that you should write that in the G protein complex, TM6 moves 2.8Å away from TM5 in comparison to the Nb-bound state.

Line 142: Asparagine can form H-bonds but no salt-bridge interactions.

Line 181: I would not consider GDP to be a substrate for G α subunits, because they do not hydrolyze it to GMP.

Fig. 2d: Please, change GR8,9696 to GR89,696

Fig. 3c-f: Please, include labels for the TMs.

Fig. 4a: This figure is very crowded and hard to read. I suggest to present LogEC50 values as a bar

graph similar to Fig. 2d.

Line 245-248: Please, also cite Rose et al., JACS (2014) and Glukhova et al., ACS Pharm. Transl. Sci. (2018)

Line 251: Please, also cite Flock et al., Nature (2017); Inoue et al., Cell (2019)

Line 265: The receptor also interacts with other structural components of the G protein, including β_6 and top of β_3 . Therefore, it should be highlighted that you are focusing on α_N and α_5 only.

Line 356: Please change reference 58 to Wingler et al., Cell (2019)

Line 374-377: This has been done for the B2AR receptor. Please, cite Wenzel-Seifert and Seifert, Mol. Pharm. (2000)

Line 395: Please, also consider to add references Huang et al., Cell (2021) and Furness et al., Cell (2016)

Line 549: Please, provide the reference

The model of the G γ complex contains all the hydrogens. Please, remove those from the pdb file.

Author Rebuttals to Initial Comments:

We would like to thank the reviewers for their comments. We feel that the critiques you have raised have helped improve the rigor of our statistical analyses and also clarity of presentation for our work. For example, in this revision, we have included more clear details of the methods and data analysis. Below you will find in light blue a point-by-point response to each of critiques and comments that you raised.

Referee #1 (Remarks to the Author):

This study combines the use of cryo-EM and molecular pharmacology to study the structural determinates for Ligand and G protein selectivity in Kappa Opioid receptors. The authors report 4 individual cryo-EM structures of the Kappa Opioid receptor docked to different ligands (methoxymethyl-Salvinorin B/ GR89,696) and different G proteins (Gi1, GoA, Gz, Gg). The authors then use the structures to aid interpretation of data related to signalling outcomes (cAMP inhibition) and G protein activation and explain how these features map to the different structures.

I am not able to review the cryo-EM work as I am not a structural biologist so I will restrict my comments to the molecular pharmacology and G protein work. Overall, I found this aspect of the study extremely disappointing.

We thank the reviewer's comments to help improve the rigor and quality of the work. In this revision, we have included the details of methods and data analysis.

Major criticism of the data:

ALL the data lacks any statistical analysis. The authors make continuous comments such as "significantly reduced the potency" or "not observe significant changes" or "expression levels of G α mutants were found to be comparable" etc. All these comments must be disregarded since there are no statistical analysis to back them up. Indeed, the methods section does not even contain a section related to data analysis and statistical analysis.

We thank the reviewer for pointing out this issue and apologize for allowing the manuscript to be submitted without proper statistical analysis. We have now added statistical analysis to the figures and supplemental tables. We have also accordingly modified our descriptions in the manuscript to make them consistent with the statistics analysis. Also we now have included details of how we performed the data analysis in the Methods section and statistical analyses in the figure legends.

A further concern I have is that all the experimental data is reported as S.E.M of n=3 "biological repeats". For pharmacological data I would expect this to be S.E.M. of n=3 "technical repeats" e.g., where the experiment is repeated in duplicate or triplicate on the day as an individual repeat. Based upon these 2 facts, I cannot agree with any of the authors conclusions.

We apologize for the lack of clarity and have stated this more clearly in the Methods and presentation of the data. To clarify, our dose-dependent functional assays (cAMP inhibition, BRET, or binding assays) were all performed with $n \geq 3$ independent biological repeats at different times. For every single experimental run, each compound or construct

was tested in duplicate or triplicate. Specifically, the dose-dependent graphs were from the global fit of grouped data \pm s.e.m. from $n \geq 3$ independent biological repeats done in duplicate or triplicate. The bar-graphs represent the mean data (LogEC50, Bmax, or OD₄₅₀) \pm s.e.m. from $n \geq 3$ independent biological repeats done in duplicate or triplicate. Full quantitative parameters from each graph have been summarized in the Supplemental Tables file. The details of the data analysis have been added to the METHODS section and below:

For BRET2 and cAMP-inhibition assay: In the case of comparing more than two groups, logEC50 values were first analyzed by one-way ANOVA. If significant, the Dunnett's multiple comparisons test was used to compare each mutant to the wild-type one, and the Tukey's multiple comparisons test was used to compare logEC50 values between each group. In the case of comparing two groups, logEC50 values were analyzed via the unpaired two-tailed student's t-test to compare each mutant to the wild-type receptor.

For ELISA assays: The OD₄₅₀ values of each mutant were normalized by the wild-type KOR receptor (normalized as 100%), then resultant values were first analyzed by one-way ANOVA. If significant, a Dunnett's multiple comparisons test was used to compare each mutant to the wild-type receptor.

For G protein expression studies : The Rluc values of each mutant were normalized by the wild-type G protein. (normalized as 100%), then resultant values were first analyzed by one-way ANOVA. If significant, a Dunnett's multiple comparisons test was used to compare each mutant to the wild-type G protein.

For radio ligand binding and GTP turnover assays: Data were analyzed via the unpaired two-tailed student's t-test.

In one-way ANOVA and unpaired two-tailed student's t-test analysis, the significance threshold was set at $\alpha = 0.05$. Asterisk (*) represents $p < 0.05$. (**) represents $p < 0.01$, (***) represents $p < 0.001$. (****) represents $p < 0.0001$. 'ns' represents no significance.

Further examples where this is most prevalent relate to expression data for the considerable mutant receptors and G proteins. We are told that "expression levels of G α mutants were found to be comparable to those of the wild-type G α , as quantified by the luminescence they produced in the BRET assays" however when you consider at the data provided this is impossible to conclude e.g. "for K314A the Rluc counts are ~90,000 compared to WT of 42,000. This is at least 2-fold." This is but only one example. Many more are prevalent in the receptor expression data and all the other G proteins. Without some attempt to correct for the expression I cannot see how the authors can conclude anything about the activities etc.

We thank the reviewer for pointing this out. We have now updated this part to make the measurement of expression level consistent to that in the cAMP inhibition and BRET assays. Specifically, we re-examined the expression of each construct at 48 h after transfection, which was the same timing when we tested the compounds or constructs. We then performed statistical analysis compared to the wild type receptor (**Figure 1A**) or G proteins (**Figure 1B**). As the reviewer pointed out, a few mutants expressed

significantly higher or lower than the wild type (**Figure 1**). To test whether these increased or decreased expressions affect G protein coupling, we performed BRET2 assays (**Figure 2A**) by transfecting different amounts of KOR (**Figure 2B**) or G proteins (**Figure 2C**), and confirmed that the ~2-fold variation of the expression minimally affect the ligand potency or efficacy. We have added this information to the figure legends in Extended data fig. 4e, 11a, and 14b, and have updated the bar-graph figures with statistical analysis.

Figure 1. Measurement of expression levels of KOR or G α protein mutants. A. Expression level of KOR binding pocket mutants (left panel) or KOR-G protein interface mutants (right panel) examined by ELISA. Bar graphs represent mean OD450 \pm s.e.m. values from n=3 independent biological replicates performed in triplicate. Statistical significance for each mutant is compared in a one-way analysis of variance (ANOVA) with the Dunnett's multiple comparisons test to the wild-type KOR (* = $p < 0.05$, ** = $p < 0.01$, *** = $p < 0.001$, **** = $p < 0.0001$). **B.** Expression levels of G α mutants (Gi1, GoA, Gz, and Gg) examined luminescence counts. Each G α is tagged with a luciferase. GoA has an Ala at position 345, so it was mutated to the corresponding residue Gln (Q) in Gz. Bar graphs represent mean luminescence counts \pm s.e.m. from n=3 independent biological replicates. Statistical significance for each mutant is compared in a one-way analysis of variance (ANOVA) with the Dunnett's multiple comparisons test to the wild-type G α (* = $p < 0.05$, ** = $p < 0.01$, *** = $p < 0.001$, **** = $p < 0.0001$, 'ns' represents no significance).

A Cell-Surface Expression of KOR-Binding Pocket Mutants (ELISA) Cell-Surface Expression of KOR-G Protein Interface Mutants (ELISA)

B

Figure 2. Effects of different expression levels of KOR or G α proteins. **A.** Schematic representation of BRET2 based functional assay. The G α is luciferase (Rluc) tagged and G γ is GFP tagged. Cells were transfected with receptor, G α -Rluc, G β , and G γ -GFP plasmids. **B.** Effects of different KOR expression levels on KOR-mediated Gi1 activation in the BRET2 assay. Curve graphs represent the global fit of grouped data \pm s.e.m. from n=3 independent biological replicates. **C.** Effects of different G α expression levels on KOR-mediated G protein activation in the BRET2 assay. Curve graphs represent the global fit of grouped data \pm s.e.m. from n=3 independent biological replicates performed in triplicate.

This lack of statistical rigour really concerns me and therefore I am also unsure as to the validity of the models and cryo-em work based upon the flippant comments related to the expression and pharmacology work.

We agree that the lack of statistical analysis weakened the rigor of the work. Thus, in our revised work we have now included more analysis and experiments to address the concerns.

We have further refined the structures to improve the quality based on other reviewers' comments. In addition to statistical parameters that was previously provided in Extended Data Table 1 and Extended Data Figure 1 and 2, a validation report related to each structure from Protein Data Bank is also provided.

Specific questions comments etc:

For the cryo-EM structures of the complexes:

1. In the section titled: "Overall structures of KOR-G protein complexes" – authors mention "GoA and Gz are among the most abundant G proteins expressed in central and peripheral neuronal", what about Gi2 which has highest expression level in nearly all tissues? Indeed, why use Gi1 not Gi2 for a structure? What is the rationale for the 4 G proteins used?

We agree with the reviewer that other G proteins, such as Gi2, Gi3, and GoB, are equally interesting and important given their different expression pattern and roles. In this study, the rationale of choosing Gi1, GoA, Gz, and Gg, is based on their sequence identity. As shown in the manuscript Fig. 1b, using Gi1 as a template, the sequence identity of other G proteins is Gi2 (87.89%), Gi3 (93.79%), GoA (72.75%), GoB (72.39%), Gz (67.23%), and Gg (68.36%). Based on this sequence identity, we divided them into four sub-groups (Gi1/i2/i3, GoA/oB, Gz, and Gg) and then determined the structures of one representative in each sub-group. Although we propose the structures of other G proteins, Gi2/Gi3 and GoB, likely mimic the respective Gi1 and GoA structures based on the sequence similarity, we agree it is worth pursuing their structures in the future. We have also modified our descriptions in the manuscript to provide a clear rationale why we target the four G proteins in our study.

2. Why do the authors use different ligands for different G proteins?

i. momSalB failed to stabilise KOR-Gz and KOR-Gg but what about GR89696 with KOR-Gi1 and KOR-Goa

One of the primary goals of this work was to use momSalB to determine the complex structures of KOR with Gi1, GoA, Gz, and Gg, respectively. momSalB is a salvinorin analog and is a highly selective, non-nitrogenous KOR agonist. While we succeeded to determine the structures of momSalB bound KOR-Gi1 and KOR-GoA structures, cryo-EM experiments of KOR-Gz or KOR-Gg bound to momSalB only yielded low-resolution reconstructions (~4.5-5 Å resolution) that prevented delineation of detailed molecular interactions (**Figure 3**).

We thus selected another highly selective and potent KOR agonist, GR89,696, that displays stronger potency in activating Gz and Gg compared to several other KOR agonists (Olsen et al., Nat Chem Biol, 2020) and allowed us to determine the structures of KOR-GR89,696-Gz and KOR-GR89,696-Gg complexes successfully. We believe that GR89,696 bound KOR-Gi1 or GoA could be feasible given its high potency, the four

structures we determined provides us not only insight into the G protein subtype selectivity, but also more details of ligand-specific interactions at KOR given that momSalB and GR89,696 are both highly selective agonists at KOR. We have modified this part in the manuscript to make it clearer “However, cryo-EM experiments of KOR-Gz or KOR-Gg bound to momSalB only yielded low-resolution reconstructions (~4.5-5 Å resolution) that prevented the delineation of detailed molecular interactions. Thus, we leveraged another highly potent KOR agonist, GR89,696, to obtain high-resolution structures of the KOR-Gz and KOR-Gg.”

Figure 3. Comparison of 2D-classification images from the KOR-momSalB-Gi1, KOR-momSalB-Gz, or KOR-momSalB-Gg samples. The images showed that KOR-Gi1 particles were well aligned and had clear features in each class, whereas KOR-Gg and KOR-Gz particles in each class were low resolution likely caused by the dynamic conformation of particles in vitro.

ii. If the reference compound U50,488 can activate all the Gi/o proteins, why not use this if they want to study the critical residues involved in differential G proteins coupling with KOR

It is true that U50,488 could activate all Gi/o subtypes. In the functional validation of critical residues, U50,488 was always tested in parallel with momSalB and GR89,696 and was used as a reference ligand. Because our novel structures of KOR-momSalB or KOR-GR89,696 were resolved, they enabled us to gain better understanding of the correlations between the KOR residues and ligand's functional activity considering the structure of U50,488 bound KOR is unavailable.

There are another two reasons we used momSalB or GR89,696 for structural determination. First, to determine the structure of KOR-G protein complex, the ligand should not only activate the G protein robustly, but also the ligand must bind and stabilize the KOR-G protein complex for purification, vitrification, and subsequent data collection.

Thus, the selection criteria of a ligand for stable GPCR-G protein assembly is generally that this ligand should have a slow off-rate and highly potent agonist, with respect to the partner proteins in the complex. In our study, both momSalB and GR89,696 display higher potency than U50,488 for each G protein subtype (Figure 4A). Second, in an unpublished study, we found that U50,488 poorly stabilized the active-state KOR compared to another KOR agonist MP1104 (Figure 4B). We believe that the structure of KOR-U50,488-G protein complex could be attempted in future studies with thermostabilized mutations to increase the complex stability.

Figure 4. Measurement of different G protein subtype activation by KOR agonists by BRET2 assays. **A.** Ligand-dependent G protein subtype activation showed that GR89,696 is more potent than momSalB or U50,488 in BRET2-based assays. Curve graphs represent the global fit of grouped data \pm s.e.m. from n=3 independent biological replicates performed in triplicate. **B.** Purification of KOR in the presence of MP1104 or U50,488 showed that U50,488 led to lower yield of KOR compared to MP1104 (an agonist co-crystallized with the active-state KOR-Nb39), indicating that U50,488 does not stabilize KOR as well as MP1104.

3. The sentence “It is worth pointing out that momSalB failed to generate a stable complex of KOR-Gz or KOR-Gg compared to GR89,696, suggesting a ligand-specific transducer coupling.” Was this tested using the BRET G protein assay?

We thank the reviewer for pointing out this confusion. The BRET2-G protein assay showed that momSalB could activate all four G protein subtypes with different potency (**Figure 4A**). However, during the in-vitro assembly of KOR-momSalB-Gz or Gg complex, we failed to readily obtain high-quality complexes and high-resolution reconstructions during cryoEM data processing, which prevented us from further analysis. For example, during the 2D-classification of our cryoEM data, the 2D images of the G protein part displayed low resolution, suggesting the formation of a dynamic or incomplete assembly KOR-G protein complex (**Figure 3**).

We have now included the BRET-G protein results of momSalB and GR89,696 in activating all four G protein subtypes, and we have modified this sentence in Page 5 in the manuscript.

4. The comment “The overall structures of KOR in Gi1/oA/z/g bound states are similar to the previously nanobody-stabilized active conformation (KOR-Nb39)²¹ – does this limit novelty here in the study?

The nanobody 39 (Nb39) is known as a G protein mimetic that could stabilize an active-like state KOR. Thus, the overall conformation of KOR-Nb39 shares similarities with the active-state KOR-G protein complex. However, there are several novelties from this study compared to the previously published KOR-MP1104-Nb39 work. First, the structures of KOR in complex with different G proteins provide us an opportunity to understand the G protein subtype selectivity considering that these G proteins are endogenous signal transducers downstream of KOR activation. Second, although both stabilize an overall active-like state, there exist novel differences in receptor conformation. Specifically, these observed atomic differences between G protein-coupled KOR and Nb39-coupled KOR, which emphasizes the importance of obtaining structures of KOR complexed with G protein signal transducers. Third, we revealed the binding poses of two highly selective KOR agonists at KOR (momSalB and GR89,696 vs. nonselective MP1104 in KOR-Nb39), which helps elucidate the structural determinants for KOR’s ligand selectivity.

Interactions of KOR with hallucinogen salvinorins.

1. A continuous use of the phrases “Significant effect” but no stats.

We apologize for the missing information. As pointed out above, we have gone through the manuscript to include more rigorous statistical analyses where appropriate. You may now find requisite statistical analysis in the Figures and Tables, and we have modified our descriptions in the manuscript accordingly.

2. Expression data –what does this mean for mutants with higher levels of expression? How is this factored into the data?

We have re-examined the expression levels of all mutants (KOR or G protein mutations) in a setting consistent with the cAMP inhibition or BRET-G protein assays. We measured the cell surface expression of KOR mutants by ELISA or the luminescence counts of G α mutants 48 hours after transfection, which is the same time-point as we measured

mutational effects and drug responses. The primary goal of the expression measurement was to test whether the mutants maintain robust expression levels.

To test whether the changes of expression levels affect the KOR-G protein coupling (**Figure 1**), we transfected the HEK 293T cells with different amounts of KOR or G α proteins (**Figure 2**), and we observe similar potency and efficacy compared to the group we used in the manuscript. We have included this analysis in the manuscript and have also pointed out the cell lines and assay system we used.

3. Figure 2C – why is D138 not shown on the figure?

We have now included the D138 in the Fig. 2c.

4. Extended Figure 4 - No consistency for the color scheme of the mutants

We thank the reviewer for pointing out this. We have made the color scheme of mutants consistent between the cAMP assays and the expression patterns. We have also reorganized the signaling curves to ensure that each curve is clearly visible.

5. Extended Figure 5 - How did the authors select the mutants

Generally, we focus on those residues that are within 4.5 Å of the ligand, which suggest they may form important interactions for ligand binding and signaling. Specifically, we targeted those residues that differentially engage in the interaction with momSalB and GR89,696. We then performed mutagenesis studies based on the side-chain subtypes.

Why D138N but not mutate to A like the other positions

There are several reasons we specifically discussed the D138N mutation here. First, D138N has been known as the key mutation in the KOR DREADD that abolishes the binding of endogenous dynorphin binding but maintains or improves the binding of salvinorin A or B, respectively (Vardy et al., Neuron 2015). We now have the structural evidence to explain this interesting observation. Second, compared to Ala (A) mutation that removes the whole side chain, Asn (N) maintains most of side chain of Asp (D). Asn (N) with an amide is neutral compared to Asp (D) with a carboxylic acid which is negatively charged. This also allows us to evaluate the role of salt-bridge interaction more closely. Mutation to Asn (N) likely switches the previous electrostatic repulsion (between Asp and momSalB) to attraction (between Asn and momSalB), which could form new interactions and improve the binding affinity or signaling potency observed from salvinorin A or B. Third, the D138A effect has been well studied in previous studies on inactive-state and active-state KOR structures, which also did not affect the binding or signaling of salvinorin ligands (Wu et al., Nature 2012; Che et al., Cell 2018).

We have added this information in the manuscript “As salvinorin ligands (e.g., momSalB) lack the basic nitrogen atom, there are no attractive electrostatic interactions observed between the salvinorins and D138^{3,32}. Indeed, neither D138^{3,32}A nor the D138^{3,32}N (the mutation in KOR DREADD³²) showed detrimental effects in the binding affinity or agonistic potency of SalA, whereas both mutants abolished the interaction with endogenous dynorphin ligands³²⁻³⁴. The mutation D138^{3,32}N reduced the potency of GR89,696 by 1000-fold but had minimal effects on momSalB (Fig. 2b).”.

Why do they only select D138 and H291 for double mutation?

Whether forming an H-bond or salt-bridge interaction with D138 is one of the major differences between salvinorin and other typical KOR agonists (e.g., U50,488 or GR89,696). H291A is one of the mutations that reduces momSalB's potency more than 100-fold in G protein activation. Our result showed that the double mutation (D138N+H291A) can rescue the potency in H291A alone. We have also tested several other mutations (V108A, V230A, Q115N, and M142A) in combination with the D138N (**Figure 5**). We found that both Q115N-D138N and V230A-D138N could slightly rescue the potency of momSalB by 2-3 fold compared to Q115N or V230A alone. While M142A-D138N or V108A-D138N has minimum effect or slightly decreases the potency of momSalB, these double mutants abolished the agonist activity of U50,488 or further decreases the potency of GR89,696 by 10,000-fold. The phenomenon that D138^{3.32}N did not affect or could further increase the potency of momSalB is likely due to the switch from electrostatic repulsion to attraction resulting from the new H-bond interactions between the side chain of mutated asparagine and methoxyl oxygen of the ligand. We have included these results in the Extended Data Figure 5.

Figure 5. Molecular determinants of momSalB agonism. The positive effect of additional D138N mutation on the momSalB-mediated cAMP inhibition through KOR. D138N was tested in combination with the mutation KOR-H291A, V230A, M142A, or Q115N, respectively. The additional D138N mutation did not rescue U50,488 or GR89,696-mediated cAMP inhibition. KOR V230A-D138N or H291A-D138N led to an inactive U50,488 or a significant loss of potency for GR89,696 in V230A-D138N (11,000-fold) or H291A-D138N (9300-fold) mediated cAMP inhibition. Data represent the global fit of grouped data \pm s.e.m. from $n = 3$ independent biological replicates performed in triplicate. Here we use Salvinorin A as a reference because U50,488 is inactive in the double mutants.

What is meant by “high vulnerability of momSalB”?

Based on the Fig. 2d and Extended Data Figure 4c in the manuscript, the mutagenesis screening showed that the binding pocket mutations have larger effects on momSalB-mediated cAMP inhibition than GR89,696 does (e.g., for H291A, $\Delta\text{LogEC}_{50}^{\text{mutant-wt}} = 2.15 \pm 0.25$ (momSalB) and 0.65 ± 0.27 (GR89,696)). We hypothesized that this might be due to the lack of H-bond interactions with the D138 in momSalB, as supported by the double mutation studies in **Figure 5**. We have also modified this description in the manuscript Page 7 to avoid the confusion.

6. How do the authors conclude that “This ligand-binding similarity, irrespective of G protein subtypes, is partially due to the nucleotide-free state of Gasubunits”

We have updated this part to clarify the point we would like to convey: **the overall similarity of the receptor conformation, irrespective of G protein subtypes, is partially due to the nucleotide-free state of G α subunits**. These nucleotide-free state of G α subunits (G α i1, G α oA, G α z, and G α g) tend to stabilize a specific conformational state of KOR. In the absence of G proteins or in the presence of nucleotide-bound G proteins, the receptor can adopt dynamic conformations different from that captured by nucleotide-free G α . For example, in the presence of GDP, the KOR-G-protein complex becomes more dynamic and could have different TM6 conformations in the intracellular region of the receptor when compared to the presumably more stable nucleotide-free G α heterotrimeric state (Gregorio et al., Nature 2017). This is also indirectly supported by our pharmacological evidence in Fig.5 that shows the presence of GDP reduces the KOR-G-protein stability and agonist binding.

This part has been included in the Discussion section Page 17-18.

Structural basis of G protein subtype selectivity

1. The authors use “BRET-based transducerome profiling” what are these reporters, the methods makes no mention of the Rluc8 in the constructs, the different G β and G γ used. What are the reporters?

We thank the reviewer for pointing out the missing information. We have included more details of the plasmid information and assay protocols in the Methods section.

Briefly, the transducer screening was based on a luminescence-based BRET2 reporter assay (**Figure 2A**). HEK 293T cells were transiently transfected with 4 plasmids, human KOR, G α (Gi1, GoA, Gz, or Gg)-Rluc, G β 1, and G γ 2-GFP. The G α is tagged with a luciferase (Rluc) and G γ 2 is tagged with a GFP. The insertion sites for Rluc in G α and GFP in G γ were previously described in Olsen et al. (Olsen et al., Nature Chem Biol 2020). The BRET2 assay measures the proximity of G α and G β 1G γ 2 subunits (**Figure 2A**). In the absence of agonists, KOR remains inactive, and G α G β 1G γ 2 remains as a heterotrimer complex and produces high BRET signal. After adding the agonists, the change of BRET ratio (GFP/Rluc) was recorded as the agonist activates the receptor and causes the dissociation of G α from G β 1G γ 2.

2. There is inconsistent use of reference compounds, colours in figures, symbols etc. Why do they use U50,488 for Gi1-3, GoA, GoB and Gz but use GR89,696 for Gg? Why was the same compound not used to test all G protein complexes first?

We have updated the reference to U50,488 in the cAMP inhibition and BRET-G protein assays. As indicated in a few situations, the mutations have made U50,488 inactive at KOR, in which other ligands were chosen as a reference ligand.

We did test U50,488, momSalB, and GR89,696 in parallel with all four G protein subtypes (**Figure 4A**). We have now included them in the manuscript as Fig. 3a.

We have also made the colors and symbols consistent between different plots.

3. Extended Figure 6, reference is U50,588 instead of U50,488

We thank the reviewer for pointing out this error. We have corrected the y-axis labeling as U50,488.

4. Extended Figure 7, use U50,488 as reference for Gg

We have updated this in the Extended Data Figure using U50,488 as the reference.

5. Why were not all the mutants shown in cAMP assay/cell surface expression tested in the G protein assay

This work focused on two binding pocket residues: the orthosteric (ligand binding) pocket and the G protein binding pocket. To study the effects of orthosteric site on the ligands' overall functional activity, we utilized the cAMP inhibition assay and compared the ligand's potency in G protein mediated cAMP (nonspecifically) between wild type and mutants. To study the effects of G protein binding pocket, besides the cAMP inhibition assay, we also performed BRET2-G protein assays to see whether the residues in the KOR-G interface play different roles in individual G protein coupling.

i. Where is R170A?

We have added the KOR R170A data to the Extended Data Figure 9 and 10.

b. Representation of the G protein assay data

The data represents three independent biological repeats, and in each single repeat each compound or construct was tested in duplicate or triplicate.

i. Extended Figure 6 and extended Figure 7b are not informative at all

One of the primary goals of this work was to identify the structural basis for G protein coupling and subtype selectivity. We specifically focused on the residues that are involved in the KOR-G protein interactions. Residues from both the KOR side and the G α protein side have been analyzed. We found: 1) mutation of the residues in the KOR interface reduced G protein coupling in a similar manner, except some (KOR-R156 and KOR-N336A) displayed subtype selectivity. 2) mutation of the residues in individual G α did not affect the G protein coupling as significant as the KOR residues did, except the C351A in Gi1/oA/g or I352A in Gz that led to an 8-fold loss of potency for momSalB or GR89,696. Interestingly, the Cys in all Gi/o subtypes except Gz can be ribosylated and thus inactivated by pertussis toxin (PTX).

We have also performed molecular dynamics simulation on specific residues to confirm their involvement in KOR-G protein interaction.

c. Consistency of the color

i. Figure 4

1. For b and c, colors are correlated with G protein

2. For d and e, not correlated here

We thank the reviewer for pointing out this. We have updated the color of Figure 4e to be consistent with the G protein structures.

d. What is the rationale of the choosing mutants"

i. For the KOR

1. What was the rational for showing the data in Figure 3c-f?

Fig. 3c-f in the manuscript shows the KOR-G protein interface residues that are potentially important for KOR-G protein coupling and signaling. As shown by each complex structure, the $\alpha 5$ helix in each G protein forms the major interactions with the receptor. Our mutagenesis studies were based on the interaction pattern shown in Fig. 3c-f. We have also included a 2D figure as Extended Data Fig.7 that shows the atomic distances of the potential interactions.

a. Only select the "working" ones?

As shown in Fig. 4a in the manuscript, first we unbiasedly examined the interface residues located in both KOR and G protein sides. The results showed that most of the mutations affected all four G protein subtypes in a similar pattern, whereas there were some differentially affecting individual G protein interaction, and were highlighted in Fig. 4b-4e in the manuscript.

ii. For G proteins,

1. GoA, why do they use G350Y but not G350A?

We have included new data of G350A in the GoA mutagenesis screening (Figure 6). We have also included the data in Extended Data Figure 13 in the manuscript.

Figure 6. Effects of the $G\alpha A$ - $\alpha 5$ helix residues on KOR and $G\alpha A$ coupling. The nonconserved residues were mutated to Ala and were then characterized by BRET2 assays. $G\alpha A$ has an Ala at position 345, and was mutated to the corresponding residue Gln (Q) in $G\alpha z$. Curve graphs represent the global fit of grouped data \pm s.e.m. from n=3 independent biological replicates performed in triplicate.

a. Also, in extended Figure 7 c, only GoA has a different color scheme for the mutants from other G proteins

We have modified the color of GoA curves to be consistent with other G protein subtypes. We also normalized the color between signaling curves and expression patterns.

6. For most of the graphs it is hard to see any differences with the symbols used. No more than 6 lines should be used on graphs for easy of viewing.

We have re-organized the data to make sure that the curves are clearly visible in each plot.

7. The sentence “Unexpectedly, R1563.50A conversely affects the BRET signal during KOR-Gg coupling, likely by slowing down Gg dissociation and thus leading to accumulation of stable KOR-Gg complexes.” How is this concluded? Why not show the kinetic traces for the data?

We thank the reviewer for this point. We followed the advice and have now included BRET2 kinetic data regarding KOR WT-Gg or KOR-R156A-Gg activation and confirmed that KOR-R156A led to a decreased efficacy of momSalB (**Figure 7A**). We found that the KOR-R156A reduced the potency of momSalB-mediated Gg activation (106 ± 19 nM (R156A) vs. 1.27 ± 0.26 nM (WT)), an effect similarly observed in the other Gi/o subtypes. Different from the other subtypes, KOR-R156A also decreased the efficacy of momSalB (27 ± 3 % (R156A) vs. 104 ± 4 % (WT)) (**Figure 7B**). The similar effect of KOR-R156A was also observed in U50,488 (see below) or GR89,696 (Fig.4c in the manuscript) -mediated Gg activation. To further substantiate these observations, we have tested two different sets of G protein combinations, G β 3-G γ 1 (**Figure 7B**), and G β 1-G γ 2 (**Figure 7C**).

Figure 7. The effect of KOR-R156A mutation on KOR-mediated Gg activation. A. The kinetic traces of KOR WT- or KOR-R156A-mediated Gg activation in the presence of 10 μ M momSalB. The kinetic assay was set up using the BRET2 assays except that the cells were treated with a single dose of agonists and luminescence was acquired as a function of time. Data represent the grouped data \pm s.e.m. from n=3 independent biological replicates. **B and C.** BRET2 assays showed that KOR-R156A reduced potency and efficacy of U50,488- or momSalB-mediated Gg activation. The G β 1G γ 2 (used in the KOR-Gg structure complex) or G β 3G γ 1 (reported by Olsen et al., Nat Chem Biol 2020) were tested. Curve graphs represent the global fit of grouped data \pm s.e.m. from n=3 independent biological replicates performed in triplicate.

8. The sentences: “However, we did not observe significant changes in potency in agonist-mediated G protein activation (Extended Data Fig. 7a and 7b). The expression levels of G α mutants were found to be comparable to those of the wild-type G α , as quantified by the luminescence they produced in the BRET assays” I do not believe this without statistical analysis as discussed above.

As shown in **Figure 1** and **2** above, we have re-examined the expression level of G protein mutants following the steps we performed the BRET-G protein assays. We measured the cell surface expression 48 hours after transfection. We have also included the statistical analysis using the wild type G protein as a reference. The new data is now included as Extended Data Figure 14 in the manuscript.

9. The sentence “The high-affinity binding sites for 3H-U69,593 were increased 38-, 20-, 9-, and 3-fold in the presence of Gi1, GoA, Gz, and Gg, respectively.” How can this be concluded without statistical analysis?

The saturation binding plots (Fig. 5b in the manuscript) in the presence of Gi1, GoA, Gz, or Gg were from four independent biological repeats performed at different time. In each repeat, each condition (KOR+buffer, KOR+Gi1, KOR+GoA, KOR+Gz, or KOR+Gg) was tested in duplicate by row. The saturation binding curves shown in the manuscript Fig.5b were from the global fit of grouped data from n=4 independent biological repeats. Full quantitative parameters from this experiment are derived from the Prism and listed in Extended Data Table 7. The fold changes represent the ratio of each Bmax (KOR+Gi1, KOR+GoA, KOR+Gz, or KOR+Gg) compared to the control (KOR+buffer).

10. The authors state “The calculated KB and α -cooperativity displayed a similar pattern to that observed in the saturation binding, in which Gi1 has the highest binding affinity and allosteric effects at KOR in the presence of agonists, and Gg has the least (KB, nM= 383(Gi1), 563(GoA), 1023(Gz), 1587(Gg); Log α = 1.41(Gi1), 1.21(GoA), 1.06(Gz), 0.89(Gg)) (Fig. 5d).” How were the Log α calculated? What model was used? Why is there no error? How can they conclude anything about how important these values are without stats?

Conclusion: This section cannot be reviewed objectively since the lack of statistical analysis means that any conclusions the authors make are subjective and do not stand up to rigorous questioning.

We thank the reviewer for pointing out this missing information. The saturation binding in the manuscript Fig. 5b and 5c were from four biological repeats and each repeat was tested in duplicate. The saturation binding curves in Fig.5b were from the global fit of grouped data from n=4 independent biological repeats. The bar-graph in Fig.5c represent mean Bmax \pm s.e.m. from n = 4 independent biological replicates.

Figure 5d was from three independent biological repeats and each repeat was tested in duplicate. Plots are global fit of grouped data \pm s.e.m. of n = 3 independent biological replicates. The competition binding assay was conducted in the presence of different concentration of G proteins as indicated. The curve was then plotted in Prism software using an allosteric IC50 shift mode (A. Christopoulos and T. Kenakin, Pharmacol Rev, 54: 323-374, 2002) as shown below. An allosteric modulator can reduce (negative allosteric modulator) or enhance (positive allosteric modulator) agonist binding. This model fits

dose-response curves in the absence (0 nM) and presence (1.9, 19, 190, and 1900 nM) of modulator to determine the affinity of the modulator binding, as well as the ternary complex constant α .

$$Y = Bottom + (Top - Bottom) / (1 + 10^{(LogIC50 * \frac{1 + \frac{C}{KB}}{1 + \alpha * \frac{C}{KB}} - X) * Hillslope})$$

X is the log(concentration) of agonist

C is the concentration (not log) of modulator, entered in column titles

IC50 is the concentration of agonist that inhibits half maximal response in the absence of modulator.

KB is the equilibrium dissociation constant (Molar) of modulator binding to its allosteric site.

α is the ternary complex constant. When $\alpha=1.0$, the modulator won't alter binding. If alpha is less than 1.0, then the modulator reduces ligand binding. If alpha is greater than 1.0, then the modulator increases binding.

Top and Bottom are plateaus in the units of the Y axis.

The $\text{Log}\alpha$ and LogKB values were obtained by the global fit of the above equation in Prism. Full quantitative parameters from this experiment are listed in Extended Data Table 8. We have also added this equation to the Methods section.

Methods: Lack of any data analysis, models used and of course no mention of statistical analysis. The methods lack any detail for how to reproduce the study for example, we are not told any information about the “BRET-based transducerome” except for a passing reference to the TRUPATH system. When one looks at the methods section the we are told that “HEK239T cell were transfected with a 1:5 of KOR : (G α (Gi1, GoA, Gz)-G γ -G β DNA or a 6:5 ratio of KOR : G γ -G γ -G β DNA.” There is no mention of Rluc or YFP for the BRET expts and no explanation as to why different ratios were used. If this is TRUPATH, which G γ -G β subunits were used? If not, then more detail is needed on these “BRET-based transducerome”. This is very important because we are required to compare the reporter results, yet we have no information about the constructs.

Full quantitative parameters from all plots and bar-graphs are listed in the Extended Data Tables. We have performed statistical analysis for each bar-graph and all data in the Extended Data Tables.

We have also added the information of construct subtypes and the details of assay protocols to the Methods section, as highlighted by red.

We have performed KOR and G γ BRET assays at both 1:5, 3:5, and 6:5 ratio, which displayed similar EC50s as shown below (**Figure 8**). The reason we used different ratio is that we observed better signal-to-noise ratio when using higher concentration of the receptor. We now present the 1:5 ratio results for consistency.

Figure 8. Effects of receptor concentrations on KOR-Gg coupling. Higher concentrations of KOR could improve the signal-to-noise ratio. KOR:G-protein ($G\alpha_g$, $G\beta_1$, and $G\gamma_2$ have equal amounts in each transfection) were tested at 1:5 (KOR(100ng): $G\alpha_g$ (500ng), $G\beta_1$ (500ng), and $G\gamma_2$ (500ng)), 3:5, or 6:5 ratio in a 10-cm plate. Curve graphs represent the global fit of grouped data from n=3 independent biological replicates performed in triplicate.

Referee #2 (Remarks to the Author):

This is an interesting manuscript that has determined the active state structure of the kappa opioid receptor with multiple G protein heterotrimers. These structures show differences for different chemotypes and provide insight into the molecular requirements for G protein selectivity.

The work is highly original and this reviewer is unaware of a previous Cryo-EM structure of the kappa opioid receptor. The data and methodology are appropriate and valid. The use of statistics also appears appropriate. One issue that is concerning is there is no mention where the compounds were procured or their level of purity. At a minimum, this information should be added to the supplemental material.

We thank the reviewer for pointing out the missing information. The compounds (-)-U50,488 (Cat#0496) and GR89,696 (Cat#1483) were purchased from Tocris. momSalB was a gift from David E. Nichols at University of North Carolina Chapel Hill. Briefly, momSalB was synthesized via the method described by Lee et al., *Bioorg. Med. Chem. Lett.* 15: 3744-3747 (2005). After purification by silica gel column chromatography, it was a single spot on TLC (silica/20%EtOAc/DCM) with an R_f of 0.49. We have also performed NMR to confirm the chemical structure and purity, as shown below (**Figure 9**), which is consistent with the expected spectrum by Munro et al. *Bioorganic & Medicinal Chemistry* 16 (2008).

We have also added this information to the Methods section, and now include Dr. David E. Nichols as co-authors in the manuscript. The NMR spectrum of momSalB has been included as the Extended Data Figure 21 in the manuscript.

Figure 9. NMR spectroscopy of momSalB. **A.** NMR spectrum of the momSalB used in the KOR-G protein assembly. **B.** NMR spectrum of the momSalB reported by Munro et al. *Bioorganic & Medicinal Chemistry* 16 (2008).

The following points are raised to improve the work presented:

1. Page 3, Lines 62-65: A citation from 2011 (ref 13) does not qualify as "recently".
We have removed "recently" from the sentence.

2. Page 5, Line 90: Reference 18 is not appropriate. It does not show that momSalB is psychotropic. A more appropriate reference would be Baker 2011. The ref 18 reports the synthesis of MOMSalB.
We have replaced the ref 18 with the new reference (Peet MM and Baker LE, Behav Pharmacol 2011).

3. Page 6, Line 122: Reference 25 is not appropriate. This reference reports biological activity. A more appropriate citation would be to the first isolation which is Ortega et al 1982. Suggest including both.
We have added the reference (Ortega A et al., J. Chem. Soc. 1982) to the suggested place.

4. Page 8, Line 168: The structure of salvinorin B is incorrect in the sentence and in extended data fig 5c and 5d. Salvinorin B contains an OH group, not OCH₃. Similarly, the groups for momSalB and (OCH₂OCH₃) and ethoxySalB (OCH₂OCH₂CH₃) need to be corrected in the text.
We thank the reviewer for pointing out his error. We have corrected the chemical structure (**Figure 10**) in the Extended Data Figure 5d. In the main text, we have also corrected the SalB formula and subscripts: SalB (-OH), momSalB (-O-CH₂-O-CH₃), and ethoxymethyl SalB (-O-CH₂-O-CH₂-CH₃).

Figure 10. The chemical structures and binding poses of salvinorin B, momSalB, and EOM-SalB. Differences are highlighted by red color. The agonist activity of each analog is shown in the parentheses. EC50 values were taken from Kivell et al., *Adv Pharmacol* 2014. Binding poses of SalB and EOM-SalB at KOR were revealed by molecular docking performed in the Schrodinger Maestro. The three ligands occupy a similar binding pocket with different extensions toward the hydrophobic pocket.

5. Pages 15-18, Lines 331 - 405: The manuscript contains little discussion of the present work is similar or different than Vardy et al 2013 which described the chemotype-selective modes of action of kappa agonists.

We would like to thank the reviewer for this suggestion. We have included more analysis in the Discussion (Page 18-19) by comparing the momSalB pharmacology/structure of this study with the mutational results/model reported by Vardy et al., 2013. Briefly, the mutational effects of binding pocket residues on salvinorin A (SalA) maintain on momSalB, which is attributed to their similar chemical scaffold. For example, they showed that D138A did not significantly affect the EC₅₀ of SalA-mediated-cAMP inhibition assays (9.6 ± 0.13 vs. 9.87 ± 0.05 in wild type), whereas D138N further increased the potency (10.6 ± 0.13 vs. 9.87 ± 0.05 in wild type). Several hydrophobic residues, such as KOR V108 and I316, have also been identified as key residues for SalA's agonist activity at KOR in their pharmacological studies, which is consistent with our observation that these residues directly interact with the momSalB in our structures. It is interesting that, compared with their reported SalA model, the binding pose of momSalB in KOR shows a 180° horizontal flip-over (**Figure 11**), which highlights the importance of the KOR-momSalB structure in this study.

Figure 11. Comparison of momSalB and Salvinorin A binding poses. **A.** The binding pose of momSalB in KOR-Gi1 structure. The density map of the momSalB was shown in the top-left. **B.** The binding pose of salvinorin A by molecular docking (adopted from Vardy et al., J Biol Chem 2013). Different binding poses are observed between momSalB and Sal A.

Referee #3 (Remarks to the Author):

In their manuscript “Ligand and G Protein Selectivity in Kappa Opioid Receptor Revealed by Structural Pharmacology”, Han and coauthors provide novel insights into the selective G protein coupling for the kappa opioid receptor (KOR). The recent advances in single particle cryo-EM have allowed structure determination of a number of individual GPCRs in complex with distinct classes of heterotrimeric G proteins (such as US28-Gi1 and -G11, GP139-Gs and -Gi1, NK1R-Gq and -Gs, MRGX2-Gi1 and -Gq, CCK2-Gi2 and -Gq, CCK1-Gs, -Gi1 and -Gq, GCGR-Gs and -Gi). While these structures provided important insights into the molecular determinants of G protein coupling of different G protein families to GPCRs, there is no clear consensus sequence or structural feature known that clearly determines G protein coupling specificity. Even less is known for the selective coupling of a single GPCR to different members of the same G protein family. To my knowledge only one receptor ($\alpha 2B$) has been structurally described in complex with two distinct G protein subtypes (Gi1 and GoA) from the Gi/o G protein family. On this background, the authors determined four cryo-electron microscopy structures of the fully active state of KOR in complex with four different G protein subtypes (Gi1, GoA, Gz and Gg) and bound to the psychotropic salvinorin analog momSalB or the highly potent KOR agonist GR89,696. Based on the obtained structural insights, the authors propose molecular determinants critical for subtype-selective ligand binding and G protein subtype specificity. Furthermore, the authors provide comprehensive mutagenesis, cell signaling and ligand binding studies to support their structural findings. The major success of this study is that it provides four high resolution structures of the KOR in complex with four different G protein subtypes. Two of these complexes provide a first structural view on Gz and Gg subtype coupling to a GPCR. Another important finding is the structural determination of the binding pose of a salvinorin A analog (momSalB) to the orthosteric ligand binding site of the KOR. Salvinorin A is a unique naturally occurring hallucinogenic ligand that binds very selective to the KOR without containing a single nitrogen atom. The KOR structures bound to momSalB and the selective agonists GR89,696 thus provide important insights into ligand binding specificity at the KOR, a promising target for the treatment of pain and addiction. Overall, this is an interesting study that will be of broad interest in the GPCR field.

While the conclusions of the study are sound, some caveats need to be addressed:

Specific comments:

1) While previous studies have shown that KOR can couple to Gg in transfected HEK293 cells (Olsen et al., 2020) and when expressed in bitter taste cells of genetically engineered mice (Mueller et al., 2005), it is not really known, if this interaction is of physiological relevance. As stated by the authors, the KOR is mainly expressed in the brain in pain-related neurons, whereas Gg is mostly found in taste receptor cells and in the gut. This should be discussed in the manuscript, especially with regard to the mutagenesis analyses that show the biggest effects for Gg in comparison to the other Gi/o family subtypes.

We agree with the reviewer that our pharmacological and structural data support that the KOR can interact with Gg, but cannot be extrapolated to in vivo without further experiments. We have expanded the Discussion section and point out the potential limitations of our study. As the reviewer points out, there are very few studies on the Gg interaction with the non-taste GPCRs. We thus use a NIH-based online resource (pharos.nih.gov) to examine the locations (e.g., tissues) that may co-express KOR and Gg. As shown in **Figure 12** below, while KOR (OPRK1) is highly expressed in the central nervous system, it is also expressed in the peripheral systems, such as Urogenital system and Gastrointestinal tract, which is consistent with the frequently observed diuretic effects from KOR agonists. On the other hand, Gg (GNAT3) is highly expressed in the taste and gut system, as well as the central nervous system. Again, this expression pattern together with previous study (Mueller et al., Nature 2005) support that KOR and Gg may co-localize and interact with each other, but further evidence is needed on the single-cell or single-neuron level to validate whether the KOR-Gg interaction is involved in KOR-mediated behavioral responses.

Using Gi1 as a template, the sequence identity of the other Gi/o subtypes is Gi2 (87.89%), Gi3 (93.79%), GoA (72.75%), GoB (72.39%), Gz (67.23%), and Gg (68.36%). Two further G protein subtypes, G_{S(s)} and Gq, are 39.34% and 53.22% respectively. The pharmacological data showed that Gg is the weakest coupled transducer compared to other Gi/o subtypes (Olsen et al., Nat Chem Biol 2020), and our mutational data showed that Gg is more susceptible to KOR mutations, suggesting its non-canonical role in KOR-mediated signaling.

Figure 12. The tissue-dependent expression patterns of human KOR (OPRK1) and human Gustducin (GNAT3). Data resources were adopted from a NIH-based online resource (pharos.nih.gov). Tissues with top expression levels of each protein were listed. Tissues that express both KOR and Gg were highlighted.

Expression Source	Symbol	Expression Numeric Value	Expression Tissue	Symbol	Expression Numeric Value	Expression Tissue
JensenLab TISSUES	OPRK1	4.579	Whole body	GNAT3	3.509	Taste bud
JensenLab TISSUES	OPRK1	4.571	Nervous system	GNAT3	3.464	Tongue epithelium
JensenLab TISSUES	OPRK1	4.564	Central nervous system	GNAT3	3.121	Tongue
JensenLab TISSUES	OPRK1	4.555	Brain	GNAT3	2.723	Mouth
JensenLab TISSUES	OPRK1	4.553	Head	GNAT3	2.678	Vallate papilla
JensenLab TISSUES	OPRK1	4.483	Urogenital system	GNAT3	2.865	Enteroendocrine cell
JensenLab TISSUES	OPRK1	4.45	Reproductive system	GNAT3	2.863	APUD cell
JensenLab TISSUES	OPRK1	4.413	Female reproductive system	GNAT3	2.654	Head
JensenLab TISSUES	OPRK1	4.401	Internal female genital organ	GNAT3	2.56	Whole body
JensenLab TISSUES	OPRK1	4.38	Embryonic structure	GNAT3	2.508	Alimentary canal
JensenLab TISSUES	OPRK1	4.208	BTO:0000449	GNAT3	2.485	Gastrointestinal tract
JensenLab TISSUES	OPRK1	4.195	Placenta	GNAT3	2.429	Nervous system
JensenLab TISSUES	OPRK1	2.698	Forebrain	GNAT3	2.402	Viscus
JensenLab TISSUES	OPRK1	2.692	Limbic system	GNAT3	2.366	Respiratory system
JensenLab TISSUES	OPRK1	2.645	Kidney	GNAT3	2.357	Intestine
JensenLab TISSUES	OPRK1	2.645	Urinary system	GNAT3	2.352	Epithelium
JensenLab TISSUES	OPRK1	2.644	Urinary tract	GNAT3	2.337	Integument
JensenLab TISSUES	OPRK1	2.633	Excretory gland	GNAT3	2.261	Lung
JensenLab TISSUES	OPRK1	2.632	Cerebral lobe	GNAT3	2.211	Palate
JensenLab TISSUES	OPRK1	2.631	Nucleus accumbens	GNAT3	2.187	Glossopharyngeal nerve
JensenLab TISSUES	OPRK1	2.626	Gland	GNAT3	2.187	Gland
JensenLab TISSUES	OPRK1	2.6	Temporal lobe	GNAT3	2.185	Pancreas
JensenLab TISSUES	OPRK1	2.56	Alimentary canal	GNAT3	2.167	Central nervous system
JensenLab TISSUES	OPRK1	2.553	Diencephalon	GNAT3	2.155	Nerve
JensenLab TISSUES	OPRK1	2.549	Endocrine gland	GNAT3	2.144	Retina
JensenLab TISSUES	OPRK1	2.542	Cerebral cortex	GNAT3	2.119	Brain
JensenLab TISSUES	OPRK1	2.542	Telencephalon	GNAT3	2.119	Sensory cell
JensenLab TISSUES	OPRK1	2.541	Cerebral hemisphere	GNAT3	2.116	Cranial nerve
JensenLab TISSUES	OPRK1	2.533	Hypothalamus	GNAT3	2.115	Mucosa
JensenLab TISSUES	OPRK1	2.527	Gastrointestinal tract	GNAT3	2.07	Embryonic structure
JensenLab TISSUES	OPRK1	2.515	Lateral ventricle	GNAT3	2.068	Endocrine gland
JensenLab TISSUES	OPRK1	2.513	Ganglion	GNAT3	2.065	Urogenital system
JensenLab TISSUES	OPRK1	2.506	Basal ganglion	GNAT3	2.063	Small intestine

2) The authors discuss differences in the GR89,696 and momSalB-dependent generation of biochemically stable complexes, especially for Gz and Gg complexes, as possible effects of ligand-specific transducer coupling. Do the authors have cell signaling data that would support this? The data presented in Fig. 3a and 3b shows significant differences in the potency of GR89,696 for activation of Gz and Gg, but what about Gi1 and GoA? It would be helpful to test both ligands for all four G protein subtypes to support the ligand-specific transducer coupling mentioned above.

The BRET-G protein assay showed that U50,488, momSalB, or GR89,696 could activate all four G protein subtypes with different potency (Figure 13). We hypothesized that momSalB does not form a stable KOR-Gz or Gg complex while GR89,696 does was based on the in vitro observation that, during the assembly of KOR-momSalB-Gz or Gg complex, we had limited success in obtaining stable complex. which prevented further structural analysis. Particularly, during the 2D-classification of our cryoEM data, the 2D images of the G protein heterotrimer lacked secondary structure (Figure 14), suggesting poor KOR-G protein complex stability during the vitrification process. We thus proceeded with another potent KOR agonist, GR89,696.

We have followed reviewer's advice and have now included the BRET-G protein results of momSalB and GR89,696 in activating all four G protein subtypes in Fig.3a in the manuscript. We have also modified the sentence "However, cryo-EM experiments of KOR-Gz or KOR-Gg bound to momSalB only yielded low-resolution reconstructions

(~4.5-5 Å resolution) that prevented the delineation of detailed molecular interactions. Thus, we leveraged another highly potent KOR agonist, GR89,696, to obtain high-resolution structures of the KOR-Gz and KOR-Gg.”

Figure 13. BRET2 measurement of different G protein subtype activation by KOR agonists. Ligand-dependent G protein subtype activation showed that GR89,696 is more potent than momSalB or U50,488 in BRET2-based assays. Curve graphs represent the global fit of grouped data \pm s.e.m. from n=3 independent biological replicates with each performed in triplicate.

Figure 14. Comparison of 2D-classification images from the KOR-momSalB-Gi1, KOR-momSalB-Gz, or KOR-momSalB-Gg samples. The images showed that KOR-Gi1 particles were well aligned and had clear features in each class, whereas KOR-Gg and KOR-Gz particles in each class were low resolution possibly caused by the dynamic conformation of particles in vitro and/or suboptimal complex formation.

3) I am not sure what the authors want to say in the last paragraph on page 8. I totally agree that structures of nucleotide-free GPCR-G protein complexes represent a very transient state in which the conformation of the receptor on the intracellular side is stabilized by the G protein. However, I am not sure why the ligand binding pose should necessarily be different in the presence of GDP. I would rather assume that in the presence of GDP, the complex becomes more dynamic, thus allowing TM6 and other TMs to sample alternative conformational states. The authors need to clarify this part.

We strongly agree with the reviewer's comment. We have updated this part to clarify the point we would like to convey: the overall similarity of the receptor conformation, irrespective of G protein subtypes, is partially due to the nucleotide-free state of $G\alpha$ subunits. These nucleotide-free state of $G\alpha$ subunits ($G\alpha i1$, $G\alpha oA$, $G\alpha z$, and $G\alpha g$) tend to stabilize a specific conformational state of KOR. In the absence of G proteins or in the presence of nucleotide-bound G proteins, the receptor can adopt dynamic conformations different from that captured by nucleotide-free $G\alpha$. For example, in the presence of GDP, the KOR-G-protein complex becomes more dynamic and could have different TM6 conformations in the intracellular region of the receptor when compared to the presumably more stable nucleotide-free $G\alpha$ heterotrimeric state (Gregorio et al., Nature 2017). This is also indirectly supported by our pharmacological evidence in Fig.5 that shows the presence of GDP reduces the KOR-G-protein stability and agonist binding.

This part has been included in the Discussion section in Page 17-18.

Regarding the ligand binding similarity, we have compared the ligand binding poses with more details (Figure 15). We found that momSalB in KOR-Gi1 or KOR-GoA adopts similar conformation and strength (in terms of distance). Whereas GR89,696 also shows

similar conformation in KOR-Gz and KOR-Gg structures, it appears to bind stronger in the KOR-Gz than in the KOR-Gg, as the overall distances between ligand and surrounding residues are much smaller. Particularly, GR89,696 makes closer contact with the D138^{3.32} (2.9 and 3.4 Å), Q115^{2.60} (2.8 Å), and H291^{6.52} (3.3 Å) in the KOR-Gz than D138^{3.32} (3.5 and 3.9 Å), Q115^{2.60} (3.9 Å), and H291^{6.52} (4.0 Å) in the KOR-Gg. These subtle receptor contractions around the ligand binding pocket represent small but critical differences that likely contributes to GR89,696's higher potency in activation of Gz compared to Gg.

Figure 15. Ligand-specific interaction patterns in the KOR-Gi1, GoA, Gz, and Gg structures. The dash line represents the closest distance between the ligand and indicated residues. H-bond or salt-bridge interactions are shown in red. The distance cutoff is 4.5 Å.

4) The authors discuss the role of N336 in G protein coupling specificity. Based on the structure, they report that N336 does not form a H-bond interaction with the C-terminal $\alpha 5$ helix of the G protein. However, the density in this region is not great and the refined map suggest a different side chain position that would allow formation of a polar interaction with K349. Also, for the G γ complex the density does not show enough features to model the side chain orientation of N336. The authors should try to improve the map quality/model or perform MD simulations to analyze intermolecular interactions in this area .

We have followed reviewer's advice and improved the density map by performing the local refinement on the KOR-ligand region. The ligand binding pose, and modeling of side chains of KOR are now based on two maps, globally refined, and local-refined (receptor and G α - $\alpha 5$ helix) maps. The modeling of G protein heterotrimers is based on the raw and hDEEP maps. As shown in **Figure 16**, the side chain of KOR N336 in KOR-Gi1, -GoA, and -Gz maps shows complete density that allows us to correctly build the model. Whereas the side chain density of N336 in KOR-G γ is partial, we have further performed the molecular dynamics (MD) simulation to support that the side chain of N336 in KOR can form H-bond interactions with the K349 and D350 in G γ (**Figure 17**). It is also worth pointing out that, based on the MD simulations, this N336 displays dynamic interactions with the $\alpha 5$ helix of each G protein, which could be a possible reason for the different effect observed from KOR-N336A mutant.

Figure 16. Coulombic potentials of the side chain of KOR N336 residue in the structure of KOR-Gi1, GoA, Gz, and G γ . The side chain was highlighted in red dash line.

Figure 17. Molecular dynamic (MD) simulations reveal distance traces of interactions between the KOR-N336^{8,49} with corresponding G protein residues. Closest distances of polar residues N336^{8,49} in KOR with residues in G α i1, G α oA, G α z, or G α g were shown. Distance histograms for each plot were also shown in parallel. Five independent simulations of KOR-G complex (coral, orange, green, cyan, blue) are shown, spanning 0.55 μ s (KOR-Gi1, GoA, Gz) and 0.75 μ s (KOR-Gg) of cumulative time per system, with the sampling rate of 10 frames per ns, solid lines and same-color shadows representing moving average values and one standard deviation respectively from 50 frames in all cases.

5) The key point of the discussion of the interfaces between KOR and the different G protein subtypes and the comparison with the β 2AR-Gs and 5HT2A-Gq complex is not clear. Do the authors want to say that the overall size of the interacting surface area of the individual G protein subtypes is relatively conserved across different ClassA GPCRs complexes? Furthermore, is there a real correlation between the interaction area and the kinetics of G protein association and dissociation or G protein activation in general? It would be helpful to list or include a schematic figure with the interacting residues and the size of the interacting area. Since no experimental kinetic data is presented, time-resolved BRET studies could be performed to support the model of the authors that the receptor-G protein interface impact the kinetics of G protein activation.

As suggested by the reviewer, we listed the KOR-G protein interface between KOR and different G protein subtypes (**Figure 18**). The interface area was calculated by an online server PDBePISA for each structure, KOR-G α i1 (1219 Å²), KOR-G α oA (1096 Å²), KOR-G α z (1262 Å²), and KOR-G α g (1221 Å²).

We have included the details of interacting residues between receptor (β 2AR, 5-HT2A, and KOR) and G α subunit (Gs, Gq, and Gi1, respectively) (**Figure 19**). We also compared the interface area between the receptor and each G α using the PDBePISA (Krissinel, E. & Henrick, K. J Mol Biol 2007). The interface area (Å²) is 1260 ((β 2AR-Gs, PDB 3SN6); 1077 (5-HT2A-Gq, PDB 6WHA); 1219 (KOR-Gi1 structure in this study).

Figure 18. Comparison of interface details and areas between KOR and each G protein subtypes. The dash line represents the closest distance between the intracellular KOR residues and the G α residues. Red dash lines represent possible H-bond interactions. The distance cutoff is 4 Å. The interface area was calculated by the online server PDBePISA.

Figure 19. Comparison of receptor-G α interface between β 2AR-G α_s , 5-HT2A-G α_q , and KOR-G α_i1 . The dash line represents the closest distance between the intracellular receptor residues and the G α residues. The distance cutoff is 4 Å. The interface area shown in the brackets was calculated by PDBePISA. Red dash lines represent possible H-bond interactions. The KOR-G α_i1 structure is from this study; β 2AR-Gs and 5-HT2A-Gq analysis were based on the structures PDB ID 3SN6 and 6WHA, respectively.

Because the receptors and their coupling G-protein subtypes are different, we think it will be difficult to compare the kinetics of G protein activation and confirm their relations with the interface area. When we performed the time dependent BRET2 assay (**Figure 20**), Gs, Gq, or Gi1 all reach an equilibrium within a very short time range. Due to the limitation of the experimental set up and the instrument we use, it is difficult to obtain earlier time-point values and calculate the reliable dissociation rate in a high-throughput real-timemanner.

Figure 20. Time-dependent measurement of GPCR-G protein interaction. The kinetic assays were conducted using the BRET2, in which the $G\alpha$ is Rluc tagged and $G\gamma$ is GFP tagged. Cells were first transfected with receptor, $G\alpha$ -Rluc, $G\beta 1$, and $G\gamma 2$ -GFP plasmids. During the experiment day, luminescence was first acquired at the basal level (no drug treatment), then 10 μM agonists were added, and BRET signal was recorded. The net BRET ratio was calculated by the subtraction of $\text{BRET}_{(\text{agonist})} - \text{BRET}_{(\text{basal})}$. Data represent the grouped data \pm s.e.m. from $n=3$ independent biological replicates.

We then switched to an alternative BRET1-based system to study the stability of GPCR-G complex (Che et al., Nature Communications 2020) using the miniG protein (miniGs, miniGq, and miniGi1) system previously designed by Nehme et al. (Nehme et al., PLOS ONE 2018; Wan et al., J Biol Chem 2018). In this assay we examined the stability of the receptor-miniG α complex (β 2AR-miniGs, 5HT2A-miniGq, and KOR-miniGi1) in the presence of endogenous ligands or analogs. The miniGi1 in this study was generated based on the miniGi by mutating the α 5 helix back to the wild-type Gi1 sequence.

Using the BRET1 kinetic assays (**Figure 21A**), we did observe that 1) the β 2AR-miniGs complex was quickly formed upon stimulation by the agonist isoproterenol, and this complex couldn't be dissociated by the high-affinity antagonist ICI 118,551 (**Figure 21A**). Different from that in β 2AR-miniGs, the complex of 5HT2A-miniGq or KOR-miniGi1 could be dissociated by their respective antagonist risperidone or JDTC (**Figure 21A**), but with a different rate ($t_{1/2} = 8.91 \pm 0.65$ s for 5HT2A-miniGq, $t_{1/2} = 18.87 \pm 2.25$ s for KOR-miniGi1). The antagonists used in this assay displays nM inhibitory affinity against the corresponding agonist isoproterenol, 5-HT, and Dyn A 1-13 (**Figure 21B**). As a comparison, the β 2AR-Gs has a slightly larger interface area (1260 Å²) than the KOR-Gi1 interface (1219 Å²) and much larger than the 5HT2A-Gq interface (1077 Å²).

Figure 21. Time-dependent measurement of GPCR-G protein interaction. A. The kinetic assays were conducted using the BRET1, in which the receptor (β_2 AR, 5-HT_{2A}, KOR) is Rluc tagged and miniG (miniGs, miniGq, miniGi1) is EYFP tagged. Cells were first transfected with receptor-Rluc and miniG-EYFP plasmids. During the experiment day, luminescence was first acquired at the basal level (no drug treatment), then 10 μ M agonists were added and BRET signal was recorded for 60 s; then 10 μ M of antagonists were added to the cells and BRET signal was recorded for another 40 s. The half-life ($t_{1/2}$, s) was calculated using the exponential equation “one phase decay” in the prism software. Data represent the global fit of grouped data from n=3 independent biological repeats. **B.** Measurement of the antagonist activity (ICI 118,551, risperidone, and JDTC) in the respective receptor and agonists. Curve graphs represent the global fit of grouped data from n=3 independent biological replicates with each performed in triplicate.

6) Please, provide the error for the K_B and α values in Fig. 5d

We have added the \pm S.E.M. for both $\text{Log}K_B$ and $\text{Log}\alpha$ to the Fig. 5d in the manuscript.

7) Inspection of the maps and models provided to the reviewers shows that several side chains were incorrectly modeled. The entire models should be carefully checked and remodeled, but especially the extracellular and intracellular loops of the receptor and regions around the P-loop and TCAT motif of the G proteins that show rather fragmented density. Entire residues or side chains that do not show any density should be deleted, e.g. in all maps, no or very fragmented density is observed for the N-terminal residues 51-61, indicating that this region is very flexible. I would suggest to remove this part completely.

We would like to thank the reviewer for the advice to improve the models. **We have re-examined models by deleting the regions that show no density or fragmented density in the corresponding maps, including the N terminal of the KOR in each model and low-resolution regions in each G protein model.** It is worth pointing out that the final model building is guided by the density from the B-factor sharpened map, and deep enhanced sharpened maps in addition to the receptor and $G\alpha\text{-}\alpha 5$ locally refined map. For example, the side chains of a few residues in KOR may show no or poor density in the global B-factor sharpened map but show improved density in the local refined map for KOR-ligand, we still built in the side chains of those residues. We will also upload and release the local refined map to the PDB.

Minor points:

Line 110: Differences between the Nb and G protein-bound KOR structures should be described for the G protein complexes, meaning that you should write that in the G protein complex, TM6 moves 2.8Å away from TM5 in comparison to the Nb-bound state.

We have re-written this sentence as the reviewer suggested: “Notably, comparison of these two structures reveals that the intracellular end of transmembrane helix (TM) 6 in the KOR-Gi1 protein complex moves 2 Å closer to TM7.”

Line 142: Asparagine can form H-bonds but no salt-bridge interactions.

We have changed the salt-bridge interactions to H-bond interactions in this sentence: “...likely due to the switch from electrostatic repulsion to attraction resulting from the new H-bond interactions between the side chain of mutated asparagine and methoxy oxygen of the ligand.”

Line 181: I would not consider GDP to be a substrate for $G\alpha$ subunits, because they do not hydrolyze it to GMP.

We have re-written this paragraph as the reviewer suggested earlier. The ‘substrate’ has been removed in this paragraph and corrected in other places.

Fig. 2d: Please, change GR8,9696 to GR89,696

We have corrected the GR8,9696 in Fig.2d to GR89,696.

Fig. 3c-f: Please, include labels for the TMs.
 We have added TM labels to all helices shown in Fig. 3c-3f.

Fig. 4a: This figure is very crowded and hard to read. I suggest to present LogEC50 values as a bar graph similar to Fig. 2d.
 We would like to thank the reviewer for the good suggestion. We are now presenting the bar-graphs that include the LogEC50s of wild type and mutants with statistical analysis (Figure 22). We moved the dose-dependent curves to the supplemental figures and re-organize the curves to ensure that each curve is clearly presented in the plot.

Figure 22. The effect of KOR-G-protein interface residues on the G protein coupling. Mutagenesis analysis of intracellular KOR residues by the G protein-mediated cAMP inhibition assays. Data are mean LogEC50 ± s.e.m. from n = 3 biological replicates with each performed in triplicate. Statistical significance for each mutant is compared in a one-way analysis of variance (ANOVA) with Dunnett’s multiple comparison test to the wild type (* = $p < 0.05$, ** = $p < 0.01$, *** = $p < 0.001$, **** = $p < 0.0001$, “ns” represents no significance).

Line 245-248: Please, also cite Rose et al., JACS (2014) and Glukhova et al., ACS Pharm. Transl. Sci. (2018)
 We have added the two references to the suggested place.

Line 251: Please, also cite Flock et al., Nature (2017); Inoue et al., Cell (2019)
 We have added the two references to the suggested place.

Line 265: The receptor also interacts with other structural components of the G protein, including β_6 and top of β_3 . Therefore, it should be highlighted that you are focusing on α_N and α_5 only.
 We have re-written this part as the reviewer suggested: “The overall interfaces of $G\alpha_i1$, $G\alpha_oA$, $G\alpha_z$, and $G\alpha_g$ with KOR are highly conserved, but there are critical differences in

the $\alpha 5$ and αN helices. The major contacts made by $G\alpha oA$ with KOR are through residues in the $\alpha 5$ helix, whereas contacts made by $G\alpha i1$, $G\alpha z$, and $G\alpha g$ involve regions in both the $\alpha 5$ and αN helices.”

Line 356: Please change reference 58 to Wingler et al., Cell (2019)

We thank the reviewer for pointing out this error. We have changed the reference 58 to Wingler et al., Angiotensin Analogs with Divergent Bias Stabilize Distinct Receptor Conformations. Cell 2019.

Line 374-377: This has been done for the B2AR receptor. Please, cite Wenzel-Seifert and Seifert, Mol. Pharm. (2000)

We have added this reference to the suggested place.

Line 395: Please, also consider to add references Huang et al., Cell (2021) and Furness et al., Cell (2016)

We have added the two references to the suggested place.

Line 549: Please, provide the reference

We would like to thank the reviewer for pointing out the missing reference. We have added the reference to the cAMP inhibition assay section.

The model of the Gg complex contains all the hydrogens. Please, remove those from the pdb file.

We have removed the hydrogens from the Gg model, and have refined it against the maps. The new parameters are now included in the Extended Data Table 1.

Reviewer Reports on the First Revision:

Referees' comments:

Referee #1 (Remarks to the Author):

I would like to commend the authors on a substantially improved version of this manuscript.

There are a few minor points that now need clarification.

1. In the section "Overall structures of KOR-G protein complexes" the authors suggest that on Line 87: "Compared with other opioid receptors, KOR exhibits a stronger preference for Gg reference 17". Is this the correct reference since this is not shown in the Olsen paper? In the previous version of the manuscript (Figure 5f), there was data showing that the MOR does not activate Gg. Please add this data back in to show that KOR is better at Gg compared to MOR etc.
2. Ref 88 is the same as ref 17.
3. Line 107 - the authors state "alignments of the MOR-Gi1 structure²⁰ with each of the four G protein complex structures" but they only show the output for Gi1 with KOR.
4. Inconsistent use of abbreviation of Transmembrane Helix, please correct.
5. In Figure 1d the 2A seems to be in the wrong place?
6. Line 160-163 - it would be nice to have a bit better rationale for the double mutant as it is not clear to this reviewer why this was used.
7. For agonist U50,488 please provide data for the single mutation D138N - as this has been shown for the other agonists in figure 2.
8. The title for U50,488-KOR V230A cAMP data is incorrect, it shows as "momsalB-cAMP inhibition".
9. On Line 242 - the authors comment about Interaction between R156 and L354 in Gz. This is not shown in the extended figure 7-Gz. Is this due to the 4A cut off?
10. The mutation N336A has no effect on Gi1 but abolishes Gg. This seems odd since these positions are identical in the two proteins. Could the authors comment on these differences please?
11. As with comment 9, why is there no interaction between N336 and D350? Is it again the 4A cut off?
12. Extended figure 14b - there are two panels labelled BRET2-KOR-Gz? Please check this and should there be data for GR89,696?
13. The conclusion on line 310-312 seems to exaggerate the actual data. Here the authors have compared only Go (mini G) with 5HT1bR. Comparison between mini-G and full G proteins is not valid and also should include all Gi/o proteins. I would consider reducing the impact of this sentence.
14. Labels on Extended figure 15 need to read "5HT1BR" not "5HT1B"
15. Same on Extended figure 16 except 5HT2A" need "R" adding.
16. Figure 5d - are the LogKb values correct? These should be nM so how can the log value be 2.59? Please change to nM units and do not log?
17. Extended data table 8 the units are not correct - should be nM.
18. Line 383-385: Gz and Gg almost identical but 100-fold in potency - why is this the case? Please provide an explanation (this is mentioned in the results".
19. Methods: Transfection methods are unclear What reagent, amounts, mentioned in the response letter, please add to the manuscript.

20. Please describe the choice of the $G\alpha\beta\gamma$ combinations use in the BRET2 expts.

Overall this is a much improved manuscript.

Referee #2 (Remarks to the Author):

This is an interesting manuscript that has determined the active state structure of the kappa opioid receptor with multiple G protein heterotrimers. These structures show differences for different chemotypes and provide insight into the molecular requirements for G protein selectivity.

The work is highly original and this reviewer is unaware of a previous Cryo-EM structure of the kappa opioid receptor. The data and methodology are appropriate and valid. The use of statistics also appears appropriate.

The authors have effectively addressed the concerns raised previously.

Referee #3 (Remarks to the Author):

The authors have addressed most of my concerns sufficiently well and significantly improved the manuscript by performing statistical analyses of their data and additional experiments to support their claims.

The discussion of the impact of the GPCR-G protein interfaces on the kinetics of complex formation, however, is still not clear and the new BRET data provided do not necessarily support the proposed model. The differences in the observed dissociation rates between the receptor-G protein complexes of the b2AR, 5-HT2A and the KOR could be strongly influenced by differences in the agonist off-rate or antagonist on-rate of the ligands tested. This is supported by the available cryo-EM/X-ray structures of the investigated receptors in the G protein-bound active state and previous publications on the effect of G protein binding on ligand binding of the b2AR (DeVree et al., 2016; Yao et al., 2009). The slow antagonist-dependent dissociation of the b2AR-mGs complex shown in Extended Data Fig. 17a could be explained by the more closed conformation of the extracellular vestibule of the receptor when bound to Gs or the active-state stabilizing nanobody Nb80 that sterically prevents both ligand association and dissociation. In the corresponding structures of the KOR and 5-HT2A, the orthosteric ligand binding site is more open to the extracellular side and thus may allow faster ligand exchange and as a result G protein dissociation. These potential caveats should be mentioned and discussed. Furthermore, it is not explained how the dose response curves in Extended Data Fig. 17b were generated. I assume that the authors collected the kinetic BRET data in the presence of different ligand concentrations, but I am not sure how the ICI118,551 data can result in a dose response curve, when the addition of 10 uM showed no effect on complex stability and thus the BRET signal.

Minor points:

Page 7, lines 141-144: In a recently published paper by Wang et al., describe the cryo-EM structure of the KOR in complex with its endogenous agonist dynorphin and the heterotrimeric G protein Gi. In this structure, D138 interacts with the N-terminus of dynorphin. This work should be referenced in this context.

Page 14, line 308: Please, change Fig. 18a to Fig. 17a

Author Rebuttals to First Revision:

We thank great comments and suggestions from reviewers. In this revision, we have corrected errors pointed out by reviewers in the manuscript. Below you will find in light blue a point-by-point response to each of comments that you raised.

Referee #1 (Remarks to the Author):

I would like to commend the authors on a substantially improved version of this manuscript.

We thank the reviewer for great comments and recognition of our work.

There are a few minor points that now need clarification.

1. In the section "Overall structures of KOR-G protein complexes" the authors suggest that on Line 87: "Compared with other opioid receptors, KOR exhibits a stronger preference for Gg reference 17". Is this the correct reference since this is not shown in the Olsen paper? In the previous version of the manuscript (Figure 5f), there was data showing that the MOR does not activate Gg. Please add this data back in to show that KOR is better at Gg compared to MOR etc.

We would like to thank the reviewer for this suggestion. We decided to take out the MOR and Gg data in the last revision because we found that MOR and Gg interaction may be ligand specific. As the reviewer pointed out, we showed MOR does not interact with Gg in the presence of DAMGO or morphine. However, based on the unpublished data of Olsen et al., they observed weak interaction between MOR and Gg in the presence of highly potent agonists, such as BU72, fentanyl, or carfentanyl. This is a reminiscence of weak KOR-Gg interaction (efficacy <10%) mediated by Dynorphin A, although other KOR agonists robustly recruit Gg to KOR (Olsen et al., Nat Chem Biol 2020). We have modified this sentence to avoid the confusion.

2. Ref 88 is the same as ref 17.

We thank the reviewer for pointing out this error. We have corrected it.

3. Line 107 - the authors state "alignments of the MOR-Gi1 structure²⁰ with each of the four G protein complex structures" but they only show the output for Gi1 with KOR.

We thank the reviewer for pointing out this missing information. We have modified the main text to make it clear that it is the comparison between KOR-Gi1 and MOR-Gi1 structures.

4. Inconsistent use of abbreviation of Transmembrane Helix, please correct.

We thank the reviewer for pointing out this error. We have corrected the abbreviation of Transmembrane Helix to make it consistent in the manuscript.

5. In Figure 1d the 2A seems to be in the wrong place?

We thank the reviewer for pointing out this mistake. We have corrected this to the suggested place, which is now the Figure 1b.

6. Line 160-163 - it would be nice to have a bit better rationale for the double mutant as it is not clear to this reviewer why this was used.

From the single mutagenesis studies, we observed several residues (e.g., V108, Q115, M142, V230, H291, I316) had larger effects (in terms of ΔEC_{50} compared to wild type) on momSalB than GR89,696. **We reasoned that this is likely due to the lack of the anchoring interactions with D138^{3,32}, which makes salvinorins more sensitive to other residue contacts.** Forming a salt-bridge or H-bond interaction with the D138^{3,32} in KOR has been shown critical for the binding of many KOR agonists, including the GR89,696 or endogenous peptide dynorphins. **We thus focused on those specific residues and hypothesized that by introducing an additional hydrogen bond between momSalB and KOR through mutation of D138 to N138, we could rescue the loss of EC₅₀s resulting from the single mutations.** This is the rationale we examined the double mutations.

Our results showed that some double mutants (e.g., H291A-D138N, Q115N-D138N, or V230A-D138N) could slightly rescue the potency of momSalB, while others did not (e.g., M142A-D138N). This could be due to the different binding poses resulted from the mutations. Such changes cause momSalB to adopt different conformations, some of which may allow for the formation of a hydrogen bond with N138, while others may not.

7. For agonist U50,488 please provide data for the single mutation D138N - as this has been shown for the other agonists in figure 2.

We have now included the data of U50,488-mediated cAMP inhibition for the KOR-D138N mutant in the main Figure 2b.

8. The title for U50,488-KOR V230A cAMP data is incorrect, it shows as "momSalB-cAMP inhibition".

We thank the reviewer for pointing out this error. We have corrected the title "momSalB-cAMP inhibition" to "U50,488-cAMP inhibition" in the suggested place, which is now in the Extended Data Figure. 3a

9. On Line242 - the authors comment about Interaction between R156 and L354 in Gz. This is not shown in the extended figure 7-Gz. Is this due to the 4A cut off?

We thank the reviewer for pointing out this confusion. We now add distance between the KOR-R156 and Gz-L354, which is now in the Extended Data Figure. 4c. The atomic coordinates of KOR-Gz structure show that the distance between the R156 in KOR and L354 in Gz is around 4.4 Å, allowing weak hydrophobic interactions to occur, so we didn't include it originally.

10. The mutation N336A has no effect on Gi1 but abolishes Gg. This seems odd since these positions are identical in the two proteins. Could the authors comment on these differences please?

In our analysis of the KOR-G protein interface, we found that 15-20 residues, among them N336, participate in the interactions. The observed loss of Gg interaction resulting from the N336A mutation, while having only a minor effect on Gi1, indicates that N336 plays a key role in the KOR-Gg interaction. N336 likely involves in KOR-Gg interaction at an intermediate step that is not captured by our structures, as evidenced by molecular dynamics simulations that also revealed a dynamic pattern in the interactions between N336 and Gg.

11. As with comment 9, why is there no interaction between N336 and D350? Is it again the 4Å cut off?

We now add the distance between KOR-N336 and Gi1-D350. The structural snapshot of KOR-Gi1 shows the distance between N336 in KOR and D350 in Gi1 is around 4 Å. The molecular dynamics simulation provide support that the two residues can form strong interactions as evidenced by the < 3 Å distance.

12. Extended figure 14b - there are two panels labelled BRET2-KOR-Gz? Please check this and should there be data for GR89,696?

We thank the reviewer for pointing out this error. We have corrected the title "BRET2-KOR-Gz" to "BRET2-KOR-Gg" in the suggested place and added the data for GR89,696, which is now in the Supplementary figure. 5

13. The conclusion on line 310-312 seems to exaggerate the actually data. Here the authors have compared only Go (mini G) with 5HT1bR. Comparison between mini-G and full G proteins is not valid and also should include all Gi/o proteins. I would consider reducing the impact of this sentence.

In response to the reviewer's suggestion, we have revised this paragraph to avoid any potential misunderstandings. It has been proposed that different GPCR-G protein interfaces may contribute to the diverse kinetics of G protein association and dissociation with the receptor. This is a significant question that warrants further research, specifically time-resolved kinetic measurements of the KOR-G protein assembly rate.

14. Labels on Extended figure 15 need to read "5HT1BR" not "5HT1B"

We have corrected "5HT1B" to "5HT1BR".

15. Same on Extended figure 16 except 5HT2A" need "R" adding.

We thank the reviewer for pointing out this error. We have corrected "5-HT2A" to "5-HT2AR".

16. Figure 5d - are the LogK_b values correct? These should be nM so how can the log value be 2.59? Please change to nM units and do not log?

We thank the reviewer for the suggestion. We have changed it to K_B with “nM” units.

17. Extended data table 8 the units are not correct - should be nM.

We thank the reviewer for pointing out this error. We have corrected it to “nM” units.

18. Line 383-385: Gz and Gg almost identical but 100-fold in potency - why is this the case? Please provide an explanation (this is mentioned in the results).

The potency of a ligand is determined by its binding affinity and the kinetics of G protein binding. Our data suggests that two observations may explain the variation in potency between Gz and Gg. Firstly, our mapping of the ligand-KOR interactions indicates that the presence of Gz results in a smaller binding pocket for GR89,696 in KOR-Gz than in KOR-Gg. This suggests that GR89,696 binds more strongly in KOR-Gz than in KOR-Gg. Secondly, our analysis of the KOR-Gz and KOR-Gg interfaces shows that a different number of residues are involved, which may also affect the rates of G protein association and dissociation.

19. Methods: Transfection methods are unclear what reagent, amounts, mentioned in the response letter, please add to the manuscript.

We thank the reviewer for pointing out this confusion. We have included the experimental details for the transfection in the Methods.

20. Please describe the choice of the Gαβγ combinations use in the BRET2 expts.

We thank the reviewer for pointing out this missing information. The BRET2 assays presented in the manuscript followed the published TRUPATH platform (Olsen et al., Nature Chem Biol 2020). Specifically, Gα_{i1}-Gβ₃-Gγ₉, Gα_{oA}-Gβ₃-Gγ₈, Gα_Z-Gβ₃-Gγ₁, and Gα_{Gustducin}-Gβ₃-Gγ₁ were used for BRET2-Gi1, GoA, Gz, and Gg experiments, respectively. We have also tested BRET2-Gi1, GoA, Gz, and Gg using the Gβ₁-Gγ₂ combination for all Gα subtypes. While the agonists showed similar potencies to the TRUPATH combinations, their signal to noise ratio within the dataset varied. Notably, the Gα_{Gustducin}-Gβ₁-Gγ₂ combination did not produce a robust signal for some mutants. We thus present the data from the optimized TRUPATH combinations.

Overall this is a much improved manuscript.

We are grateful for the reviewer's comments, which have significantly enhanced the rigor and quality of our work.

Referee #2 (Remarks to the Author):

This is an interesting manuscript that has determined the active state structure of the kappa opioid receptor with multiple G protein heterotrimers. These structures show differences for different chemotypes and provide insight into the molecular requirements for G protein selectivity.

The work is highly original and this reviewer is unaware of a previous Cryo-EM structure of the kappa opioid receptor. The data and methodology are appropriate and valid. The use of statistics also appears appropriate.

The authors have effectively addressed the concerns raised previously.

We thank the reviewer's comments and feedback, which has significantly improved the quality of the work.

Referee #3 (Remarks to the Author):

The authors have addressed most of my concerns sufficiently well and significantly improved the manuscript by performing statistical analyses of their data and additional experiments to support their claims.

The discussion of the impact of the GPCR-G protein interfaces on the kinetics of complex formation, however, is still not clear and the new BRET data provided do not necessarily support the proposed model. The differences in the observed dissociation rates between the receptor-G protein complexes of the β 2AR, 5-HT_{2A} and the KOR could be strongly influenced by differences in the agonist off-rate or antagonist on-rate of the ligands tested. This is supported by the available cryo-EM/X-ray structures of the investigated receptors in the G protein-bound active state and previous publications on the effect of G protein binding on ligand binding of the β 2AR (DeVree et al., 2016; Yao et al., 2009). The slow antagonist-dependent dissociation of the β 2AR-mGs complex shown in Extended Data Fig. 17a could be explained by the more closed conformation of the extracellular vestibule of the receptor when bound to Gs or the active-state stabilizing nanobody Nb80 that sterically prevents both ligand association and dissociation. In the corresponding structures of the KOR and 5-HT_{2A}, the orthosteric ligand binding site is more open to the extracellular side and thus may allow faster ligand exchange and as a result G protein dissociation. These potential caveats should be mentioned and discussed. Furthermore, it is not explained how the dose response curves in Extended Data Fig. 17b were generated. I assume that the authors collected the kinetic BRET data in the presence of different ligand concentrations, but I am not sure how the ICI118,551 data can result in a dose response curve, when the addition of 10 μ M showed no effect on complex stability and thus the BRET signal.

We agree with the reviewer that this part is not fully convinced due to the limitation of experiments. Direct measurement of the KOR-G protein assembly rate is required to investigate whether the size of the interface contributes to receptor-G protein kinetics. This is a significant question that merits further study. In order to avoid any potential misunderstandings, we have revised this paragraph to provide a balanced conclusion and discussion.

To address the reviewer's questions regarding the techniques used in our study, we would like to highlight the differences in assay protocols between the original Extended Data Figure 17a and 17b. In Extended Data Figure 17a, cells transfected with β 2AR-Rluc8 and miniGs-EYFP were plated in a 96-well plate. To activate the receptor, 10 μ M isoproterenol was added and the BRET signal was recorded for 60 seconds. Following this, another 10 μ M (final concentration) of ICI 118,551 was added to the cells along with isoproterenol, and the BRET signal was recorded for another 60 seconds. **This assay measures the ability of ICI 118,551 to reverse the interaction between β 2AR-miniGs, indirectly reflecting the stability of the β 2AR-isoproterenol-miniGs complex.**

In Extended data figure 17b, cells transfected with β_2 AR-Rluc8 and miniGs-EYFP were plated onto a 96-well plate. The cells were then incubated with various concentrations of ICI 118,551 (-5, -6, -7, ...-12) for 15 minutes. Next, 5 nM of Isoproterenol, which equals to EC80 potency as an agonist, was added and incubated for an additional 5 minutes. **The resulting BRET signal was recorded to measure ICI 118,551's antagonist activity in inhibiting the isoproterenol-mediated beta2AR-miniGs interaction.**

Extended Data Figure 17

Minor points:

Page 7, lines 141-144: In a recently published paper by Wang et al., describe the cryo-EM structure of the KOR in complex with its endogenous agonist dynorphin and the heterotrimeric G protein Gi. In this structure, D138 interacts with the N-terminus of dynorphin. This work should be referenced in this context. We thank the reviewer's suggestion. We have included this reference in the manuscript.

Page 14, line 308: Please, change Fig. 18a to Fig. 17a

We thank reviewer for pointing out this error. We have corrected it accordingly.